# Cosmos-Eval: Towards Explainable Evaluation of Physics and Semantics in Text-to-Video Models

## Abstract

Recent text-to-video (T2V) models have achieved impressive visual fidelity, yet they remain prone to failures in two critical dimensions: adhering to prompt semantics and respecting physical commonsense. Existing benchmarks, including VIDEOPHY and VIDEOPHY-2, formalize these axes but provide only scalar scores, leaving model errors unexplained and hindering reliable evaluation. To address this, we present **Cosmos-Eval**, an explainable evaluation framework that jointly assesses semantic adherence and physical consistency. Cosmos-Eval produces fine-grained 5-point scores *with natural-language rationales*, leveraging the physically grounded ontology of Cosmos-Reason1 and an LLM-based rationale refinement pipeline. This enables precise identification of semantic mismatches and violations of physical laws, such as floating objects or momentum inconsistencies. Experiments on VIDEOPHY-2 show that Cosmos-Eval matches state-of-the-art auto-evaluators in score alignment (Pearson 0.46 vs. 0.43 for semantics; Q-Kappa 0.33 vs. 0.33 for physics) *while also delivering state-of-the-art rationale quality* (e.g., best BERTScore F1 and BLEU-4 on both SA and PC). Beyond this benchmark, our framework generalizes to other evaluation suites, establishing a unified paradigm for explainable physics-and-semantics reasoning in T2V evaluation and enabling safer, more reliable model development.

## 1 Introduction

Recent breakthroughs in text-to-video (T2V) generation—from diffusion-based models like Lumiere (Bar-Tal et al., 2024) and Stable Video Diffusion (Blattmann et al., 2023) to transformer-driven systems like VideoPoet (Kondratyuk et al., 2024)—have enabled realistic video synthesis. Yet today's systems are still far from acting as "general-purpose physical world simulators" (Bansal et al., 2025a): clips may look sharp but objects float, collisions miss responses, or the scene fails to reflect what the prompt describes. Importantly, evaluation protocols tell us *that* a video is wrong but rarely *why*.

A growing body of work converges on two complementary axes for judging T2V. VIDEOPHY (Bansal et al., 2025a) formalizes *Semantic Adherence (SA)*—whether entities, actions, and relations requested by a caption are grounded in the video—and *Physical Commonsense (PC)*—whether the dynamics (stability, contact, collisions, causality) are plausible even without the caption. The follow-up VIDEOPHY-2 (Bansal et al., 2025b) expands to hundreds of real-world actions and releases VIDEOPHY-2-AUTOEVAL, an automatic evaluator that outputs five-point SA/PC scores strongly correlated with human judgments, as reported in their published experiments. However, these evaluators primarily return *numbers*; they do not surface concrete evidence behind a grade, which makes it hard to diagnose failure modes or trust the assessment.

At the same time, advances in physical reasoning and multimodal explainability suggest a way forward. NVIDIA's **Cosmos-Reason1** (NVIDIA et al., 2025) organizes physical commonsense into a hierarchical ontology (e.g., conservation, object permanence, spatial/temporal relations) and demonstrates video-based reasoning. In parallel, explainable evaluation methods show that structured prompting, multi-step verification, and LLM-as-a-judge pipelines can improve specificity and reliability of textual feedback (Mou et al., 2025; Gu et al., 2024). What is missing is a unified evaluator that marries the *score fidelity* of VIDEOPHY-2-AUTOEVAL with *physically grounded rationales* that make scores interpretable and therefore actionable.

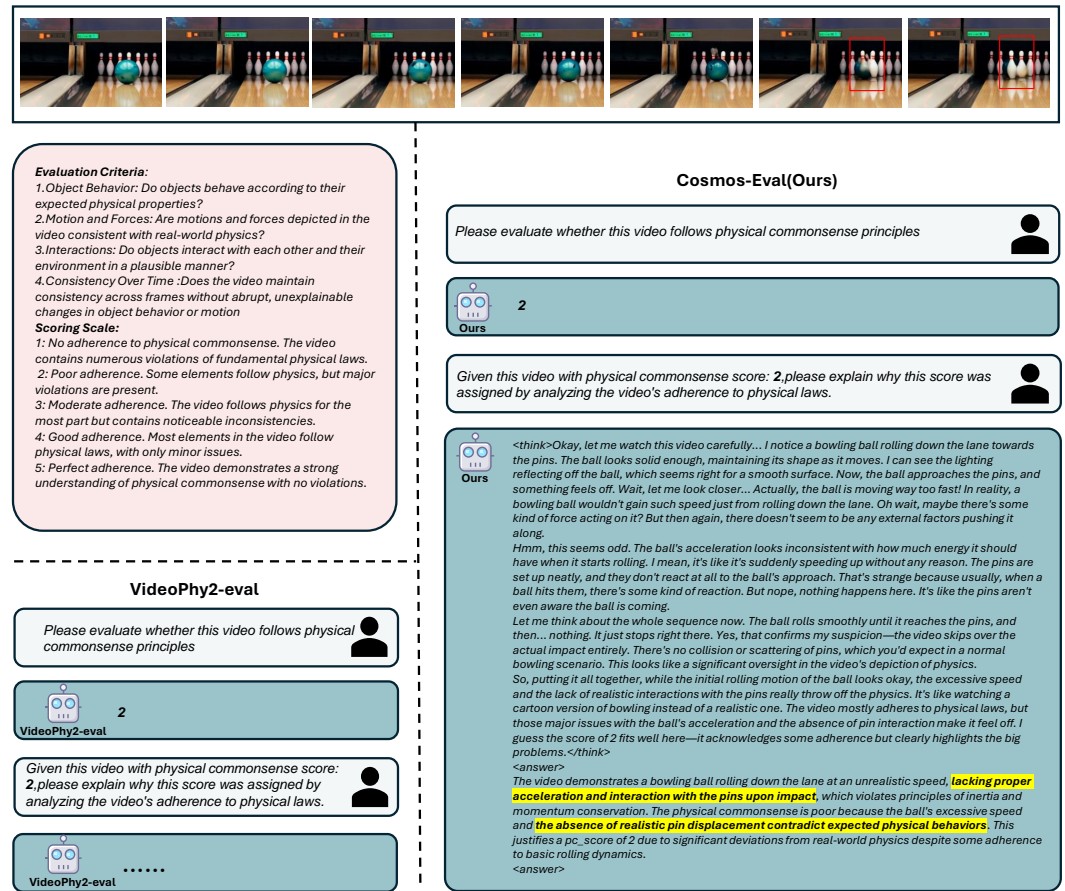

Figure 1: **Score-only vs. explainable evaluation.** Qualitative PC example: VIDEOPHY-2-AUTOEVAL outputs only a numeric score (e.g., PC= 2) without justification, while *Cosmos-Eval* augments the score with a concise, physics-grounded rationale (e.g., implausible acceleration and missing collision dynamics), improving diagnosability and trust.

**Our solution: Cosmos-Eval.** We introduce *Cosmos-Eval*, an explainable SA/PC evaluation framework that reports five-point scores *and* concise, evidence-based rationales for each test case by default. Cosmos-Eval builds on Cosmos-Reason1 to reason about physics, and uses a reference-seeded, judge-verified controller to iteratively refine rationales into an evidence-grounded chain of thought, then distills this behavior into a lightweight model for deployment. As illustrated in Fig. 1, a score-only evaluator such as VIDEOPHY-2-AUTOEVAL might return "PC= 2" for a bowling clip; Cosmos-Eval produces the same score and adds a short rationale (e.g., implausible acceleration and missing collision response), enabling concrete, actionable diagnostics.

**Core Contributions.**

- *Explainable SA/PC paradigm.* Within the VIDEOPHY/VIDEOPHY-2 setting, we pair five-point SA/PC scores with detailed rationales that support auditing, ablations, and failure localization (e.g., SA: "caption mentions a red ball, but video shows a blue cube"; PC: "object floats mid-air, violating gravity"), addressing the interpretability gap of prior benchmarks.

- *Score alignment with state-of-the-art auto-evaluators.* On the official VIDEOPHY-2 test set, our scores match VIDEOPHY-2-AUTOEVAL (SA Pearson: 0.46 vs. 0.43; PC Q-Kappa: 0.33 vs. 0.33) while adding rationales, avoiding the accuracy–interpretability trade-off.

- *Physically grounded rationale quality.* Leveraging Cosmos-Reason1's ontology and our Stage-2 controller, our rationales achieve state-of-the-art similarity to references for SA/PC (e.g., SA

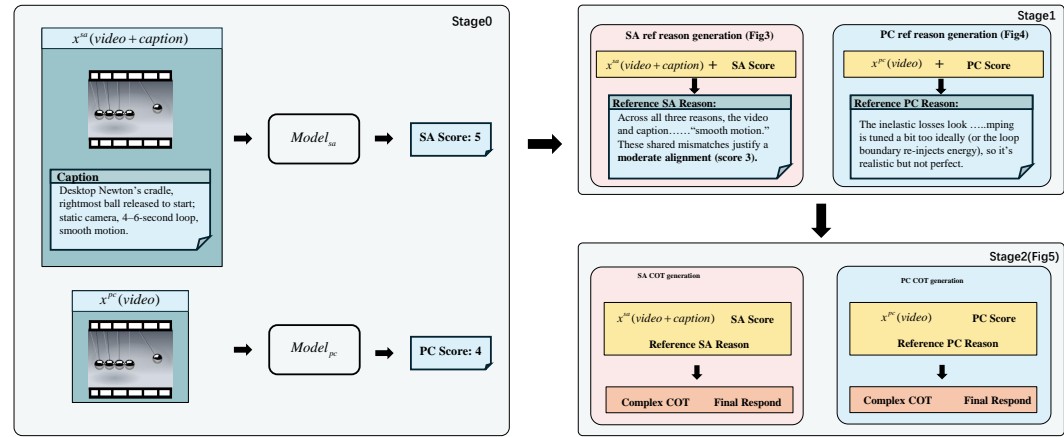

Figure 2: **Pipeline overview (Stages 0–2; Stage 3 training). Stage 0** (frozen VideoPhy scorers) maps inputs to discrete labels $s_{\mathrm{SA}}, s_{\mathrm{PC}}$ (Eqs. 1–2). **Stage 1** (reason generation) produces SA/PC reference rationales $r_{\mathrm{ref}}^{\mathrm{sa}}, r_{\mathrm{ref}}^{\mathrm{pc}}$ (Figs. 3, 4). **Stage 2** (reason-augmented CoT) uses a judge-verified controller to build evidence-grounded chains and final responses (Fig. 5). *Stage 3 (two-run SFT; training)* first fine-tunes a *score head* to predict 5-point labels $\{1, \ldots, 5\}$, then fine-tunes *rationale generation conditioned on the predicted score* with CoT-style prompting, so the system outputs calibrated scores and concise, reference-faithful explanations at test time.

BERTScore F1 52.44 / BLEU-4 26.70; PC BERTScore F1 54.50 / BLEU-4 27.86), outperforming generic VLMs (e.g., Qwen-2.5-VL on PC: 36.31 / 4.44).

- *Generalizable pipeline.* Our reference-seeded, judge-verified rationale workflow and two-run SFT are scorer- and dataset-agnostic. In this work we evaluate on VIDEOPHY-2; extending to additional suites (e.g., T2VPhysBench) is a promising direction for future validation.

## 2 METHOD

We present the pipeline in execution order: **Stage 0** (VideoPhy scorers $\rightarrow$ discrete SA/PC scores), **Stage 1** (reason generation), **Stage 2** (reason-augmented CoT), and **Stage 3** (SFT on textualized scores and Stage-2 `<think>`/`<answer>`). Stages 0–2 are generative (no parameter updates); Stage 3 sets training objectives (Sec. 4). The stages form a causal flow—*scores as priors $\rightarrow$ reference reason $\rightarrow$ evidence-verified chain $\rightarrow$ distilled model*. Removing any stage degrades this flow: omitting **Stage 0** weakens ultimate agreement with human judgments; **Stage 1** is necessary to provide a score-aligned anchor $r_{\mathrm{ref}}^{\tau}$; omitting **Stage 2** removes evidence verification and reduces rationale reliability; omitting **Stage 3** forces deployment to run Stages 0–2 online (high latency, unstable consistency). Overall, Stages 0–3 instantiate an information-theoretic pipeline (IB at Stage 0; conditional MI at Stages 1–3). Fig. 2 provides the high-level view of Stages 0–2: we first compute $s_{\mathrm{SA}}, s_{\mathrm{PC}}$ via Eqs. equation 1–equation 2 (Stage 0), then synthesize score-aligned reference reasons (Stage 1), and finally run an evidence-verified controller that yields an explicit CoT and the final judgment (Stage 2).

**Task summary (SA/PC).** Following Bansal et al. (2025b), we evaluate two axes: **SA**—given video $v$ and caption $c$, check whether key entities/actions/relations in $c$ are grounded in $v$; and **PC**—given $v$ only, judge whether the observed dynamics (stability, contact, collisions, causality) are physically plausible. Both use a 5-point integer scale $\{1, \ldots, 5\}$ and are evaluated independently (high SA need not imply high PC). Evaluations are per input instance.

**Notational conventions.** We adopt compact notation for clarity. We index tasks by $\tau \in \{\mathrm{sa}, \mathrm{pc}\}$ with inputs $x^{\mathrm{sa}} = (v, c)$ and $x^{\mathrm{pc}} = v$. Frozen VideoPhy scorers output labels $s_{\mathrm{SA}}, s_{\mathrm{PC}} \in \{1, 2, 3, 4, 5\}$. A stand-alone reason is $r$; evidence snippets $e$ appear only in Stage 2 (CoT), not in Stage 1. Task prompts are $\mathbf{P}^{\tau}$. In Stage 1 (SA) we query an ensemble $\{\mathcal{M}_m\}_{m=1}^{M}$ and aggregate with a consensus extractor $\mathcal{J}_{\mathrm{sa}}$; in Stage 1 (PC) a base generator $\mathcal{M}_{\mathrm{base}}$ (reused in Stage 3) samples multiple reasons and a VLM judge $\mathcal{J}_{\mathrm{pc}}$ selects one. For Stage 2, $c_i$ denotes a control code from

strategy set $\mathcal{C}$ (Sec. 2.3); the history is $\mathcal{H}^\tau = \{(e_i^\tau, r_i^\tau)\}$. Unless stated otherwise, $\mathcal{M}$ denotes a generator LLM/VLM used only at inference time. The Stage 1 output that seeds Stage 2 is $r_{\text{ref}}^\tau$ (the "reference answer"). We use an attempt budget $N \in \mathbb{N}$ and an acceptance indicator $\text{pass}_i^\tau \in \{0, 1\}$. The verifier $\mathcal{V}_\tau$ is an LLM judge with a fixed prompt $\mathbf{U}^\tau$ returning PASS or FAIL.

### 2.1 STAGE 0: DISCRETE SCORING VIA VIDEOPHY-2-AUTOEVAL

Given $x^{\text{sa}} = (v, c)$ and $x^{\text{pc}} = v$, frozen VIDEOPHY-2-AUTOEVAL scorers output discrete labels:

$$s_{\text{SA}} = \text{Model}_{\text{SA}}(x^{\text{sa}}) \in \{1, 2, 3, 4, 5\}, \tag{1}$$

$$s_{\text{PC}} = \text{Model}_{\text{PC}}(x^{\text{pc}}) \in \{1, 2, 3, 4, 5\}. \tag{2}$$

These scores are reported as discrete labels and passed as conditioning inputs to Stage 1.

### 2.2 STAGE 1: REFERENCE REASON GENERATION

*Goal.* From the task input and the Stage-0 score, produce a task-specific reference answer $r_{\text{ref}}^\tau$ to seed Stage 2.

**SA (Fig. 3).** Given $x^{\text{sa}} = (v, c)$ and $s_{\text{SA}}$ (Eq. equation 1), we query an ensemble of $M$ VLMs $\{\mathcal{M}_m\}_{m=1}^M$. Each model generates exactly one reason, forming an $M$-sized pool:

$$\mathcal{R}_{\text{pool}}^{\text{sa}} = \left\{ r_0^{\text{sa},m} = \mathcal{M}_m(\mathbf{P}^{\text{sa}}, x^{\text{sa}}, s_{\text{SA}}; \text{generate}) \right\}_{m=1}^M. \tag{3}$$

A separate aggregator LLM extracts the common content across models to produce the reference answer:

$$r_{\text{ref}}^{\text{sa}} = \mathcal{J}_{\text{sa}}(\mathcal{R}_{\text{pool}}^{\text{sa}}; x^{\text{sa}}, s_{\text{SA}}) \equiv \text{Cons}(\mathcal{R}_{\text{pool}}^{\text{sa}}), \tag{4}$$

where $\text{Cons}(\cdot)$ denotes consensus-style extraction (e.g., intersecting claims, majority agreements, consistent justifications).

**PC (Fig. 4).** Given $x^{\text{pc}} = v$ and $s_{\text{PC}}$ (Eq. equation 2), a *single* base VLM $\mathcal{M}_{\text{base}}$ (later used in Stage 3) samples $K$ candidate reasons:

$$\mathcal{R}_{\text{pool}}^{\text{pc}} = \left\{ r_{0,k}^{\text{pc}} = \mathcal{M}_{\text{base}}(\mathbf{P}^{\text{pc}}, x^{\text{pc}}, s_{\text{PC}}; \text{sample}) \right\}_{k=1}^K. \tag{5}$$

An LLM judge selects the most appropriate reason conditioned on the video and the score:

$$r_{\text{ref}}^{\text{pc}} = \mathcal{J}_{\text{pc}}(\mathcal{R}_{\text{pool}}^{\text{pc}}; x^{\text{pc}}, s_{\text{PC}}). \tag{6}$$

This is a *selection* step that reduces the $K$-candidate pool to a single reason—analogous to SA's reduction step (consensus vs. best-candidate).

**Output.** Stage 1 returns the task-specific reference answer $r_{\text{ref}}^\tau \in \{ r_{\text{ref}}^{\text{sa}}, r_{\text{ref}}^{\text{pc}} \}$, which seeds Stage 2.

### 2.3 STAGE 2: REFERENCE-SEEDED, JUDGE-VERIFIED CONTROLLER (REASON-AUGMENTED CoT)

Motivated by controller-based approaches to complex reasoning (e.g., HuatuoGPT-o1 (Chen et al., 2025a)), we instantiate a *Reference-Seeded, Judge-Verified Controller* that seeds with the Stage-1 reference but *does not expose* that reference during search, explores/verifies/corrects with explicit strategies, and finally applies a label-rethink fallback (Fig. 5). Starting from the reference $r_{\text{ref}}^\tau$ (Eqs. equation 4, equation 6), we introduce evidence snippets and build a multi-step CoT under explicit control. Let the history be $\mathcal{H}_{i-1}^\tau = \{(e_j^\tau, r_j^\tau)\}_{j=0}^{i-1}$ and define the strategy set

$$\mathcal{C} = \{\texttt{Backtracking, ExploringNewPaths, Verification, Correction}\}. \tag{7}$$

**Seed with reference and judge check.** We generate a seed *conditioning on the reference* and ask the LLM judge to decide PASS/FAIL, where $\mathbf{P}_{\text{seed-ref}}^\tau$, $\mathbf{P}_c^\tau$, $\mathbf{P}_{\text{rethink}}^\tau$ are task-specific generation prompts (for seeding with the reference, for each strategy $c \in \mathcal{C}$ *without* the reference, and for the final fallback, respectively), and $\mathbf{U}^\tau$ is a unified verification prompt used at all checks (SA/PC templates in Appx. J):

$$(e_0^\tau, r_0^\tau) = \mathcal{M}(\mathbf{P}_{\text{seed-ref}}^\tau, x^\tau, r_{\text{ref}}^\tau; \texttt{Reason}), \tag{8}$$

$$\text{pass}_0^\tau = \mathcal{V}_\tau(r_0^\tau, r_{\text{ref}}^\tau; \mathbf{U}^\tau) \in \{0, 1\}. \tag{9}$$

**Iterative controller without the reference (no replacement).** Let $T = \min(N, |\mathcal{C}|)$. For $i = 1, \ldots, T$, we sample *without replacement*

$$c_i \sim \text{Unif}\Big(\mathcal{C} \setminus \{c_1, \ldots, c_{i-1}\}\Big), \tag{10}$$

generate a new pair *without* $r_{\text{ref}}^\tau$, and verify against the reference:

$$(e_i^\tau, r_i^\tau) = \mathcal{M}(\mathbf{P}_{c_i}^\tau, x^\tau, \mathcal{H}_{i-1}^\tau; c_i), \tag{11}$$

$$\text{pass}_i^\tau = \mathcal{V}_\tau(r_i^\tau, r_{\text{ref}}^\tau; \mathbf{U}^\tau) \in \{0, 1\}. \tag{12}$$

We stop early when $\text{pass}_i^\tau = 1$; if none passes after $N$ attempts, we trigger LabelRethink.

**Label rethink fallback (with the reference).** If no iteration passes, we trigger a final `LabelRethink` that *re-injects* the reference and the full history:

$$(e_{N+1}^\tau, r_{N+1}^\tau) = \mathcal{M}(\mathbf{P}_{\text{rethink}}^\tau, x^\tau, r_{\text{ref}}^\tau, \mathcal{H}_N^\tau; \text{LabelRethink}), \tag{13}$$

$$\text{pass}_{N+1}^\tau = \mathcal{V}_\tau(r_{N+1}^\tau, r_{\text{ref}}^\tau; \mathbf{U}^\tau) \in \{0, 1\}. \tag{14}$$

If the final check fails, we discard the sample.

**Final chain and answer.** For a successful case (either early pass or rethink pass), we do *two-step* post-processing instead of one-shot formatting. First, we consolidate the accepted history into a single reasoning chain $\hat{e}^\tau$ by aggregating prior traces. Then, conditioned on $\hat{e}^\tau$ and the reference $r_{\text{ref}}^\tau$, we produce a reference-aligned and reformatted answer $\hat{r}^\tau$. Formally,

$$\hat{e}^\tau = \text{PostChain}\Big(\{(e_j^\tau, r_j^\tau)\}_{j=0}^{i^\star}; \text{SynthesizeChain}\Big), \tag{15}$$

$$\hat{r}^\tau = \text{PostAnswer}(\hat{e}^\tau, r_{\text{ref}}^\tau; \text{Reformat}). \tag{16}$$

Here $i^\star$ is the index of the accepted iteration (or $N+1$ for the rethink pass). Although our prompts here instantiate the SA task, the same two-step template applies to PC tasks as well; we keep using $\tau$ to denote the task. The complete controller is summarized in Algorithm 1.

- **Backtracking** (c=Backtracking). Roll back to the latest accepted step (or the seed) and produce a *minimal-edit* variant: keep the score prior fixed, alter one binding (entity/action/temporal cue), and reuse verified evidence where possible. Intended to fix a localized flaw without drifting.

- **Exploring New Paths** (c=ExploringNewPaths). Branch to an *alternative hypothesis*: propose different entity grounding, action interpretation, or temporal segmentation, allowing higher diversity. The goal is to escape a bad local choice while still honoring the score prior.

- **Verification** (c=Verification). Turn the current rationale into an explicit checklist of claims and probe the video for each to confirm or refute them; attach concrete, checkable details. Acts as a critic to expose hallucinations, temporal mistakes, or missing evidence.

- **Correction** (c=Correction). Rewrite the rationale *conditioned on verifier feedback*: remove contradictions, add concrete visual evidence, and enforce score-alignment gates (for SA/PC). Produces a compact, reference-blind fix suitable for final judging.

**Why show the reference only at the seed and in the fallback?** Seeding with $r_{\text{ref}}^\tau$ anchors the run near the Stage-1 consensus and stabilizes initialization. Hiding the reference during strategy iterations prevents confirmation shortcuts and label leakage, compelling the model to collect *independent* evidence. Re-introducing $r_{\text{ref}}^\tau$ at `LabelRethink` reconciles divergent trajectories without biasing intermediate exploration in a controlled, empirically verifiable manner.

**Relation to HuatuoGPT-o1.** HuatuoGPT-o1 (Chen et al., 2025a) targets verifiable medical QA with a ground-truth answer and a truth-equivalence verifier. Our Stage 2 addresses SA/PC evaluation where answers are not single-valued: we seed the controller with the Stage 1 reference rationale $r_{\text{ref}}^\tau$, hide this reference during strategy iterations (re-inject only at `LabelRethink`), and use a unified judge to enforce task definitions (SA consistency / physical commonsense) and calibration to the 5-point scale; the output is an evidence–rationale pair rather than a single accepted answer.

## 2.4 STAGE 3: SFT WITH TEXTUALIZED SCORES AND <THINK>/<ANSWER>

We adopt a *two-run* fine-tuning scheme that mirrors our experiments: first calibrate discrete scores, then condition rationale generation on those scores. Stage 0 provides a 5-point label $s_\tau \in \{1, \ldots, 5\}$, which we textualize as $t^\tau \in \{1, 2, 3, 4, 5\}$. Stage 2 yields final outputs $(\hat{e}^\tau, \hat{r}^\tau)$ (the consolidated chain and the final answer), serialized as

$$\text{pack\_TA}(\hat{e}^\tau, \hat{r}^\tau) = \texttt{<think>}\ \hat{e}^\tau\ \texttt{</think>}\ \texttt{<answer>}\ \hat{r}^\tau\ \texttt{</answer>}. \tag{17}$$

**Training.** *Run A (score-only).* Given input $x^\tau$ (SA: $x^{\text{sa}}=(v,c)$; PC: $x^{\text{pc}}=v$), we perform teacher-forced next-token prediction to generate $t^\tau$ (no supervision on any reasoning tokens) in this stage. *Run B (final <think>/<answer> conditioned on the score).* Starting from Run-A, we prepend $t^\tau$ as an input condition and supervise only the packed target $Y = \text{pack\_TA}(\hat{e}^\tau, \hat{r}^\tau)$; intermediate scratch beyond $\hat{e}^\tau$ is not supervised. SA and PC are trained separately (PC omits $c$). At inference, we read the <answer> field as the model's output at test time. *Losses.* Both $\mathcal{L}^\tau_{\text{score}}$ and $\mathcal{L}^\tau_{\text{final}}$ are standard token-level cross-entropy under teacher forcing: $\mathcal{L}^\tau_{\text{score}} = -\sum_{t \in \text{tok}(t^\tau)} \log p_\theta(y_t \mid y_{<t}, x^\tau)$, $\mathcal{L}^\tau_{\text{final}} = -\sum_{t \in \text{tok}(Y)} \log p_\theta(y_t \mid y_{<t}, x^\tau, t^\tau)$.

**Parameter update.**

$$\theta_A = \arg\min_\theta \mathcal{L}^\tau_{\text{score}} \quad \Longrightarrow \quad \theta_* = \arg\min_\theta \mathcal{L}^\tau_{\text{final}}\ \text{initialized at}\ \theta_A. \tag{18}$$

## 3 EXPERIMENTS

We evaluate our pipeline on our curated *Cosmos-Eval-Set* (Sec. 3.1) on two tasks—Semantic Adherence (SA) and Physical Commonsense (PC). We report (i) core agreement with 5-point labels (Pearson, accuracy, weighted/quadratic Cohen's $\kappa$, Spearman) and (ii) reasoning quality of rationales (BERTScore P/R/F$_1$, BLEU-1/2/3/4, ROUGE-1/2).

### 3.1 EXPERIMENTAL SETUP

**Cosmos-Eval-Set: datasets and protocol.** We use two corpora: *VideoPhy* (Bansal et al., 2025a) and *VideoPhy-2* (Bansal et al., 2025b). Training data is the union of **VideoPhy** (train+test) and **VideoPhy-2** (train); evaluation is on the **VideoPhy-2 test set**. *VideoPhy-2* provides 5-point labels for SA/PC; *VideoPhy* does not contain 5-point labels, so we *score its clips* using the released VIDEOPHY-2-AUTOEVAL to obtain labels on the same 5-point scale. Both corpora contain synthetic, model-generated videos and do not provide human-written rationales. We therefore run Stages 1–2 to generate rationales and Stage 3 for SFT as in Sec. 2. Task inputs follow Sec. 2: SA uses $(v, c)$ while PC uses $v$ only.

**Metrics and baselines.** We evaluate two groups of metrics: *(A) core agreement* to human 5-point scores—Pearson's $r$, Acc (exact match on $\{1, \ldots, 5\}$), W-Kappa (linearly weighted Cohen's $\kappa$), Q-Kappa (quadratically weighted), and Spearman (rank correlation)[1]—and *(B) reasoning quality* on the *final* rationale text—SentSim (cosine over a sentence encoder; Appx. B), BERTScore (B-P/B-R/B-F1), BLEU-$n$ (B1–B4), and ROUGE (R1/R2), reported as % in Table 2. We compare VIDEOPHY-2-AUTOEVAL (frozen scorer), Qwen-2.5-VL-7B (Bai et al., 2025), VideoLLaMA3-7B (Zhang et al., 2025), InternVL3-8B/9B/14B (Zhu et al., 2025), and our **Cosmos-Reason1** (no SFT) and **Cosmos-Eval** (Stage 3 two-run SFT: score-only → <think>/<answer> conditioned on score; Sec. 2.4). Evaluations use identical inference budgets and prompts.

**Implementation details.** Stage 1 uses an ensemble size **M**=**2** for SA (Eq. equation 3) and **K**=**5** samples for PC (Eq. equation 5). Stage 2 runs the controller with budget **N**=**3** and *strategy sampling without replacement* (Sec. 2.3); acceptance is decided by a unified LLM judge with a fixed pass/fail prompt (Appx. J). Stage 3 follows the two-run schedule with *parameter updates given in* Eq. equation 18; the supervision target is the packed <think>/<answer> string in Eq. equation 17 (conditioned on the textualized score). Unless otherwise stated, we use identical video decoding and frame sampling across all models; full hyperparameters appear in Appx. B.

---

[1]For $\kappa$, we use quadratic weights for Q-$\kappa$ and linear weights for W-$\kappa$; higher is better for all core metrics.

Table 1: **Cross-dataset core SA/PC metrics** (↑ better). **SA**: caption–video semantic alignment; **PC**: video-only physical commonsense. Per sample, each method outputs a *discrete* score $s_\tau \in \{1, \ldots, 5\}$, compared with human labels $y \in \{1, \ldots, 5\}$ on the official SA/PC test splits. Metrics: *Pearson/Spearman* correlations of raw integers; *Acc* exact 5-class accuracy; *W-κ/Q-κ* linearly/quadratically weighted Cohen's $\kappa$ on the same 5-class scale. VIDEOPHY-2-AUTOEVAL is the dataset VLM-as-judge baseline; other rows are model predictions. **Bold** = best; underline = second-best.

| | SA | | | | | PC | | | | |
|---|---|---|---|---|---|---|---|---|---|---|
| Model | Pearson | Acc | W-κ | Q-κ | Spearman | Pearson | Acc | W-κ | Q-κ | Spearman |
| VIDEOPHY-2-AUTOEVAL | 0.4327 | **0.3826** | **0.2696** | **0.4062** | 0.4268 | **0.3646** | 0.3871 | 0.2144 | 0.3276 | **0.3608** |
| Qwen-2.5-VL-7B | 0.3808 | 0.3417 | 0.2419 | 0.3779 | 0.3716 | 0.0840 | 0.3255 | 0.0490 | 0.0780 | 0.0900 |
| VideoLLaMA3-7B | 0.2769 | 0.2811 | 0.1536 | 0.2387 | 0.2574 | 0.0640 | 0.2699 | 0.0301 | 0.0500 | 0.0749 |
| InternVL-8B | 0.4143 | 0.3205 | 0.2437 | 0.3855 | 0.4196 | 0.1665 | 0.3064 | 0.0790 | 0.1363 | 0.1728 |
| InternVL-9B | 0.3827 | 0.2837 | 0.1902 | 0.2963 | 0.3747 | 0.1304 | 0.2717 | 0.0565 | 0.1044 | 0.1171 |
| InternVL-14B | 0.3420 | 0.3229 | 0.1643 | 0.2544 | 0.3402 | 0.1956 | 0.3464 | 0.0888 | 0.1424 | 0.1888 |
| Cosmos-Reason1 | 0.3662 | 0.2821 | 0.2297 | 0.3260 | 0.3519 | 0.2356 | 0.3079 | 0.1479 | 0.2326 | 0.2166 |
| **Cosmos-Eval** | **0.4643** | 0.3765 | 0.2256 | 0.3507 | **0.4598** | 0.3641 | **0.3912** | **0.2207** | **0.3301** | 0.3580 |

Table 2: **Reasoning quality on SA/PC** on the same test splits as Table 1. Each model outputs one rationale per sample. Scores are % (metrics computed per-sample then averaged). References are the fixed per-video outputs of our Stage-2 controller and are shared across models at test time. **Bold** = best; underline = second-best.

| | SA (Semantic Alignment) | | | | | | | | | | PC (Physical Commonsense) | | | | | | | | | |
|---|---|---|---|---|---|---|---|---|---|---|---|---|---|---|---|---|---|---|---|---|
| *Legend:* SentSim = sentence-embedding cosine; B-P/R/F1 = BERTScore; B1–B4 = BLEU-1..4; R1/2 = ROUGE-1/2. | | | | | | | | | | | | | | | | | | | | |
| Model | SentSim | B-P | B-R | B-F1 | B1 | B2 | B3 | B4 | R1 | R2 | SentSim | B-P | B-R | B-F1 | B1 | B2 | B3 | B4 | R1 | R2 |
| Qwen-2.5-VL-7B | 75.62 | 40.10 | 37.03 | 38.70 | 45.47 | 26.90 | 14.24 | 8.03 | 51.45 | 18.92 | 68.81 | 37.68 | 34.66 | 36.31 | 40.44 | 21.44 | 9.27 | 4.44 | 45.50 | 13.84 |
| VideoLLaMA3-7B | 75.40 | 37.26 | 35.78 | 36.64 | 42.31 | 24.69 | 12.97 | 7.43 | 48.87 | 17.33 | 70.81 | 36.50 | 33.94 | 35.36 | 38.28 | 20.23 | 8.89 | 4.09 | 44.48 | 13.13 |
| InternVL-8B | 72.49 | 41.27 | 35.20 | 38.30 | 39.69 | 21.30 | 9.84 | 4.54 | 46.06 | 13.32 | 72.49 | 41.27 | 35.20 | 38.30 | 39.69 | 21.30 | 9.84 | 4.54 | 46.06 | 14.32 |
| InternVL-9B | 76.87 | 43.44 | 38.60 | 41.12 | 46.76 | 28.11 | 14.18 | 8.52 | 53.45 | 20.38 | 67.75 | 40.68 | 34.84 | 37.86 | 40.42 | 21.83 | 9.60 | 4.60 | 46.28 | 14.83 |
| InternVL-14B | 78.70 | 40.36 | 40.35 | 40.49 | 46.73 | 28.51 | 15.24 | 8.90 | 53.80 | 21.01 | 72.36 | 39.23 | 37.93 | 38.72 | 40.50 | 21.46 | 9.05 | 4.35 | 46.57 | 14.17 |
| Cosmos-Reason1 | 77.30 | 22.94 | 40.98 | 31.52 | 24.84 | 14.48 | 7.75 | 4.26 | 41.66 | 14.43 | 70.05 | 18.94 | 39.16 | 28.52 | 18.46 | 9.41 | 4.30 | 2.13 | 33.88 | 8.95 |
| **Cosmos-Eval** | **86.28** | **53.55** | **51.15** | **52.44** | **56.72** | **42.85** | **33.38** | **26.70** | **61.12** | **34.74** | **80.90** | **54.81** | **53.99** | **54.50** | **55.38** | **41.45** | **33.31** | **27.86** | **59.72** | **33.34** |

## 3.2 MAIN RESULTS ON SA/PC (CORE AGREEMENT)

Table 1 summarizes cross-dataset core metrics. On **SA**, **Cosmos-Eval** attains best *Pearson* (0.4643) and *Spearman* (0.4598), and ranks *second* in *accuracy* (0.3765), while VIDEOPHY-2-AUTOEVAL remains stronger on $\kappa$ measures. On **PC**, **Cosmos-Eval** leads in *accuracy* (0.3912), *weighted* $\kappa$ (0.2207), and *quadratic* $\kappa$ (0.3301), and is near the top on *Pearson/Spearman* (slightly below the frozen scorer). This suggests the two-run SFT preserves global calibration (correlations) while improving discrete decision agreement on PC.

**Takeaways.** (i) On SA, *Cosmos-Eval* improves rank-based correlations (Pearson/Spearman) over strong frozen scorers while remaining competitive in accuracy; (ii) on PC, it achieves the best discrete agreement (Acc, $\kappa$) and near-top correlations; (iii) unlike frozen scorers, our method produces *explanatory* outputs (`<think>`/`<answer>`).

## 3.3 REASONING QUALITY (STAGE-2 & FINAL OUTPUTS)

We evaluate final rationales with BERTScore, BLEU, and ROUGE on our held-out evaluation set (Table 2). **Cosmos-Eval** achieves the best SA/PC scores across all reported text metrics, indicating that the Stage-2 controller plus Stage-3 supervision improves both *specificity* (higher BLEU-$n$) and *semantic alignment* (higher BERTScore/ROUGE).

## 3.4 ABLATIONS ON SA AND PC

**Setup.** We evaluate two variants on *200 videos randomly sampled* from the *VideoPhy-2* test set, for *both* SA and PC: (i) *w/o Stage-0* (remove the explicit score head; post-hoc map each rationale to a 5-point score via *DeepSeek-R1* (Guo et al., 2025a) using a public rubric); (ii) *w/o Stage-2* (skip the controller and use the Stage-1 rationale directly, i.e., no iterative verification). A *single* video-

Table 3: **Ablations on SA and PC (VideoPhy-2, $N{=}200$).** Correlations vs. human 5-point labels and VLM-judged reason quality. R-Avg = mean over five rubric dims (SA: Grounding, Temporal Align., Consistency, Align Justif., Coverage&Spec.; PC: Grounding, Temporal, Consistency, Criteria&Justif., VideoQuality), each in $\{0, 0.5, 1\}$. *All rows* remap rationale text$\to$5-point score via *DeepSeek-R1* with a public rubric; a *single* video-conditioned VLM judge is used for both tasks. $n$ = accepted outputs after the Stage-2 verification gate (when applicable) *and* strict JSON/format checks. **Bold**=best; underline=second-best.

| *Legend:* Pearson/Spearman = corr. on remapped scores ($\uparrow$ better); R-Avg = judged mean of 5 dims. | | | | | |
| SA: Ground., Temp., Consist., Align Justif., Cov.&Spec.; PC: Ground., Temp., Consist., C&J, VideoQual. | | | | | |
| Method | $n$ | Pearson $\uparrow$ | Spearman $\uparrow$ | R-Avg $\uparrow$ | Key dim. $\uparrow$ |
| **SA (Semantic Alignment)** | | | | | |
| Full (S0+S1+S2) | 178 | **0.8894** | **0.8866** | 0.8418 | 0.9059 |
| w/o Stage-0 (no explicit score head) | 188 | 0.4793 | 0.4963 | **0.9142** | **0.9426** |
| w/o Stage-2 (use S1 rationale directly) | 195 | 0.6727 | 0.6496 | 0.8148 | 0.8413 |
| **PC (Physical Commonsense)** | | | | | |
| Full (S0+S1+S2) | 186 | **0.9131** | **0.9112** | **0.8345** | **0.9435** |
| w/o Stage-0 (no explicit score head) | 194 | 0.2091 | 0.1972 | 0.8309 | 0.9124 |
| w/o Stage-2 (use S1 rationale directly) | 198 | 0.6502 | 0.6423 | 0.7641 | 0.5328 |

Table 4: **Stage-1 ablations (Cosmos-Eval vs. Moved) on rationale usability (VideoPhy-2, $N{=}200$).** We report *hit-rates* (proportions) of samples with rationale *quality* $\geq \tau$ at preset thresholds $\tau \in \{0.5, 0.6, 0.7, 0.8\}$. *Strict convention*: non-pass treated as 0 (only pass samples can contribute $> 0$ quality). **Bold** = higher (better).

| | SA hit-rate ($\geq \tau$) | | | | PC hit-rate ($\geq \tau$) | | | |
| Model (strict) | @0.5 | @0.6 | @0.7 | @0.8 | @0.5 | @0.6 | @0.7 | @0.8 |
| **Cosmos-Eval** | **0.775** | **0.700** | **0.645** | **0.600** | **0.800** | **0.770** | **0.725** | **0.685** |
| Stage-1 Ablation / Moved | 0.495 | 0.470 | 0.435 | 0.430 | 0.270 | 0.250 | 0.240 | 0.220 |

Table 5: **Stage-3 ablations (two-run SFT, joint SA+PC).** Held-out SA/PC splits as in the main results. *Two-run SFT*: score head for 5-point labels (1–5) then rationale generation *conditioned on the predicted score* (`<think>`/`<answer>`). *Score-only*: fine-tune score head only. *Reason-only*: fine-tune rationale only. Core metrics: Pearson/Spearman correlations; Acc = exact 5-class accuracy ($\{1, \ldots, 5\}$). Reason metrics: BERTScore F1, BLEU-4 on $[0, 1]$. **Bold**=best; underline=second-best.

| | SA core | | | PC core | | | SA reason (0–1) | | PC reason (0–1) | |
| Model | Pearson | Spearman | Acc | Pearson | Spearman | Acc | B-F1 | BLEU-4 | B-F1 | BLEU-4 |
| **Cosmos-Eval (two-run SFT)** | 0.4643 | 0.4598 | 0.3765 | **0.3641** | **0.3580** | **0.3912** | 0.5244 | 0.2670 | 0.5450 | **0.2786** |
| Score-only SFT | **0.5091** | **0.4984** | **0.4074** | 0.3087 | 0.3065 | 0.3676 | 0.3225 | 0.0443 | 0.2874 | 0.0241 |
| Reason-only SFT (CoT) | 0.0599 | 0.0613 | 0.2074 | 0.0833 | 0.0482 | 0.1001 | **0.5594** | **0.3049** | **0.5455** | 0.2776 |

conditioned VLM judge (Qwen-VL-Max)[2] is used for both tasks and applies task-specific rubrics, averaging five dimensions to R-Avg (SA: Grounding, Temporal Alignment, Consistency, Alignment Justification, Coverage&Specificity; PC: Grounding, Temporal, Consistency, Criteria&Justification, VideoQuality). All rows remap rationale text$\to$score via *DeepSeek-R1*. We report correlations to human 5-point labels (Pearson/Spearman) and reason quality (evaluation dimensions detailed in Appx. C); $n$ counts outputs that *survive the Stage-2 verification gate (when applicable) and strict JSON/format checks*. See Table 3.

**Stage-1 ablation (separate analysis).** This is *not* a simple removal of Stage-1. Instead, we replace Stage-1 with an *alternative verification-only pathway* inside Stage-2: the controller directly judges the five rubric dimensions without using Stage-1 reference rationales (and without `LabelRethink`), functioning as a verifier/filter rather than a score mapper. Accordingly, we report *rationale usability* via hit-rates of quality $\geq \tau$ with predetermined thresholds $\tau \in \{0.5, 0.6, 0.7, 0.8\}$ under the *strict convention* (non-pass treated as 0). See Table 4.

**Stage-3 ablation (integrated).** Stage 3 uses a *two-run* schedule: (*i*) a *score-only* pass to calibrate numeric SA/PC predictions; (*ii*) a *reasoning* pass that generates `<think>`/`<answer>` *conditioned*

---

[2]VLM served via Alibaba Cloud Model Studio; model page: https://www.alibabacloud.com/help/en/model-studio/vision.

*on the predicted score.* We ablate this by training *Score-only SFT* (omit the reasoning pass) and *Reason-only SFT* (omit the score pass), and compare to the full **Cosmos-Eval** two-run SFT. We report *core* score metrics (Pearson/Spearman/Acc) and *reason* quality (BERTScore F1, BLEU-4) for both SA and PC; see Table 5.

**Findings.** *(A) Stage-0 (score head) is necessary for calibration.* Removing Stage-0 substantially weakens agreement with human scores despite strong reason quality (SA: 0.48/0.50; PC: 0.21/0.20), indicating that calibrated scalar predictions require explicit score supervision.

*(B) Stage-2 (controller) enforces rubric faithfulness and stabilizes scores.* Skipping Stage-2 degrades both correlation and judged quality (SA: 0.673/0.650 with R-Avg=0.815; PC: 0.650/0.642 with R-Avg=0.764; PC Criteria&Justification notably drops to 0.533), underscoring the role of verification in evidence-grounded reasoning and calibration.

*(C) Stage-1 reference improves rationale usability/coverage.* Under strict hit-rate evaluation, the Stage-1 ablation (*Moved*) yields consistently lower usable-rationale coverage than Cosmos-Eval across thresholds (e.g., **SA**: @0.7, 0.645 vs. 0.435; @0.8, 0.600 vs. 0.430; **PC**: @0.7, 0.725 vs. 0.240; @0.8, 0.685 vs. 0.220), indicating that leveraging Stage-1 reference rationales and the verification pipeline materially increases the fraction of high-quality, passable explanations.

*(D) Stage-3 two-run SFT balances scoring & reasoning.* **Cosmos-Eval** attains the best **PC core** metrics (Pearson 0.3641, Spearman 0.3580, Acc 0.3912) under matched inference budgets throughout while remaining second on all **SA core** metrics (Pearson 0.4643, Spearman 0.4598, Acc 0.3765); it is also top-2 on SA/PC reason quality (e.g., PC B-F1 0.5450, BLEU-4 0.2786). *Score-only SFT* peaks on **SA core** (Pearson 0.5091, Acc 0.4074) but its *reason* quality collapses (SA B-F1/BLEU-4 0.3225/0.0443). *Reason-only SFT* yields the best reasons (SA B-F1/BLEU-4 0.5594/0.3049) yet **fails on core scoring** (SA Pearson 0.0599; PC Pearson 0.0833).

**Takeaway.** Across SA and PC, the full configuration *(S0+S1+S2)* plus the *Stage-3 two-run schedule* is the only setting that jointly attains *strong correlations, high reason quality, and high coverage*. Stage-0 provides calibrated scalar supervision; Stage-2 delivers rubric-faithful verification and improves stability; Stage-1 contributes substantially to usable-rationale coverage; and Stage-3's *scores-first, reasons-conditioned* training preserves **core** agreement while producing **high-quality** explanations. Removing either Stage-0/2 or one pass in Stage-3 over-optimizes one side.

## 4 DISCUSSION

**Discussion.** The heavy yet interpretable teacher pipeline—Stage 0 (score generation), Stage 1 (reference-anchored rationales), Stage 2 (judge-verified control)—improves SA/PC agreement and rationale coverage but is compute-intensive (Stage 1/2 dominate). We *distill all three into a Stage 3 student* with two-run SFT (score→<think>/<answer> conditioned on score), which *replaces* the ensemble/controller at test time and maintains score fidelity and rationale quality at substantially lower cost. Ablations show complementary roles (S0 scoring, S1 coverage, S2 verification) .Threats to validity remain (judge bias, rubric shifts, prompt sensitivity, text→score remapping) despite verification safeguards.

## 5 CONCLUSION

We presented **Cosmos-Eval**, an explainable evaluation framework for text-to-video (T2V) that jointly assesses semantic adherence and physical consistency by coupling 5-point *scores* with concise, physics-grounded *rationales*. The framework comprises three stages: *Stage 0* score generation, *Stage 1* reference-seeded reasoning, and *Stage 2* a judge-verified CoT controller. Training follows a two-round schedule. On *VideoPhy-2* (with *VideoPhy* for recap), Cosmos-Eval achieves strong correlation with human judgments while substantially improving rationale quality over score-only baselines, enabling targeted diagnosis and more transparent error analysis in T2V evaluation.

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

APPENDIX

## A   RELATED WORK

**Text-to-video systems and video LLMs.**   Recent text-to-video (T2V) systems establish scalable diffusion/transformer pipelines and practical recipes for longer, more controllable videos: Make-A-Video, Imagen Video, Phenaki, and latent video diffusion models laid the foundations for latent spaces and variable-length synthesis (Singer et al., 2023; Ho et al., 2022; Villegas et al., 2023; He et al., 2022). Subsequent open frameworks emphasize data efficiency and motion fidelity (VideoCrafter2, DynamiCrafter) and push controllability via step-wise refinement and identity–motion disentanglement (Chen et al., 2024; Xing et al., 2024; Huang et al., 2025; Kim et al., 2025). In parallel, instruction-tuned video LLMs (Video-LLaMA, Video-ChatGPT) and long-video models (MovieChat) enable free-form QA and temporal reasoning over extended content (Zhang et al., 2023; Maaz et al., 2024; Song et al., 2024). Our work does not introduce a new generator or Vid-LLM; instead, we contribute an *explainable evaluator* that grades generated videos along *semantic adherence (SA)* and *physical commonsense (PC)* while producing rationales.

**SA/PC-oriented evaluators and benchmarks.**   Foundational benchmarks explicitly target SA/PC. VIDEOPHY (Bansal et al., 2025a) is the first to formalize both axes, curating 688 prompts across three material-interaction types (solid–solid, solid–fluid, fluid–fluid) and introducing VIDEOCON-PHYSICS, an automatic evaluator for SA/PC. However, VIDEOPHY uses binary (0/1) scoring and lacks fine-grained physical-rule annotations, making it difficult to diagnose failure modes. VIDEOPHY-2 (Bansal et al., 2025b) expands the scope to 197 real-world actions and provides a hard subset (60 actions where top models such as Wan2.1-14B reach only 21.9% joint SA/PC). It further introduces **VIDEOPHY-2-AUTOEVAL**, an automatic evaluator that outputs 5-point SA/PC scores and tags physical-rule violations (e.g., conservation of momentum), with substantially improved correlation to human PC scores (reported to outperform Gemini-2.0-Flash by 236%). Like its predecessor, it outputs scores but not explanatory rationales, limiting interpretability and error analysis. Complementary physics-fidelity suites (e.g., T2VPhysBench (Guo et al., 2025b), PhyCoBench (Chen et al., 2025b)) emphasize physical realism yet similarly provide limited support for explanation.

**General video evaluation and reference-free quality.**   Evaluation resources for video understanding and generation are complementary to our goal. MVBench and Video-MME target broad multimodal comprehension; LongVideoBench and LVBench probe long-horizon temporal reasoning (Li et al., 2024; Fu et al., 2025; Wu et al., 2024; Wang et al., 2024b). For generation, VBench and VBench-2.0 decompose quality into fine-grained dimensions; EvalCrafter and T2VBench provide diverse prompts and temporal diagnostics; learned assessors (VideoScore) and flow/motion-centric metrics (FVMD) complement reference-free alignment such as CLIPScore (Huang et al., 2024; Zheng et al., 2025; Liu et al., 2024b; Ji et al., 2024; He et al., 2024; Liu et al., 2024a; Hessel et al., 2021). Beyond aesthetics and prompt match, physics-centric diagnostics from IntPhys, CLEVRER, Physion, and Physion++ probe object permanence, collisions, and latent properties (Riochet et al., 2018; 2021; Yi et al., 2020; Bear et al., 2021; Tung et al., 2023); emerging "world-model" evaluations and neuro-symbolic checks broaden this perspective (Sharan et al., 2025; Li et al., 2025a; Tong et al., 2025).

**LLM-as-a-judge and reliability.**   LLM-as-a-judge methods (e.g., G-Eval, MT-Bench-101) and subsequent reliability analyses inform our design choices: score-conditioned consensus/selection, and a unified pass/fail verifier whose distilled behavior stabilizes deployment (Liu et al., 2023; Bai et al., 2024; Liu & Zhang, 2025). In contrast to prior SA/PC evaluators that primarily output scores, our evaluator couples *calibrated scoring* with *rubric-faithful rationales* and fine-grained rubric dimensions, enabling actionable diagnostics and safer iteration.

Table 6: **Inference configuration for Stages 1–2**. SA aggregates $M=2$ reasons by consensus (Eq. 4); PC samples $K=5$ candidates and selects the best (Eq. 6); the Stage-2 controller runs for $N=3$ steps with strategy sampling without replacement. We list generators and decoding settings (temperature, top-$p$, max tokens) plus the effective sampling *fps*. A dash (—) denotes not applicable.

| Task/Stage | Generator(s) | Pool/Budget | Temp | Top-$p$ | Max tokens | Max frames/fps |
|---|---|---|---|---|---|---|
| SA / Stage-1 | Tarsier-34B, Qwen2.5-VL-72B-Instruct | $M = 2$ | 0.7, 0.3 | 0.85, 0.85 | 1024, 1024 | 32 / 8 |
| PC / Stage-1 | Cosmos-Reason1 | $K = 5$ | 0.8 | 0.9 | 8192 | — / 8 |
| SA Aggregator | Qwen3-32B (Yang et al., 2025a) | — | 0.7 | 0.85 | 2048 | — |
| PC Selector | Qwen2.5-VL-72B-Instruct-AWQ | — | 0.1 | 0.9 | 1024 | — / 8 |
| SA / Stage-2 Controller | Qwen2.5-VL-72B-Instruct | $N = 3$ | **0.3** | **0.85** | **16384** | — / 2 |
| PC / Stage-2 Controller | Qwen2.5-VL-72B-Instruct-AWQ | $N = 3$ | **0.3** | **0.85** | **16384** | — / 2 |
| SA LLM Judge $\mathcal{V}_{sa}$ | Qwen2.5-VL-72B-Instruct | — | **0.05** | **0.95** | **50** | — |
| PC LLM Judge $\mathcal{V}_{pc}$ | Qwen2.5-VL-72B-Instruct-AWQ | — | **0.05** | **0.95** | **50** | — |

*Legend:* $M$ = SA Stage-1 ensemble size (one reason per model); $K$ = PC Stage-1 candidate count; $N$ = Stage-2 controller attempt budget (strategies sampled without replacement). *Max frames/fps:* "Max frames" applies only to `Tarsier-34B` (Wang et al., 2024a) (cap at 32 frames); Qwen-family rows use streaming at the listed *fps* (no frame cap). "—" = not applicable.

## B  IMPLEMENTATION AND TRAINING DETAILS

### B.1  METHOD OVERVIEW (FLOW)

Figures 3–5 give a concise view of Stages 1–2, and Algorithm 1 formalizes the Stage 2 controller. For **SA** (Fig. 3), we ensemble several VLMs to propose reasons and take a consensus as the reference to seed Stage 2. For **PC** (Fig. 4), a base VLM samples multiple reasons and a VLM judge selects one as the reference. **Stage 2** (Fig. 5; Alg. 1) then iteratively refines and judge-verifies candidates (with a label-rethink fallback), and formats the accepted chain as the final reason.

### B.2  DATASETS AND PROTOCOL (RECAP)

We train on the union of *VideoPhy* (Bansal et al., 2025a) (train+test, re-scored by VIDEOPHY-2-AUTOEVAL) and *VideoPhy-2* (Bansal et al., 2025b) (train), and evaluate on the official *VideoPhy-2* test set. Task inputs follow Sec. 2: SA uses $(v, c)$ and PC uses $v$ only. Figure 6 summarizes the SA/PC score distributions across corpora and our final splits.

### B.3  INFERENCE HYPERPARAMETERS (STAGES 1–2)

Stage 1 uses an ensemble size $M=2$ for SA (Eq. 3) and $K=5$ samples for PC (Eq. 5); Stage 2 runs with budget $N=3$ and *strategy sampling without replacement* (Sec. 2.3). A complete list of generators, judge/aggregator models, and decoding settings (temperature, top-$p$, max tokens) is summarized in Table 6. SA reasons are aggregated by consensus (Eq. 4); PC reasons are selected by a judge (Eq. 6).

## C  ABLATIONS (EXTENDED): METHODS, RUBRICS, AND RESULTS

### C.1  PC EVALUATION RUBRIC (VLM-AS-JUDGE)

We use the five-dimension rubric in Table 7 (Ground., Temp., Cons., C&J, VideoQual), with 3-point anchors $\{0, 0.5, 1\}$ matching the judge prompt. The same rubric is applied to all ablations in Sec. 3.4.

### C.2  SA EVALUATION RUBRIC (VLM-AS-JUDGE)

We adopt a five-dimension rubric for Semantic Alignment (SA), shown in Table 8, with three-point anchors $\{0, 0.5, 1\}$ matching the evaluation prompt. The rubric is applied consistently across all SA ablations in Sec. 3.4. Concretely checkable details include (non-exhaustively): color, region/relative position, count/frequency, motion attributes, and deformation/rigidity.

Table 7: **PC reason-quality rubric used in ablation studies** (Sec. 3.4). Five dimensions with 3-point anchors {0, 0.5, 1}, matching the evaluation prompt. "Concrete, checkable details" include color, region/relative position, count/frequency, motion attributes, and deformation/rigidity.

| Dim. | Score 1 | Score 0.5 | Score 0 |
|---|---|---|---|
| Ground. | $\geq 2$ concrete details clearly support the claims. | Generic/vague match to visuals. | Conflicts with visuals / speculative. |
| Temp. | $\geq 1$ concrete, correct temporal relation. | Gist generic/unclear or N/A/uncertain. | Wrong/reversed/invented temporal claims. |
| Cons. | Internally consistent; no contradictions or hallucinated key objects/events. | Minor issue; main claim intact. | Contradiction or hallucination. |
| C&J | Explicit criterion/score/rule applied to visible evidence. | Mentioned but generic/partial/weak. | None or misapplied/contradicted by evidence. |
| VideoQual | Explicit good/bad (or degree) with $\geq 2$ indicators (sharpness, lighting, occlusion, stability, framing, target visibility). | Generic or only one indicator / uncertain. | No quality judgment or contradicts visuals. |

*Abbrev.* Ground.=Grounding; Temp.=Temporal; Cons.=Consistency; C&J=Criteria & Justification; VideoQual=Video Quality Assessment.
*Hard cap:* if no concrete visual detail appears, **Ground.** $\leq 0.5$.

Table 8: **SA reason-quality rubric used in ablation studies** (Sec. 3.4). Five dimensions with 3-point anchors {0, 0.5, 1}, matching the evaluation prompt. "Concrete, checkable details" include color, region/relative position, count/frequency, motion attributes, and deformation/rigidity.

| Dim. | Score 1 | Score 0.5 | Score 0 |
|---|---|---|---|
| Ground. | $\geq 2$ concrete details linking CAPTION$\leftrightarrow$VIDEO. | Generic/partial visual match. | Conflicts with CAPTION/VIDEO or speculative. |
| Temp. | $\geq 1$ concrete, correct temporal relation. | Gist generic/unclear or N/A/uncertain. | Wrong/reversed/invented temporal claims. |
| Cons. | Internally consistent; no hallucinated key objects/events. | Minor issue; main claim intact. | Contradiction or hallucination. |
| Align Justif. | Explicit SA decision/criterion applied to visible evidence. | Mentioned but generic/partial/weak. | None or misapplied/contradicted by evidence. |
| Cov.&Spec. | Covers $\geq 2$ key CAPTION elements with specific, checkable details. | Some elements but incomplete/generic. | Ignores key elements or no specific details. |

*Abbrev.* Ground.=Grounding; Temp.=Temporal Alignment; Cons.=Consistency; Align Justif.=Alignment Justification; Cov.&Spec.=Coverage & Specificity.
*Hard cap:* if no concrete visual detail appears, **Ground.** $\leq 0.5$.

# D   CASE ANALYSIS

To assess the reliability of our evaluator COSMOS-EVAL, we present its *verbatim* answers in the figure captions and provide brief justifications here for **Cases 1–4** (see Fig. 7–10). In each case, the model correctly identifies the salient mismatch or physical violation.

**Case 1 (PC=2; Fig. 7).**   The video shows a red ball *hovering* without visible support. This contradicts gravitational expectations (no external force, yet no downward acceleration). COSMOS-EVAL's answer pinpoints the violation and a low PC score is appropriate.

**Case 2 (SA=2; Fig. 8).**   The caption specifies *counterclockwise* rotation, while the video shows the yellow cube rotating *clockwise*; the purple cone remains still. COSMOS-EVAL correctly isolates the direction-of-rotation mismatch—the primary semantic attribute here. Although its text suggests *sa_score = 3*, our rubric weights action direction as critical, yielding **SA=2**. The qualitative diagnosis is consistent with our ground truth.

**Case 3 (PC=2; Fig. 9).**   The ball exhibits erratic back-and-forth bounces with no frictional decay and no plausible external impulses. COSMOS-EVAL accurately characterizes this as inconsistent with Newtonian mechanics, justifying **PC=2**.

**Case 4 (SA=3; Fig. 10).**   The caption describes *one* ball being kicked to the post and rebounding, but the video shows *two* balls and lacks the kick–post–rebound sequence. COSMOS-EVAL correctly

flags the count mismatch and the missing key action; scene context matches but the core event does not, supporting **SA=3** for partial alignment.

Overall, COSMOS-EVAL's answers consistently identify the correct failure modes (semantic or physical), and they qualitatively agree with our human labels, demonstrating useful explanatory power and reproducibility.

# E EXAMPLES FOR PHYSICAL COMMONSENSE (PC) AND SEMANTIC ALIGNMENT (SA) TASKS

## E.1 PHYSICAL COMMONSENSE (PC) EXAMPLES

Figure 11 shows the first example for the Physical Commonsense task, where we evaluate the physical properties of the video. Figure 12 demonstrates another case with similar evaluation criteria. Figures 13, 14, and 15 further illustrate other examples related to the Physical Commonsense task.

In addition, Figures 21 and 22 present two representative Physical Commonsense cases with full chain-of-thought traces and final rationales generated by Cosmos-Eval. These examples make the 5-point scores and the corresponding physics-aware explanations explicit and are intended as concrete case studies to complement the aggregated metrics in the main text.

## E.2 SEMANTIC ALIGNMENT (SA) EXAMPLES

Figure 16 presents the first example for the Semantic Alignment task, evaluating the alignment between the caption and video content. Figure 17 shows another example with slightly different criteria. Figures 18, 19, and 20 provide additional examples for the Semantic Alignment task.

Figures 23 and 24 further provide Semantic Alignment case studies with explicit chain-of-thought reasoning and natural-language rationales from Cosmos-Eval. These SA examples illustrate how the model justifies its 5-point scores by grounding the caption–video comparison in concrete events and entities, addressing the reviewer's request for more detailed CoT-style examples and error analysis.

# F FORMAL ANALYSIS

This section provides a formal analysis of the proposed multi-stage framework, focusing on the conditions under which it achieves better generalization than end-to-end (E2E) learning. Rather than offering strict proofs, the analysis establishes a set of assumptions and derives conditions that characterize the effective noise reduction at different stages.

We first introduce the notation and assumptions used throughout. We then examine the noise-mitigation mechanisms in Stage 1 (consensus aggregation, Section 2.2) and Stage 2 (controlled generation, Section 2.3). Finally, drawing on information-theoretic and learning-theoretic perspectives, we identify sufficient conditions under which the multi-stage framework yields a supervision signal with a lower effective noise rate than E2E learning, thereby leading to a tighter upper bound on the generalization error.

## F.1 NOTATION AND TERMINOLOGY

To maintain consistency with Section 2, we define the unified notation for this theoretical analysis:

- **Task Index**: $\tau \in \{\text{sa}, \text{pc}\}$, denoting the Semantic Adherence and Physical Commonsense tasks, respectively.
- **Input**: $X^\tau$ or its instance $x^\tau$. For SA, $x^{\text{sa}} = (v, c)$ (video $v$ and caption $c$); for PC, $x^{\text{pc}} = v$ (video only).
- **True Label**: $Y^\tau \in \{1, \ldots, 5\}$, representing the discrete ground-truth score (5-point scale).
- **Stage 0 Output**: $S^\tau \in \{1, \ldots, 5\}$, the initial score from the VideoPhy model, serving as side information.

- **Stage 1 Reference Rationale**: $r_{\text{ref}}^\tau$, the output of Stage 1 for task $\tau$, used as the initial seed for Stage 2.

- **Stage 2 Evidence and Rationale**: $(e_i^\tau, r_i^\tau)$ denotes the evidence-rationale pair generated at the $i$-th iteration; $\mathcal{H}_i^\tau = \{(e_j^\tau, r_j^\tau)\}_{j=0}^i$ represents the history up to step $i$.

- **Pass Indicator**: $\text{pass}_i^\tau \in \{0, 1\}$, determined by the discriminator $\mathcal{V}_\tau$, indicating if the current chain passes verification.

- **Ensemble and Sampling Parameters**: $M$ is the number of models in the ensemble for SA; $K$ is the number of candidate samples for PC.

- **Correctness Indicator**:
  - For SA: $Z_m \in \{0, 1\}$ indicates if the rationale from the $m$-th model is correct; the individual accuracy is $p_0^{\text{sa}} = \Pr[Z_m = 1 \mid X^\tau, S^\tau]$.
  - For PC: $p_0^{\text{pc}}$ denotes the probability that a single sample yields a correct rationale (conditioned on input and side information).

- **Discriminator Performance**: True Positive Rate (Recall) $\alpha = \Pr[\text{pass} = 1 \mid \text{chain is correct}]$; True Negative Rate (Specificity) $\beta = \Pr[\text{pass} = 0 \mid \text{chain is incorrect}]$.

- **Strategy Coverage Lower Bound**: $q_{\min}^\tau$ (Assumption A5), the minimum probability lower bound for generating a correct chain at any step.

- **Iteration Count**: $T$ is the iteration limit in Stage 2 (excluding the seed and fallback step). The total number of attempts is $t = T + 2$ (including seed generation and the final LabelRethink fallback).

- **Effective Noise Rate**:
  - $\eta_1^\tau$: Error rate of the Stage 1 output.
  - $\eta_2^\tau$: Error rate of the Stage 2 controller's output.
  - $\eta_{\text{multi}}^\tau$: Effective noise rate of the final training data (input to Stage 3).
  - $\eta_{\text{e2e}}^\tau$: Noise rate of the E2E supervision signal.

- **Information Measures**: $I(\cdot; \cdot \mid \cdot)$ denotes conditional mutual information, $H(\cdot)$ denotes entropy.

F.2    FUNDAMENTAL ASSUMPTIONS

Our analysis is based on the following assumptions. While often relaxable, they are stated in their strong form for simplicity.

(A1) **Stage 0 Side Information Validity**: The side information $S^\tau$ provides meaningful information about the true label $Y^\tau$, i.e., $\exists \delta_S > 0$ such that:

$$I(Y^\tau; S^\tau \mid X^\tau) \geq \delta_S.$$

(A2) **Stage 1 Base Model Accuracy and Correlation**:
  - **SA**: For the $M$ base models, the correctness indicators $Z_m$ given input and side information satisfy $\Pr[Z_m = 1 \mid X^\tau, S^\tau] = p_0^{\text{sa}} > 1/2$. The Pearson correlation between any pair is bounded: $\text{Corr}(Z_m, Z_{m'}) \leq \rho \in [0, 1)$.
  - **PC**: The base model generates candidate rationales via $K$ independent samplings, with single-sample correctness probability $p_0^{\text{pc}} > 0$.

(A3) **Discriminator Competence**: The aggregator $\mathcal{J}$ in Stage 1 and the discriminator $\mathcal{V}_\tau$ in Stage 2 can effectively distinguish correct from incorrect chains, with $\alpha > 1/2$ and $\beta > 1/2$.

(A4) **Conditional Independence of Hidden Reference**: In Stage 2 iteration steps (excluding the seed step), the generated $(e_i^\tau, r_i^\tau)$ is conditionally independent of the reference rationale $r_{\text{ref}}^\tau$, given the current input $X^\tau$ and history $\mathcal{H}_{i-1}^\tau$.

(A5) **Strategy Coverage and Minimum Success Rate**: $\exists q_{\min}^\tau > 0$ such that for all $i = 0, \ldots, T$:

$$\Pr[\mathcal{G}^\tau(e_i^\tau, r_i^\tau) = 1 \mid X^\tau, \mathcal{H}_{i-1}^\tau] \geq q_{\min}^\tau.$$

This ensures a non-zero chance of generating a correct chain at any step.

(A6) **LabelRethink Fallback**: If all $T$ iterations fail, the LabelRethink module, when injected with $r_{\text{ref}}^\tau$ and $\mathcal{H}_T^\tau$, produces a correct chain with probability at least $q_{\text{re}}^\tau \geq q_{\min}^\tau$.

(A7) **(Approximate) Independence**: To apply concentration inequalities, we assume:

- For SA: The $M$ models can be partitioned into $g$ groups, with outputs independent across groups (allowing correlation within groups).
- For Stage 2: The outcomes of the $t$ attempts are approximately independent under the discriminator's judgment.

This can be approximately achieved by using diverse model sources and the hidden reference strategy.

### F.3 STAGE 1: CONSENSUS AGGREGATION AND NOISE REDUCTION

Stage 1 produces a more reliable reference rationale $r_{\text{ref}}^{\tau}$ via ensemble (SA) or sampling-selection (PC), leveraging collective intelligence to reduce the error rate.

**Lemma F.1** (Error Upper Bound for SA Consensus). *Under Assumption (A2), let $S = \sum_{m=1}^{M} Z_m$ and the majority vote be $\hat{Z} = \mathbf{1}\{S > M/2\}$. Then:*

*(a) (Variance-Based Weak Bound) Generally, the error probability is bounded by:*

$$\Pr[\hat{Z} = 0] \leq \frac{p_0^{\text{sa}}(1 - p_0^{\text{sa}})}{M_{\text{eff}}(p_0^{\text{sa}} - 1/2)^2}, \quad where \quad M_{\text{eff}} = \frac{M}{1 + (M-1)\rho}.$$

*(b) (Exponential Bound) Under the group independence assumption (A7) with $g$ groups:*

$$\Pr[\hat{Z} = 0] \leq \exp\left(-2g(p_0^{\text{sa}} - 1/2)^2\right).$$

*Proof.* (a) Let $p = p_0^{\text{sa}}$. We have $\mathbb{E}[S] = Mp$. The error event $\{S \leq M/2\}$ is equivalent to $\mathbb{E}[S] - S \geq M(p - 1/2)$. By Chebyshev's inequality:

$$\Pr\left(\mathbb{E}[S] - S \geq t\right) \leq \frac{\text{Var}(S)}{t^2}.$$

Setting $t = M(p - 1/2)$, we bound the variance:

$$\text{Var}(S) = \sum_m \text{Var}(Z_m) + \sum_{m \neq m'} \text{Cov}(Z_m, Z_{m'})$$
$$\leq Mp(1 - p) + M(M-1)\rho p(1 - p)$$
$$= p(1 - p)M\left[1 + (M-1)\rho\right].$$

Substitution yields the weak bound. (b) Partition the $M$ models into $g$ groups of size $b$ ($M = gb$). Define the group average $\bar{Z}_j = \frac{1}{b}\sum_{m \in \text{group } j} Z_m$. The $\{\bar{Z}_j\}_{j=1}^{g}$ are independent, and $\mathbb{E}[\bar{Z}_j] = p$. Majority vote failure is equivalent to $\bar{Z} = \frac{1}{g}\sum_{j=1}^{g} \bar{Z}_j \leq 1/2$. Applying Hoeffding's inequality for bounded variables gives the exponential bound. $\square$

**Lemma F.2** (Existence Lower Bound for PC Candidate Selection). *Under Assumptions (A2) and (A3), the probability that the selected reference rationale in PC is correct is bounded by:*

$$\Pr[r_{\text{ref}}^{\text{pc}} \text{ is correct}] \geq \alpha\left(1 - (1 - p_0^{\text{pc}})^K\right).$$

*Proof.* The probability that at least one candidate is correct is $1 - (1 - p_0^{\text{pc}})^K$. Conditioned on this event, the discriminator selects a correct candidate with probability at least $\alpha$ (true positive rate). The overall lower bound is the product of these probabilities. $\square$

**Corollary F.3** (Upper Bound on Stage 1 Effective Noise Rate). *Let $\eta_1^{\tau} = \Pr[r_{\text{ref}}^{\tau} \text{ is incorrect}]$. From Lemmas F.1 and F.2, we have:*

$$\eta_1^{\text{sa}} \leq \frac{p_0^{\text{sa}}(1 - p_0^{\text{sa}})}{M_{\text{eff}}(p_0^{\text{sa}} - 1/2)^2} \quad (weak \ bound),$$
$$\eta_1^{\text{pc}} \leq 1 - \alpha\left(1 - (1 - p_0^{\text{pc}})^K\right).$$

*The bound for SA can be strengthened to the exponential form if the group independence assumption holds.*

**Discussion and Practical Implications**

- Stage 1 significantly reduces the supervision noise via aggregation and selection.

- For SA, model diversity (low $\rho$) is crucial. High correlation diminishes the ensemble effect ($M_{\text{eff}}$ decreases). Using diverse models (architectures, pre-training, prompts) is recommended. Group independence enables exponential error reduction.

- For PC, increasing the sample size $K$ and improving the discriminator's TPR $\alpha$ are key to reducing the error rate.

### F.4 STAGE 2: CONTROLLER PASS PROBABILITY AND ERROR ANALYSIS

Stage 2 employs controlled iterative generation and verification to find a correct reasoning chain. Its core is using multiple attempts and discriminator validation to further enhance the probability of obtaining a correct rationale.

**Design Principle: Hiding the Reference for Information Gain** The hidden reference strategy (Assumption A4) is central to Stage 2. The following proposition shows that this conditional independence ensures each iterative step provides new information about $Y^\tau$, preventing the model from simply parroting the reference rationale and causing information redundancy.

**Proposition F.4** (Information Gain under Conditional Independence). *Under Assumption (A4), for any $i \geq 1$:*

$$I(Y^\tau; e_i^\tau \mid X^\tau, \mathcal{H}_{i-1}^\tau, r_{\text{ref}}^\tau) = I(Y^\tau; e_i^\tau \mid X^\tau, \mathcal{H}_{i-1}^\tau).$$

*Consequently, for the ultimately adopted evidence set $E^\tau = \{e_j^\tau\}_{j=1}^{i^*}$, the cumulative mutual information satisfies:*

$$I(Y^\tau; E^\tau \mid X^\tau) \geq \sum_{j=1}^{i^*} I(Y^\tau; e_j^\tau \mid X^\tau, \mathcal{H}_{j-1}^\tau).$$

*Proof.* The equality follows directly from the definition of conditional mutual information and (A4). The inequality results from the chain rule for mutual information and the non-negativity of each term. □

This property ensures the benefits of $t$ attempts in Theorem 2 stem from cumulative, incremental information gain.

Define the probability bounds for a single attempt being a true pass and a false pass:

$$\pi_{\text{TP}}^\tau \geq q_{\min}^\tau \alpha, \quad \pi_{\text{FP}}^\tau \leq (1 - q_{\min}^\tau)(1 - \beta).$$

A single attempt generates a correct chain and gets accepted with probability at least $q_{\min}^\tau \alpha$; it generates an incorrect chain but gets falsely accepted with probability at most $(1 - q_{\min}^\tau)(1 - \beta)$.

**Theorem F.5** (Controller Pass Probability and False Pass Upper Bound). *Under Assumptions (A3)–(A6) and the approximate independence assumption (A7), let the total number of attempts be $t = T+2$. Then:*

1. *The probability of eventually accepting at least one correct chain is lower bounded by:*

$$P_{\text{TP}} = \Pr[\text{Eventually accept a correct chain}] \geq 1 - (1 - \pi_{\text{TP}}^\tau)^t.$$

2. *The probability of eventually accepting at least one incorrect chain is upper bounded by:*

$$P_{\text{FP}} = \Pr[\text{Eventually accept an incorrect chain}] \leq 1 - (1 - \pi_{\text{FP}}^\tau)^t.$$

3. *The effective noise rate of the controller's output satisfies:*

$$\eta_2^\tau = \Pr[\text{Final output is incorrect} \mid \text{Accepted}] \leq \frac{P_{\text{FP}}}{P_{\text{TP}} + P_{\text{FP}}}$$

$$\leq \frac{1 - (1 - \pi_{\text{FP}}^\tau)^t}{(1 - (1 - \pi_{\text{TP}}^\tau)^t) + (1 - (1 - \pi_{\text{FP}}^\tau)^t)}.$$

*Proof.* Under approximate independence, the probability of no true pass in $t$ attempts is $\leq (1 - \pi_{\mathrm{TP}}^\tau)^t$, so $P_{\mathrm{TP}} \geq 1 - (1 - \pi_{\mathrm{TP}}^\tau)^t$. Similarly, $P_{\mathrm{FP}} \leq 1 - (1 - \pi_{\mathrm{FP}}^\tau)^t$. The noise rate $\eta_2^\tau$ is the conditional probability that the first accepted chain is incorrect. Using the bounds for $P_{\mathrm{TP}}$ and $P_{\mathrm{FP}}$ yields the conservative upper bound. $\square$

**Proposition F.6** (Iteration Complexity for Logarithmic Rate). *If attempts are independent and the single-shot success probability is lower bounded by $\pi = \pi_{\mathrm{TP}}^\tau > 0$, then to achieve* $\Pr[\text{At least one success}] \geq 1 - \epsilon$, *the number of attempts $t$ must satisfy:*

$$t \geq \frac{1}{\pi} \log \frac{1}{\epsilon}.$$

*Proof.* From $1 - (1 - \pi)^t \geq 1 - e^{-\pi t} \geq 1 - \epsilon$, solving for $t$ yields the result. $\square$

**Discussion and Practical Implications**

- $P_{\mathrm{TP}}$ approaches 1 exponentially fast with $t$, while $P_{\mathrm{FP}}$ grows slower ($\pi_{\mathrm{FP}}^\tau \ll \pi_{\mathrm{TP}}^\tau$). Thus, an accurate discriminator ($\alpha, \beta$ large) and good strategy coverage ($q_{\min}^\tau$ large) enable Stage 2 to output rationales with very low error.

- The required $t$ scales with $1/\pi$. Improving the single-shot success probability $\pi$ (via better prompts, diversity, or discriminator $\alpha$) is more efficient than blindly increasing $T$.

### F.5 STAGE 3: GENERALIZATION BOUND UNDER NOISY SUPERVISION

Stage 3 trains the scoring prediction model using the (potentially noisy) rationale-score pairs $(r^\tau, Y^\tau)$ from previous stages. We use the Massart noise model to analyze noisy supervised learning and compare the generalization bounds.

**Theorem F.7** (Generalization Upper Bound under Massart Noise (Massart & Élodie Nédélec (2006))). *Let the hypothesis space $\mathcal{H}$ have complexity measured by $d$ (e.g., VC dimension), the training set size be $n$, and the loss function $\ell$ be bounded in $[0, 1]$ and Lipschitz. If the effective noise rate of the supervision signal is bounded by $\eta < 1/2$ (Massart condition), then for the ERM solution $\hat{h}$, with probability at least $1 - \delta$, the generalization error satisfies:*

$$R(\hat{h}) - R(h^*) \leq C_1 \sqrt{\frac{d + \log(1/\delta)}{n}} + C_2 \eta.$$

*Here, $h^*$ is the Bayes optimal hypothesis under no noise, and $C_1, C_2 > 0$ are constants related to the loss function.*

*Proof Sketch.* The bound decomposes into two parts: 1. **Estimation Error (Uniform Convergence)**: For bounded loss, VC/Rademacher theory gives $\sup_{h \in \mathcal{H}} |R(h) - \hat{R}_n(h)| \leq C_1 \sqrt{(d + \log(1/\delta))/n}$. 2. **Approximation Error (Noise Bias)**: Massart noise introduces a bias term in the risk of the optimal hypothesis, linearly related to $\eta$, i.e., $|R(h^*) - R_{\mathrm{noisy}}(h_{\mathrm{noisy}}^*)| \leq C_2 \eta$. Combining these two parts yields the theorem. See standard results in noisy learning theory for a complete proof. $\square$

**Multi-Stage vs. End-to-End** Applying Theorem F.7 to the multi-stage method ($\eta = \eta_{\mathrm{multi}}^\tau$) and the E2E method ($\eta = \eta_{\mathrm{e2e}}^\tau$), it is clear that if:

$$\eta_{\mathrm{multi}}^\tau < \eta_{\mathrm{e2e}}^\tau,$$

then, for the same $n$ and $d$, the multi-stage method enjoys a tighter (smaller) generalization error upper bound.

### F.6 SUFFICIENT CONDITION FOR MULTI-STAGE SUPERIORITY

We now synthesize the results from previous stages to establish a sufficient condition under which the multi-stage framework outperforms the E2E baseline.

The final effective noise rate $\eta_{\mathrm{multi}}^\tau$ for Stage 3 is a convex combination:

$$\eta_{\mathrm{multi}}^\tau = \Pr[A] \cdot \eta_2^\tau + (1 - \Pr[A]) \cdot \eta_1^\tau,$$

where $\Pr[A]$ is the probability that a Stage 2 candidate is accepted. Consequently,

$$\min(\eta_1^\tau, \eta_2^\tau) \le \eta_{\mathrm{multi}}^\tau \le \max(\eta_1^\tau, \eta_2^\tau).$$

Crucially, if both $\eta_1^\tau$ and $\eta_2^\tau$ are less than $\eta_{\mathrm{e2e}}^\tau$, then $\eta_{\mathrm{multi}}^\tau < \eta_{\mathrm{e2e}}^\tau$ necessarily holds.

**Theorem F.8** (Sufficient Condition for Multi-Stage Superiority). *Under the assumptions of Lemmas F.1, F.2 and Theorem F.5, if the system parameters $\left(M, \rho, p_0^{\mathrm{sa}}, K, p_0^{\mathrm{pc}}, \alpha, \beta, T, q_{\min}^\tau\right)$ satisfy:*

*(SA)* $\quad \dfrac{p_0^{\mathrm{sa}}(1 - p_0^{\mathrm{sa}})}{M_{\mathrm{eff}}(p_0^{\mathrm{sa}} - 1/2)^2} < \eta_{\mathrm{e2e}}^{\mathrm{sa}},$

*(PC)* $\quad 1 - \alpha\left(1 - (1 - p_0^{\mathrm{pc}})^K\right) < \eta_{\mathrm{e2e}}^{\mathrm{pc}},$

*(Controller)* $\quad \dfrac{1 - (1 - \pi_{\mathrm{FP}}^\tau)^t}{(1 - (1 - \pi_{\mathrm{TP}}^\tau)^t) + (1 - (1 - \pi_{\mathrm{FP}}^\tau)^t)} < \eta_{\mathrm{e2e}}^\tau, \quad \tau \in \{\mathrm{sa}, \mathrm{pc}\}$

*where $t = T + 2$, $\pi_{\mathrm{TP}}^\tau \ge q_{\min}^\tau \alpha$, $\pi_{\mathrm{FP}}^\tau \le (1 - q_{\min}^\tau)(1 - \beta)$, then:*

$$\eta_{\mathrm{multi}}^\tau < \eta_{\mathrm{e2e}}^\tau.$$

*Furthermore, by Theorem F.7, the multi-stage method achieves a strictly tighter generalization error bound than the E2E method.*

*Proof.* By Corollary F.3, $\eta_1^\tau$ is upper bounded by the left-hand side of the first two inequalities. By Theorem F.5, $\eta_2^\tau$ is upper bounded by the left-hand side of the third inequality. The sufficient condition ensures $\eta_1^\tau < \eta_{\mathrm{e2e}}^\tau$ and $\eta_2^\tau < \eta_{\mathrm{e2e}}^\tau$. Since $\eta_{\mathrm{multi}}^\tau$ is a convex combination of $\eta_1^\tau$ and $\eta_2^\tau$, it must also be less than $\eta_{\mathrm{e2e}}^\tau$. Applying Theorem F.7 concludes the proof. $\square$

**Why is this Condition Plausible?** This sufficient condition is not an overly strict requirement but a achievable goal through careful design. It holds because the multi-stage framework constructs an **error-reduction pipeline**: **Stage 1** reduces noise through **statistical aggregation** (collective intelligence). If base models are better than random ($p_0 > 1/2$) and not perfectly correlated ($\rho < 1$), aggregation *provably* lowers the error rate below the single-model E2E baseline ($\eta_1^\tau < \eta_{\mathrm{e2e}}^\tau$). **Stage 2** reduces noise through **active exploration and verification** (multiple trials). If the strategy has a non-zero chance of being correct ($q_{\min}^\tau > 0$) and the discriminator is better than random ($\alpha, \beta > 1/2$), then with sufficient attempts ($T$ large enough), the probability of finding and accepting a correct chain approaches 1 exponentially fast, driving the controller's error rate very low ($\eta_2^\tau < \eta_{\mathrm{e2e}}^\tau$). The final noise rate $\eta_{\mathrm{multi}}^\tau$, being an average of these two lower rates, is therefore guaranteed to be lower than the E2E baseline. The architecture's synergistic effect ensures superiority even if no single component is perfect.

## F.7 SUMMARY AND EMPIRICAL VALIDATION SUGGESTIONS

This formal analysis indicates that, under the stated assumptions:

- **Noise Reduction Mechanism**: Stages 1 and 2 can effectively reduce the supervision noise rate $\eta_{\mathrm{multi}}^\tau$ observed in the training signal for Stage 3.

- **Generalization Advantage**: Within the Massart noise model, a reduced supervision noise rate implies a tighter generalization error bound, suggesting that the multi-stage framework may achieve better generalization than the E2E approach under such conditions.

## G ADDITIONAL EXPERIMENTS AND ANALYSES

In this appendix, we provide additional quantitative and qualitative analyses of *Cosmos-Eval*. We describe how each experiment is constructed and report the corresponding results in tables. Unless otherwise noted, all correlations are computed against human 5-point SA/PC labels and 95% confidence intervals are obtained via the standard Fisher $r \to z \to r$ transform.

Table 9: **Cross-benchmark results on AIGVE-Bench and LG-VQA.** Pearson correlations between automatic evaluators and human scores on two independent evaluation suites.

| Model | AIGVE-Bench | | LG-VQA | |
|---|---|---|---|---|
| | Pearson $r$ | 95% CI | Pearson $r$ | 95% CI |
| Cosmos-Eval | 0.1986 | [0.1561, 0.2326] | 0.2759 | [0.2414, 0.3097] |
| VideoPhy-2-AutoEval | 0.2089 | [0.1706, 0.2466] | 0.2750 | [0.2404, 0.3088] |
| Qwen2.5-VL-7B | 0.1033 | [0.0063, 0.1425] | 0.2013 | [0.1656, 0.2366] |

Table 10: **PC-based ranking of T2V generators on AIGVE-Bench.** Mean Cosmos-Eval PC score per generator.

| Rank | Model | Mean PC Score | #Videos |
|---|---|---|---|
| 1 | CogVideoX | 4.6830 | 470 |
| 2 | Pyramid | 4.6311 | 488 |
| 3 | Hunyuan | 4.6268 | 493 |
| 4 | Sora | 4.6207 | 493 |
| 5 | Genmo | 4.5658 | 486 |

## G.1 CROSS-BENCHMARK GENERALIZATION

To assess whether Cosmos-Eval overfits to the training benchmarks (VideoPhy/VideoPhy-2), we additionally evaluate it on two independent suites: *AIGVE-Bench*(Xiang et al., 2025) and *LG-VQA*(Ghosal et al., 2023). Both datasets contain videos generated by multiple T2V models with human scores. We directly apply Cosmos-Eval (without any additional fine-tuning) and compare its correlation with human scores to that of VideoPhy-2-AutoEval and Qwen2.5-VL-7B.

Table 9 reports Pearson correlations and 95% confidence intervals on both AIGVE-Bench and LG-VQA for the three evaluators.

## G.2 RANKING T2V GENERATORS ON AIGVE-BENCH

To demonstrate the practical utility of Cosmos-Eval for comparing T2V models, we use it to rank several state-of-the-art generators on AIGVE-Bench under the PC (physical commonsense) task. For each generator, we compute the mean PC score across all clips associated with that model.

Table 10 reports the resulting ranking. Newer models (e.g., CogVideoX(Yang et al., 2025b), Pyramid(Jin et al., 2025), Hunyuan(Kong et al., 2025)) achieve higher mean PC scores than earlier systems such as Genmo(Li et al., 2025b), and Sora(Liu et al., 2024c) no longer dominates once physical plausibility is explicitly emphasized.

## G.3 HUMAN EVALUATION OF SA/PC RATIONALES

We conduct a human study to directly assess the perceived quality of SA/PC rationales using a custom web interface (Fig. 50). Annotators are shown the video, the caption (for SA), and a candidate explanation from one of three models (Cosmos-Eval, GPT-4V(OpenAI et al., 2024), Qwen3-VL-Plus), with model identity hidden. For each example, annotators score the explanation along the five rubric dimensions defined in our PC and SA reason-quality rubrics (Tables 7 and 8), namely grounding, temporal alignment, internal consistency, criteria/decision justification, and either video-quality assessment (PC) or coverage & specificity with respect to the caption (SA). Each dimension is scored using the three-point anchors $\{0, 0.5, 1\}$, matching the definitions in Tables 7 and 8. In total, the study contains 1,500 evaluations across SA and PC.

Table 11 presents average scores for SA rationales across grounding, temporal alignment, consistency, alignment justification, and coverage & specificity. Table 12 shows the corresponding results for PC rationales.

Table 11: **Human evaluation of SA rationales.** Average scores on grounding, temporal alignment, consistency, alignment justification, coverage & specificity, and overall average.

| Model | Grounding | Temporal Align. | Consistency | Align. Justif. | Coverage & Spec. | Total Avg. |
|---|---|---|---|---|---|---|
| Cosmos-Eval | 0.82 | 0.57 | 0.71 | 0.81 | 0.87 | 0.76 |
| GPT-4V | 0.64 | 0.52 | 0.67 | 0.61 | 0.64 | 0.62 |
| Qwen3-VL-Plus | 0.73 | 0.53 | 0.69 | 0.73 | 0.71 | 0.69 |

Table 12: **Human evaluation of PC rationales.** Average scores on grounding, temporal reasoning, consistency, criteria & justification, video-quality awareness, and overall average.

| Model | Grounding | Temporal | Consistency | Criteria & Justif. | Video Quality | Total Avg. |
|---|---|---|---|---|---|---|
| Cosmos-Eval | 0.79 | 0.56 | 0.82 | 0.85 | 0.82 | 0.77 |
| GPT-4V | 0.64 | 0.52 | 0.56 | 0.64 | 0.56 | 0.58 |
| Qwen3-VL-Plus | 0.59 | 0.51 | 0.59 | 0.64 | 0.63 | 0.60 |

Table 13: **PC rationales scored by Qwen3-VL-Plus.**

| Model | Grounding | Temporal | Consistency | Criteria & Justif. | Video Quality | Total Avg. |
|---|---|---|---|---|---|---|
| Cosmos-Eval | 0.71 | 0.79 | 0.58 | 0.54 | 0.69 | 0.662 |
| GPT-4V | 0.60 | 0.82 | 0.30 | 0.26 | 0.72 | 0.544 |
| Qwen3-VL-Plus | 0.65 | 0.85 | 0.67 | 0.26 | 0.79 | 0.564 |

Table 14: **SA rationales scored by Qwen3-VL-Plus.**

| Model | Grounding | Temporal | Consistency | Align. Justif. | Coverage & Spec. | Total Avg. |
|---|---|---|---|---|---|---|
| Cosmos-Eval | 0.76 | 0.77 | 0.44 | 0.44 | 0.82 | 0.65 |
| GPT-4V | 0.82 | 0.80 | 0.47 | 0.46 | 0.84 | 0.678 |
| Qwen3-VL-Plus | 0.91 | 0.83 | 0.48 | 0.48 | 0.91 | 0.722 |

Table 15: **PC rationales scored by GPT-4V.**

| Model | Grounding | Temporal | Consistency | Criteria & Justif. | Video Quality | Total Avg. |
|---|---|---|---|---|---|---|
| Cosmos-Eval | 0.57 | 0.56 | 0.62 | 0.52 | 0.78 | 0.61 |
| GPT-4V | 0.52 | 0.54 | 0.44 | 0.38 | 0.88 | 0.55 |
| Qwen3-VL-Plus | 0.57 | 0.60 | 0.43 | 0.35 | 0.78 | 0.55 |

### G.4 VLM-JUDGE EVALUATION OF RATIONALES

To further probe explanation quality in a model-agnostic way, we use several strong VLMs as external judges. Each judge scores SA/PC rationales from Cosmos-Eval, GPT-4V, and Qwen3-VL-Plus along the same rubric dimensions as in the human study, producing scores in $0, 0.5, 1$. Below we report dimension-wise averages and overall means per judge.

#### G.4.1 QWEN3-VL-PLUS AS JUDGE

In this setting, we fix Qwen3-VL-Plus as the judge and ask it to assign rubric scores to PC and SA rationales produced by Cosmos-Eval, GPT-4V, and Qwen3-VL-Plus itself. Tables 13 and 14 report the dimension-wise averages and overall mean scores.

#### G.4.2 GPT-4V AS JUDGE

Here we use GPT-4V as the judge and follow the same protocol: given a video, caption (for SA), and a candidate rationale from each model, GPT-4V assigns rubric scores to SA/PC explanations. Tables 15 and 16 summarize the resulting averages.

#### G.4.3 GEMINI-2.5-PRO AS JUDGE

Finally, we repeat the same evaluation protocol with Gemini-2.5-Pro as the judge. Tables 17 and 18 report the average rubric scores for PC and SA rationales, respectively.

Table 16: **SA rationales scored by GPT-4V.**

| Model | Grounding | Temporal | Consistency | Align. Justif. | Coverage & Spec. | Total Avg. |
|---|---|---|---|---|---|---|
| Cosmos-Eval | 0.67 | 0.46 | 0.74 | 0.71 | 0.64 | 0.64 |
| GPT-4V | 0.66 | 0.50 | 0.73 | 0.72 | 0.64 | 0.65 |
| Qwen3-VL-Plus | 0.73 | 0.55 | 0.73 | 0.70 | 0.69 | 0.68 |

Table 17: **PC rationales scored by Gemini-2.5-Pro.**

| Model | Grounding | Temporal | Consistency | Criteria & Justif. | Video Quality | Total Avg. |
|---|---|---|---|---|---|---|
| Cosmos-Eval | 0.51 | 0.50 | 0.41 | 0.34 | 0.01 | 0.36 |
| GPT-4V | 0.49 | 0.46 | 0.15 | 0.10 | 0.00 | 0.24 |
| Qwen3-VL-Plus | 0.50 | 0.52 | 0.21 | 0.16 | 0.04 | 0.29 |

Table 18: **SA rationales scored by Gemini-2.5-Pro.**

| Model | Grounding | Temporal | Consistency | Align. Justif. | Coverage & Spec. | Total Avg. |
|---|---|---|---|---|---|---|
| Cosmos-Eval | 0.47 | 0.46 | 0.25 | 0.29 | 0.63 | 0.42 |
| GPT-4V | 0.63 | 0.61 | 0.35 | 0.30 | 0.64 | 0.51 |
| Qwen3-VL-Plus | 0.70 | 0.67 | 0.29 | 0.27 | 0.77 | 0.54 |

Table 19: **Score correlations on VideoPhy-2 (50 clips).** Direct SA/PC scoring by frontier VLMs vs. Cosmos-Eval.

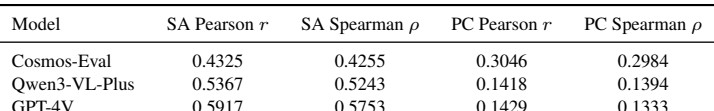

| Model | SA Pearson $r$ | SA Spearman $\rho$ | PC Pearson $r$ | PC Spearman $\rho$ |
|---|---|---|---|---|
| Cosmos-Eval | 0.4325 | 0.4255 | 0.3046 | 0.2984 |
| Qwen3-VL-Plus | 0.5367 | 0.5243 | 0.1418 | 0.1394 |
| GPT-4V | 0.5917 | 0.5753 | 0.1429 | 0.1333 |

Table 20: **Rationale similarity on VideoPhy-2 (50 clips).** BLEU-4 and BERTScore-F1 (in %) for SA/PC explanations vs. reference rationales.

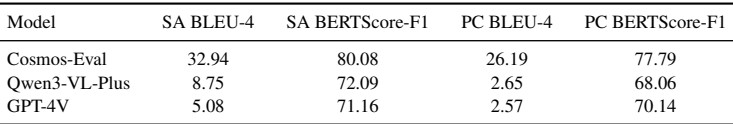

| Model | SA BLEU-4 | SA BERTScore-F1 | PC BLEU-4 | PC BERTScore-F1 |
|---|---|---|---|---|
| Cosmos-Eval | 32.94 | 80.08 | 26.19 | 77.79 |
| Qwen3-VL-Plus | 8.75 | 72.09 | 2.65 | 68.06 |
| GPT-4V | 5.08 | 71.16 | 2.57 | 70.14 |

Table 21: **Summac scores for SA rationales.**

| Metric | Cosmos-Eval | Qwen2.5-VL-7B | InternVL-8B | InternVL-9B | InternVL-14B | Cosmos-Reason1 | VideoLLaMA3-7B |
|---|---|---|---|---|---|---|---|
| summac | 26.62 | 21.50 | 23.92 | 24.22 | 23.91 | 21.23 | 22.56 |

## G.5 FRONTIER VLM BASELINES ON VIDEOPHY-2

To compare Cosmos-Eval against frontier VLMs used with direct prompting, we sample 50 VideoPhy-2 test clips with human SA/PC labels. We prompt GPT-4V and Qwen3-VL-Plus to directly output 5-point SA/PC scores and compute correlations with human labels.

Table 19 reports Pearson and Spearman correlations for scores. Table 20 reports BLEU-4 and BERTScore-F1 for SA/PC rationales against reference explanations.

## G.6 FACTUAL CONSISTENCY METRICS FOR RATIONALES

Beyond surface-level text similarity metrics, we also consider a factual/consistency-oriented metric (SummAc(Laban et al., 2022)) to assess alignment between generated and reference explanations. Table 21 reports Summac scores for SA rationales, and Table 22 for PC rationales, comparing Cosmos-Eval to several baselines.

Table 22: **Summac scores for PC rationales.**

| Metric | Cosmos-Eval | Qwen2.5-VL-7B | InternVL-8B | InternVL-9B | InternVL-14B | Cosmos-Reason1 | VideoLLaMA3-7B |
|---|---|---|---|---|---|---|---|
| summac | 23.32 | 22.69 | 23.07 | 22.86 | 23.16 | 22.25 | 23.20 |

Table 23: **SA correlations with uncertainty.** Pearson $r$ (95% CI), two-sided $p$-value, and $\Delta r$ vs. Cosmos-Eval.

| Model | $r$ [95% CI] | $p$-value | $\Delta r$ vs. Cosmos-Eval |
|---|---|---|---|
| Cosmos-Eval | 0.4643 [0.4376, 0.4904] | 2.50E-181 | — |
| VideoPhy-2-AutoEval | 0.4327 [0.4049, 0.4596] | 5.12E-155 | +0.0316 |
| Qwen2.5-VL-7B | 0.3808 [0.3517, 0.4092] | 1.02E-117 | +0.0835 |
| VideoLLaMA3-7B | 0.2769 [0.2456, 0.3077] | 7.21E-61 | +0.1874 |
| InternVL-8B | 0.4143 [0.3861, 0.4418] | 4.70E-141 | +0.0500 |
| InternVL-9B | 0.3827 [0.3536, 0.4110] | 6.34E-119 | +0.0816 |
| InternVL-14B | 0.3420 [0.3120, 0.3714] | 7.17E-94 | +0.1223 |
| Cosmos-Reason1 | 0.3662 [0.3366, 0.3952] | 3.62E-107 | +0.0981 |
| VideoLLaMA3-7B (variant) | 0.2333 [0.2034, 0.2636] | 7.78E-44 | +0.2310 |

Table 24: **PC correlations with uncertainty.** Pearson $r$ (95% CI), two-sided $p$-value, and $\Delta r$ vs. Cosmos-Eval.

| Model | $r$ [95% CI] | $p$-value | $\Delta r$ vs. Cosmos-Eval |
|---|---|---|---|
| Cosmos-Eval | 0.3641 [0.3346, 0.3929] | 4.83E-107 | — |
| VideoPhy-2-AutoEval | 0.3646 [0.3351, 0.3934] | 2.55E-107 | -0.0005 |
| Qwen2.5-VL-7B | 0.0840 [0.0512, 0.1180] | 7.59E-07 | +0.2801 |
| VideoLLaMA3-7B | 0.0640 [0.0310, 0.0980] | 1.67E-04 | +0.3001 |
| InternVL-8B | 0.1665 [0.1280, 0.1935] | 3.79E-21 | +0.1976 |
| InternVL-9B | 0.1304 [0.0973, 0.1634] | 2.26E-14 | +0.2337 |
| InternVL-14B | 0.1956 [0.1631, 0.2278] | 1.18E-30 | +0.1685 |
| Cosmos-Reason1 | 0.2356 [0.2030, 0.2665] | 7.78E-44 | +0.1285 |
| VideoLLaMA3-7B (variant) | 0.2075 [0.1795, 0.2354] | 2.30E-43 | +0.1566 |

Table 25: **GPT-4o as external scorer.** Pearson correlations between GPT-4o-implied scores and Cosmos-Eval scores under two Stage-2 judges.

| Setting | $N$ | Pearson $r$ | 95% CI |
|---|---|---|---|
| PC, 72B judge (Qwen2.5-VL-72B) | 187 | 0.9131 | [0.8857, 0.9342] |
| PC, 30B judge (Qwen3-VL-30B) | 187 | 0.8239 | [0.7716, 0.8651] |
| SA, 72B judge (Qwen2.5-VL-72B) | 178 | 0.8894 | [0.8541, 0.9166] |
| SA, 30B judge (Qwen3-VL-30B) | 184 | 0.8636 | [0.8216, 0.8963] |

### G.7 UNCERTAINTY ESTIMATES FOR MAIN SA/PC RESULTS

For the main SA/PC score correlations, we also report uncertainty estimates and effect sizes. For each model, we compute: (i) Pearson correlation $r$ with human scores, (ii) 95% confidence interval, (iii) two-sided $p$-value for $H_0 : r = 0$, and (iv) $\Delta r$ relative to Cosmos-Eval.

Tables 23 and 24 summarize SA and PC statistics, respectively.

### G.8 SENSITIVITY TO STAGE-2 JUDGE AND EXTERNAL SCORERS

To study judge-choice sensitivity and potential circularity, we vary the Stage-2 judge (Qwen2.5-VL-72B vs. Qwen3-VL-30B) while keeping the rest of the pipeline fixed. For each setting, we ask independent external LLMs (GPT-4o and DeepSeek) to read Cosmos-Eval rationales and assign SA/PC scores, then compute the correlation between these external scores and the original Cosmos-Eval scores.

Table 25 reports Pearson correlations when GPT-4o is used as the external scorer, and Table 26 reports the same when DeepSeek is used. In all cases, we observe high agreement across judge choices and external scorers, and the larger Stage-2 judge (Qwen2.5-VL-72B) consistently yields higher correlations than the 30B variant, indicating that Stage 2 benefits from stronger VLM judges; accordingly, we adopt Qwen2.5-VL-72B as the default Stage-2 judge in our main experiments.

Table 26: **DeepSeek as external scorer.** Pearson correlations between DeepSeek-implied scores and Cosmos-Eval scores under two Stage-2 judges.

| Setting | $N$ | Pearson $r$ | 95% CI |
|---|---|---|---|
| PC, 72B judge (Qwen2.5-VL-72B) | 187 | 0.9131 | [0.8857, 0.9342] |
| PC, 30B judge (Qwen3-VL-30B) | 187 | 0.8487 | [0.8031, 0.8845] |
| SA, 72B judge (Qwen2.5-VL-72B) | 178 | 0.8894 | [0.8541, 0.9166] |
| SA, 30B judge (Qwen3-VL-30B) | 184 | 0.8649 | [0.8233, 0.8973] |

Table 27: **Stage-wise cost of teacher pipeline for PC (200 samples).**

| Step | GPU Count | Inference Time | Sample Count | Avg. Time / Sample |
|---|---|---|---|---|
| Stage 0 (pre-processing) | 1 | 2m12s | 200 | 0.66 s |
| Stage 1: Qwen Gen (run 1) | 1 | 18m43s | 200 | 19.38 s |
| Stage 1: Qwen Gen (run 2) | 2 | 18m37s | 200 | 29.25 s |
| Stage 2 (reasoning ctrl.) | 2 | 188m48s | 200 | 56.64 s |

Table 28: **Stage-wise cost of teacher pipeline for SA (200 samples).**

| Step | GPU Count | Inference Time | Sample Count | Avg. Time / Sample |
|---|---|---|---|---|
| Stage 0 (pre-processing) | 1 | 2m14s | 200 | 0.67 s |
| Stage 1: Qwen Gen | 4 | 54m43s | 200 | 16.31 s |
| Stage 1: Tarsier Gen | 4 | 18m19s | 200 | 20.78 s |
| Stage 1: Qwen3 merge | 2 | 17m39s | 200 | 5.14 s |
| Stage 2 (complex reasoning ctrl.) | 4 | 340m12s | 200 | 102.06 s |

Table 29: **Score-only inference cost for PC/SA scores (200 samples).**

| Step | GPU Count | Inference Time | Sample Count | Avg. Time / Sample | GPU-Hours |
|---|---|---|---|---|---|
| VideoPhy-2-AutoEval-PC | 1 | 2m12s | 200 | 0.66 s | 0.0368 |
| VideoPhy-2-AutoEval-SA | 1 | 2m14s | 200 | 0.67 s | 0.0372 |
| Cosmos-Eval (PC-score) | 1 | 3m42s | 200 | 1.11 s | 0.0618 |
| Cosmos-Eval (SA-score) | 1 | 3m42s | 200 | 1.11 s | 0.0619 |
| Qwen2.5-VL-7B (PC-score) | 1 | 4m20s | 200 | 1.30 s | 0.0722 |
| Qwen2.5-VL-7B (SA-score) | 1 | 4m20s | 200 | 1.30 s | 0.0724 |

Table 30: **Rationale-generation inference cost for PC/SA reasons (200 samples).**

| Step | GPU Count | Inference Time | Sample Count | Avg. Time / Sample | GPU-Hours |
|---|---|---|---|---|---|
| Cosmos-Eval (PC-reason) | 1 | 19m34s | 200 | 5.87 s | 0.3272 |
| Cosmos-Eval (SA-reason) | 1 | 51m49s | 200 | 15.54 s | 0.8637 |
| Qwen2.5-VL-7B (PC-reason) | 1 | 14m14s | 200 | 4.27 s | 0.2042 |
| Qwen2.5-VL-7B (SA-reason) | 1 | 13m34s | 200 | 4.07 s | 0.2219 |

## G.9 COMPUTATIONAL COST AND EFFICIENCY

We report detailed computational costs for (i) the multi-stage teacher pipeline (Stages 0–2) used during training and (ii) the distilled student evaluator (Stage 3) used during inference. All measurements are collected on 200-sample subsets.

Tables 27 and 28 list stage-wise costs for PC and SA teacher pipelines, respectively, while Tables 29 and 30 compare score-only and rationale-generation inference costs for Cosmos-Eval, VideoPhy-2-AutoEval, and Qwen2.5-VL-7B. From these numbers, the total per-sample cost of the teacher pipeline (Stages 0–2) is about 106 s for PC and 145 s for SA, whereas the distilled evaluator (Stage 3) needs only $\approx 1.1\,\text{s}$ per sample for scores alone and $\approx 5.9\,\text{s}$ (PC) / 15.5 s (SA) for scores plus rationales. When we compare the teacher pipeline with the distilled evaluator under the score + rationale setting, this translates to roughly 9–18× speedups. In deployment, users only run the distilled evaluator, whose score-only cost is close to that of a single 7B VLM call, while additionally providing calibrated SA/PC scores and physics-grounded rationales rather than scores alone.

Table 31: **Average PC/SA scores before and after synthetic degradations on VideoPhy-2 clips.** Each row averages 100 samples per distortion type.

| Distortion Type | Before (PC) | After (PC) | PC Δ (B–A) | Before (SA) | After (SA) | SA Δ (B–A) | Count |
|---|---|---|---|---|---|---|---|
| Noise | 4.56 | 4.28 | 0.28 | 4.03 | 3.72 | 0.31 | 100 |
| Occlusion | 4.59 | 4.21 | 0.38 | 4.02 | 3.82 | 0.20 | 100 |
| Blur | 4.64 | 4.32 | 0.32 | 4.00 | 3.72 | 0.28 | 100 |
| Compression | 4.67 | 4.45 | 0.22 | 4.02 | 3.77 | 0.25 | 100 |
| Color | 4.67 | 4.37 | 0.30 | 4.01 | 3.77 | 0.24 | 100 |

Table 32: **Hyperparameter ranges for synthetic degradations.** PC and SA tasks share the same ranges with different random seeds (PC: 42, SA: 43).

| Distortion Type | Parameter | Range | Description |
|---|---|---|---|
| Noise | strength | 10–40 | Noise intensity level |
| Noise | temporal | 5–15 | Temporal noise variation |
| Blur | sigma (luma) | 1.0–4.0 | Gaussian blur radius |
| Blur | chroma_radius | 0.5–2.0 | Chroma blur radius |
| Compression | CRF | 35–45 | Constant rate factor (higher = more compression) |
| Compression | bitrate | 100–300 kbps | Target video bitrate |
| Occlusion | num_boxes | 1–3 | Number of black occlusion boxes |
| Occlusion | box_size | 50–150 px | Width and height of each box |
| Occlusion | position (x,y) | 0–500 px | Random position within frame |
| Color | saturation | 0.3–1.5 | Color saturation multiplier |
| Color | contrast | 0.5–1.3 | Contrast adjustment factor |
| Color | brightness | $-0.2$–0.2 | Brightness offset |
| Color | hue | $-30°$–$30°$ | Hue rotation angle |

Table 33: **Long-horizon evaluation on LongCat-Video (30 prompts, $\approx 33$s per video).** Segment-wise average ranges (across 11 segments) and full-clip averages for SA/PC.

| Model | SA segment range | PC segment range | Full SA avg. | Full PC avg. |
|---|---|---|---|---|
| Cosmos-Eval | 3.03–3.13 | 3.97–4.13 | 3.30 | 4.00 |
| VideoPhy-2-AutoEval | 2.73–3.00 | 3.83–4.07 | 2.97 | 3.50 |

## G.10 ROBUSTNESS TO SYNTHETIC DEGRADATIONS

We examine robustness to synthetic noise by starting from clean VideoPhy-2 clips and applying controlled degradations (noise, occlusion, blur, compression, color shifts). For each distortion type, we compute average PC/SA scores before and after degradation over 100 clips.

Table 31 reports the mean scores for each distortion type and the corresponding drops in PC/SA. Table 32 lists the hyperparameter ranges used to generate each distortion; PC and SA tasks share the same parameter ranges with different random seeds. Overall, we observe moderate but not catastrophic degradation: PC scores are most sensitive to occlusion (largest drop of 0.38), while SA scores are most affected by additive noise (largest drop of 0.31), and both tasks are comparatively robust to blur and compression. Note that these experiments are conducted on T2V-generated clips with synthetic distortions rather than real-world, in-the-wild video artifacts, so extending Cosmos-Eval to broader real-world noise conditions remains an interesting direction for future work.

## G.11 LONG-HORIZON EVALUATION ON LONGCAT-VIDEO

To probe long-horizon behavior, we evaluate Cosmos-Eval and VideoPhy-2-AutoEval on videos of length $\approx 33$ seconds generated from 30 prompts using a LongCat-style T2V setup. Each video is evaluated (i) as a full clip, and (ii) as 11 consecutive non-overlapping 3-second segments. We report mean SA/PC scores across segments and for the full video.

Table 33 summarizes segment-wise ranges and full-clip averages. Both evaluators exhibit stable SA/PC scores across time, and Cosmos-Eval behaves comparably to VideoPhy-2-AutoEval on long multi-step sequences.

## H    REPRODUCIBILITY STATEMENT

All information needed to replicate our results is provided in Appx. B (Figs. 3–5, Alg. 1, Table 6) and the main text (Eqs. 4, 6). All datasets used are publicly available and can be downloaded from their official websites (*VideoPhy* and *VideoPhy-2*; see (Bansal et al., 2025a;b)). We detail the complete prompt flow and provide all prompts in Appx. J. Model versions and full decoding hyperparameters (temperature, top-$p$, max tokens) are specified. Because inference relies on sampling, we do not fix random seeds; minor run-to-run variance is expected, but the stated configurations suffice for independent replication of the main results. Upon acceptance, we will publicly release all code, scripts, and model weights to facilitate exact reproduction.

## I    THE USE OF LARGE LANGUAGE MODELS (LLMS)

We used large language models only for light editorial assistance during manuscript preparation (grammar and wording refinement, minor style/formatting suggestions). No LLMs were used for research ideation, dataset curation, modeling, experiment design, analysis, or drafting substantive sections.

## J    PROMPT TEMPLATES

This section briefly documents the prompt flow used in Stages 1–2; figures referenced below are already included in the paper.

- **SA, Stage 1.** From the *rationale prompt* (Fig. 25) to the *consensus prompt* (Fig. 26), which aggregates two rationales into the SA reference $r_{\mathrm{ref}}^{\mathrm{sa}}$.
- **PC, Stage 1.** From the *candidate-generation prompt* (Fig. 27) to the *explanation-selection prompt* used by the judge (Fig. 28) to obtain $r_{\mathrm{ref}}^{\mathrm{pc}}$.
- **SA, Stage 2.** From the *seed-ref prompt* (Fig. 29) to the *assessment prompt* (Fig. 36) that produces a concise evidence-based justification.
- **PC, Stage 2.** From the *seed-ref prompt* (Fig. 37) to the *assessment prompt* (Fig. 44) under the PC rubric.
- **Unified CoT narration.** The accepted structured analysis from Stage 2 is converted into a natural, first-person narration using the *NaturalReasoning* prompt (Fig. 45).
- **Ablations (SA/PC).** From the *DeepSeek-R1 remapping prompt* (Fig. 46) to the *Qwen-VL-Max reason-evaluation prompt* (Fig. 49).

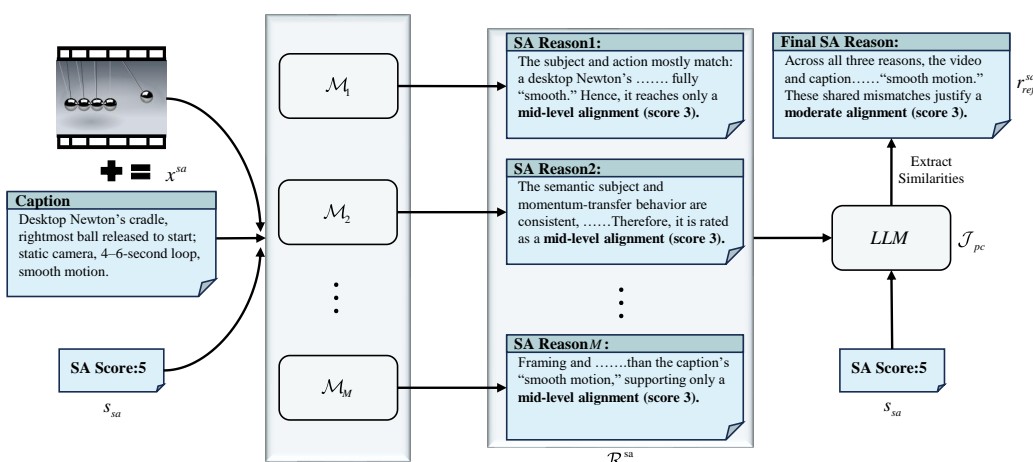

Figure 3: **Stage 1 (SA) reason generation (ensemble ⇒ consensus).** An ensemble $\{\mathcal{M}_m\}_{m=1}^M$ produces one reason each, forming the pool $\mathcal{R}_{\text{pool}}^{\text{sa}}$ (Eq. 3); an aggregator LLM then extracts shared content to yield the reference reason $r_{\text{ref}}^{\text{sa}}$ (Eq. 4), which seeds Stage 2.

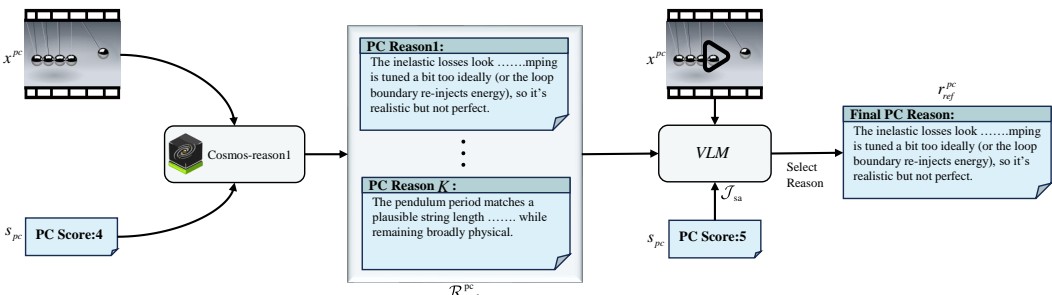

Figure 4: **Stage 1 (PC) reason generation (sampling ⇒ selection).** The base VLM $\mathcal{M}_{\text{base}}$ samples $K$ candidate reasons to form the pool $\mathcal{R}_{\text{pool}}^{\text{pc}}$ (Eq. 5); an VLM judge $\mathcal{J}_{\text{pc}}$ then selects the reference rationale $r_{\text{ref}}^{\text{pc}}$ (Eq. 6), which seeds Stage 2.

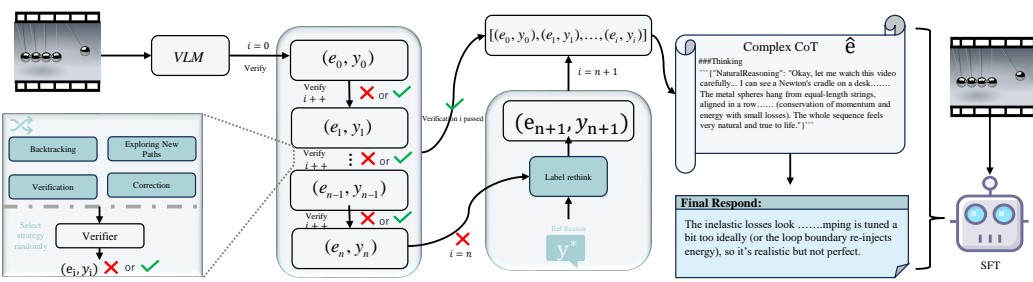

Figure 5: **Stage 2 (reason-augmented CoT).** Starting from the reference reason $r_{\text{ref}}^\tau$ (from Stage 1), a judge-verified controller iteratively explores, verifies, and corrects without exposing the reference mid-trajectory; each candidate $(e_i^\tau, r_i^\tau)$ is checked by $\mathcal{V}_\tau$ for pass or fail (Eqs. equation 9, equation 12). The controller uses the strategy set $\mathcal{C}$ (Backtracking, Exploring New Paths, Verification, Correction); if none pass, LabelRethink re-injects the reference (Eq. equation 13), and the accepted history is reformatted into $(\hat{e}^\tau, \hat{r}^\tau)$ (Eq. equation 16).

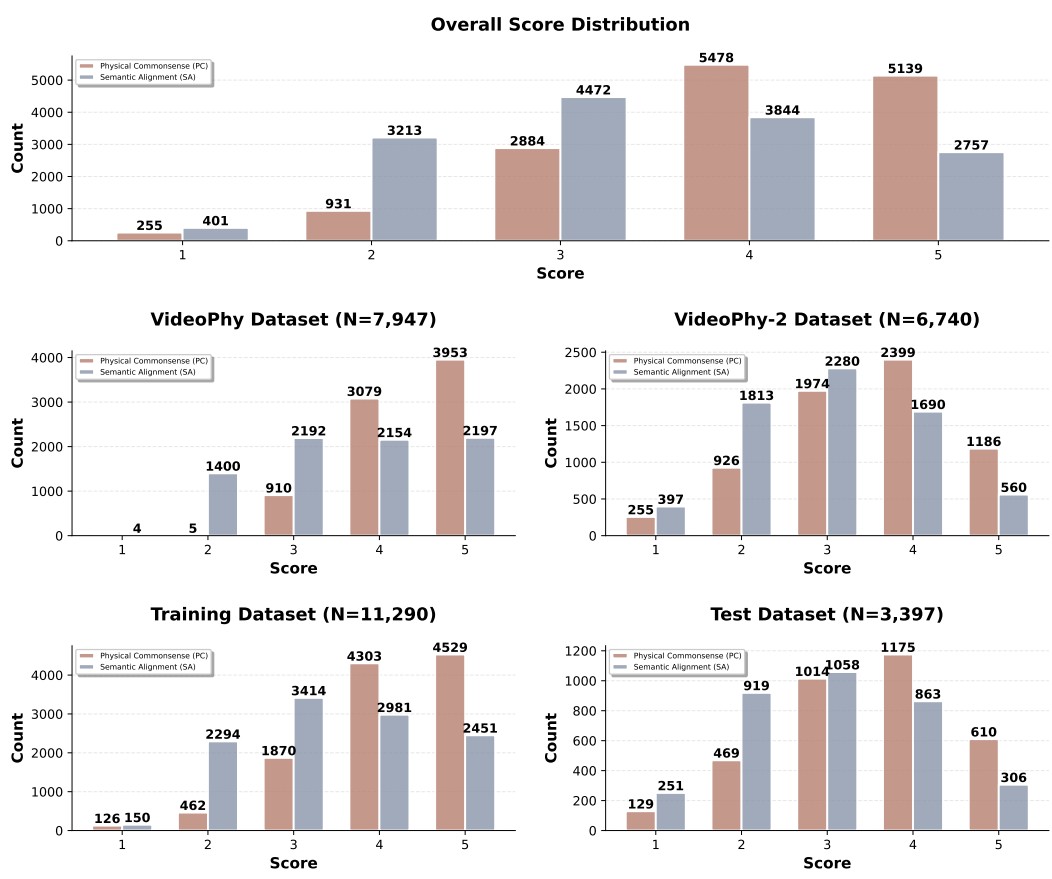

Figure 6: **Score distributions for SA and PC.** Left to right: (i) overall (train+test), (ii) *VideoPhy* subset (SA/PC scores obtained via VideoPhy-2 AutoEval), (iii) *VideoPhy-2* subset, and the final (iv) training and (v) test splits. SA is skewed toward higher scores (4–5), whereas PC concentrates on 3–4 with fewer 5's and more 2's than SA. On the *VideoPhy* portion, SA is sharply peaked at 4 (almost no 5's), while PC is roughly balanced across 2–4; *VideoPhy-2* shows a broader SA spread (nontrivial 1/5 tails) and a PC peak at 3 with a secondary mode at 4. Train/test distributions are similar, with the test split slightly flatter. These imbalances motivate reporting $\kappa$ alongside accuracy/correlation and using stratified sampling in SFT.

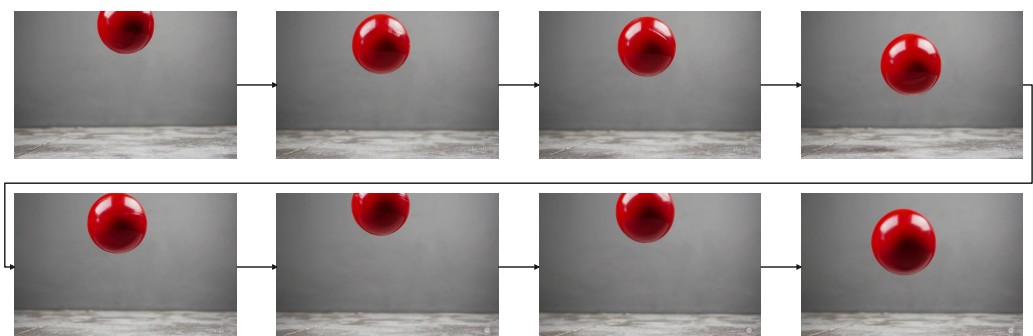

Figure 7: **Case 1 — Physical Commonsense (PC). Input caption:** *N/A (no caption provided).* **Answer (Cosmos-Eval, verbatim):** The video demonstrates a red ball hovering mid-air without any visible support, defying gravitational expectations. The physical commonsense is poor because the ball's static suspension violates basic principles of force and motion, despite maintaining a realistic appearance otherwise. This justifies a pc_score of 2 due to the significant inconsistency with gravitational effects while other visual elements remain accurate. **PC score: 2**.

---

**Algorithm 1:** Stage-2 Reference-Seeded, Judge-Verified Controller for task $\tau$

---

**Input:** $x^\tau$; prompts $\mathbf{P}^\tau_{\text{seed-ref}}, \{\mathbf{P}^\tau_c\}_{c\in\mathcal{C}}, \mathbf{P}^\tau_{\text{rethink}}$; judge prompt $\mathbf{U}^\tau$; reference $r^\tau_{\text{ref}}$; budget $N$
**Output:** $(\hat{e}^\tau, \hat{r}^\tau)$ or $\varnothing$

$\mathcal{H}^\tau \leftarrow \varnothing; \quad i^\star \leftarrow \text{nil};$
$\text{Avail} \leftarrow \mathcal{C}; \quad T \leftarrow \min(N, |\mathcal{C}|);$
$(e^\tau_0, r^\tau_0) \leftarrow \mathcal{M}(\mathbf{P}^\tau_{\text{seed-ref}}, x^\tau, r^\tau_{\text{ref}}; \texttt{Reason});$
$\mathcal{H}^\tau \leftarrow \mathcal{H}^\tau \cup \{(e^\tau_0, r^\tau_0)\};$
$pass \leftarrow \mathcal{V}_\tau(r^\tau_0, r^\tau_{\text{ref}}; \mathbf{U}^\tau);$
**if** $pass = 1$ **then**
    $i^\star \leftarrow 0;$
**else**
    **for** $i \leftarrow 1$ **to** $T$ **do**
        pick $c_i \in \text{Avail}$ uniformly;    $\text{Avail} \leftarrow \text{Avail} \setminus \{c_i\};$
        $(e^\tau_i, r^\tau_i) \leftarrow \mathcal{M}(\mathbf{P}^\tau_{c_i}, x^\tau, \mathcal{H}^\tau; c_i);$
        $\mathcal{H}^\tau \leftarrow \mathcal{H}^\tau \cup \{(e^\tau_i, r^\tau_i)\};$
        $pass \leftarrow \mathcal{V}_\tau(r^\tau_i, r^\tau_{\text{ref}}; \mathbf{U}^\tau);$
        **if** $pass = 1$ **then**
            $i^\star \leftarrow i;;$
            **break**
    **if** $pass = 0$ **then**
        $(e^\tau_{T+1}, r^\tau_{T+1}) \leftarrow \mathcal{M}(\mathbf{P}^\tau_{\text{rethink}}, x^\tau, r^\tau_{\text{ref}}, \mathcal{H}^\tau; \texttt{LabelRethink});$
        $\mathcal{H}^\tau \leftarrow \mathcal{H}^\tau \cup \{(e^\tau_{T+1}, r^\tau_{T+1})\};$
        $pass \leftarrow \mathcal{V}_\tau(r^\tau_{T+1}, r^\tau_{\text{ref}}; \mathbf{U}^\tau);$
        **if** $pass = 1$ **then**
            $i^\star \leftarrow T+1;$
**if** $pass = 1$ **then**
    $\hat{e}^\tau \leftarrow \text{PostChain}\big(\{(e^\tau_j, r^\tau_j)\}^{i^\star}_{j=0}; \texttt{SynthesizeChain}\big);$
    $\hat{r}^\tau \leftarrow \text{PostAnswer}(\hat{e}^\tau, r^\tau_{\text{ref}}; \texttt{Reformat});$
    **return** $(\hat{e}^\tau, \hat{r}^\tau);$
**else**
    **return** $\varnothing;$

---

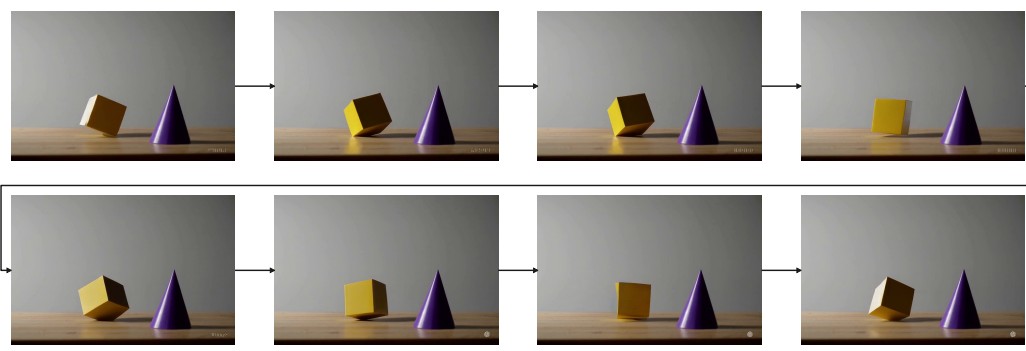

Figure 8: **Case 2 — Semantic Alignment (SA). Generation caption:** *A yellow cube rotates counterclockwise on the table while a purple cone remains still.* **Answer (Cosmos-Eval, verbatim):** The video demonstrates the yellow cube rotating while the purple cone remains stationary, with the former exhibiting a clockwise rotation contrary to the caption's description of counterclockwise movement. The alignment is moderate because while the scene and object placement are accurate, the direction of rotation significantly deviates from the stated caption. This justifies an sa_score of 3 due to the primary focus on object interaction with only a minor discrepancy in rotational direction. **SA score (assigned): 2**.

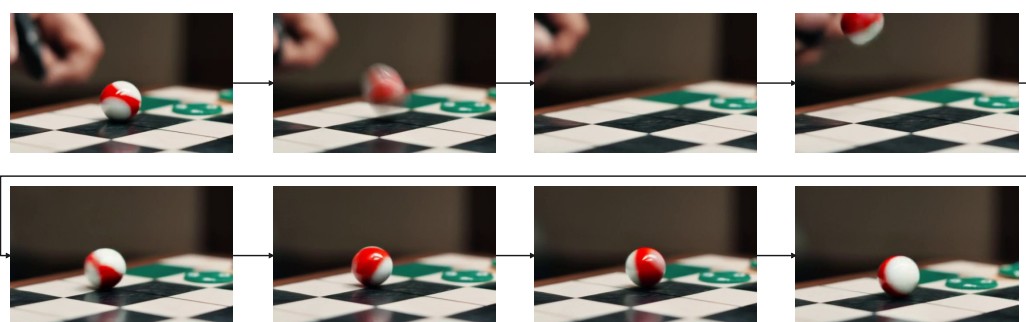

Figure 9: **Case 3 — Physical Commonsense (PC). Input caption:** *N/A (no caption provided).*
**Answer (Cosmos-Eval, verbatim):** The video demonstrates a ball rolling on a checkerboard surface
with inconsistent motion, exhibiting erratic bouncing and lack of frictional decay, which violates
principles of Newtonian mechanics. The physical commonsense is poor because the ball's behavior
cannot be explained by external forces alone, suggesting either an artificial manipulation or a
misunderstanding of physical laws. This justifies a pc_score of 2 due to significant deviations from
expected physical interactions and dynamics. **PC score: 2**.

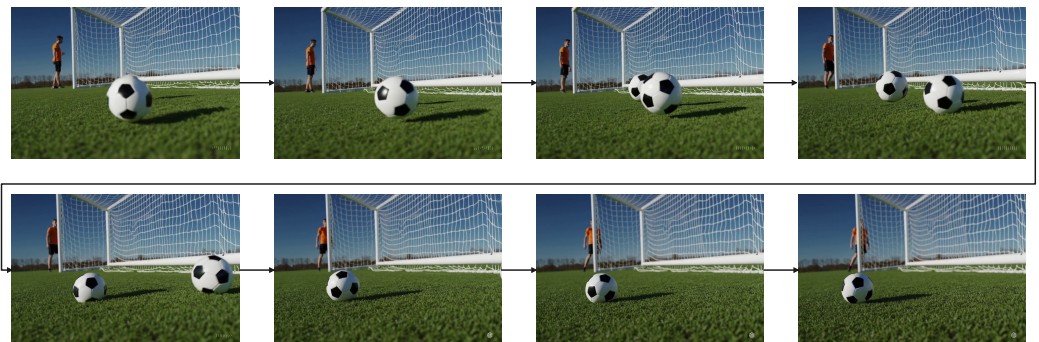

Figure 10: **Case 4 — Semantic Alignment (SA). Generation caption:** *A man kicks a soccer ball; it
strikes the goalpost and rebounds.* **Answer (Cosmos-Eval, verbatim):** The video shows a man
observing two soccer balls approaching the goalpost, whereas the caption describes a single ball
being kicked towards the goalpost. The alignment is moderate because although the setting
accurately represents a soccer field, the presence of multiple balls and the absence of a kicking action
introduce significant discrepancies. This justifies a sa_score of 3 due to the partial adherence to the
scene description despite notable differences in the number of balls and the absence of the critical
kicking action. **SA score: 3**.

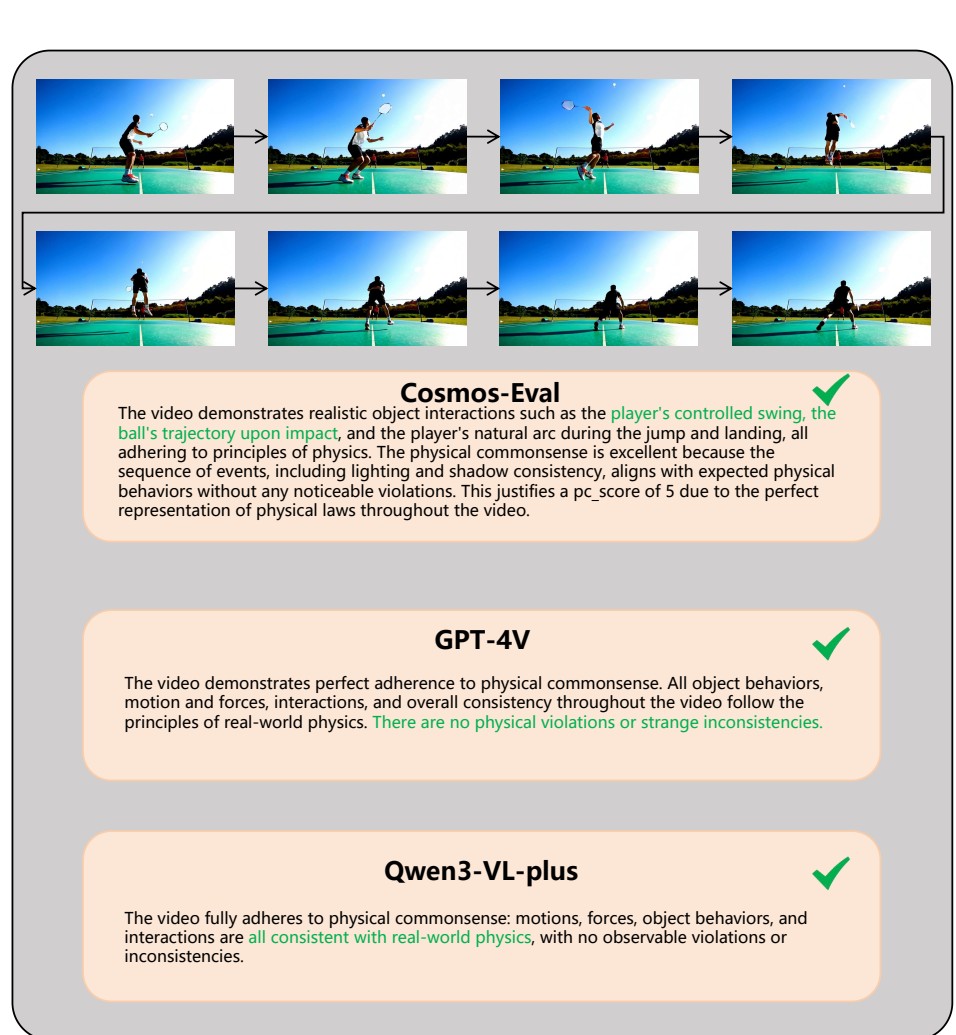

Figure 11: Example of Physical Commonsense Task 1

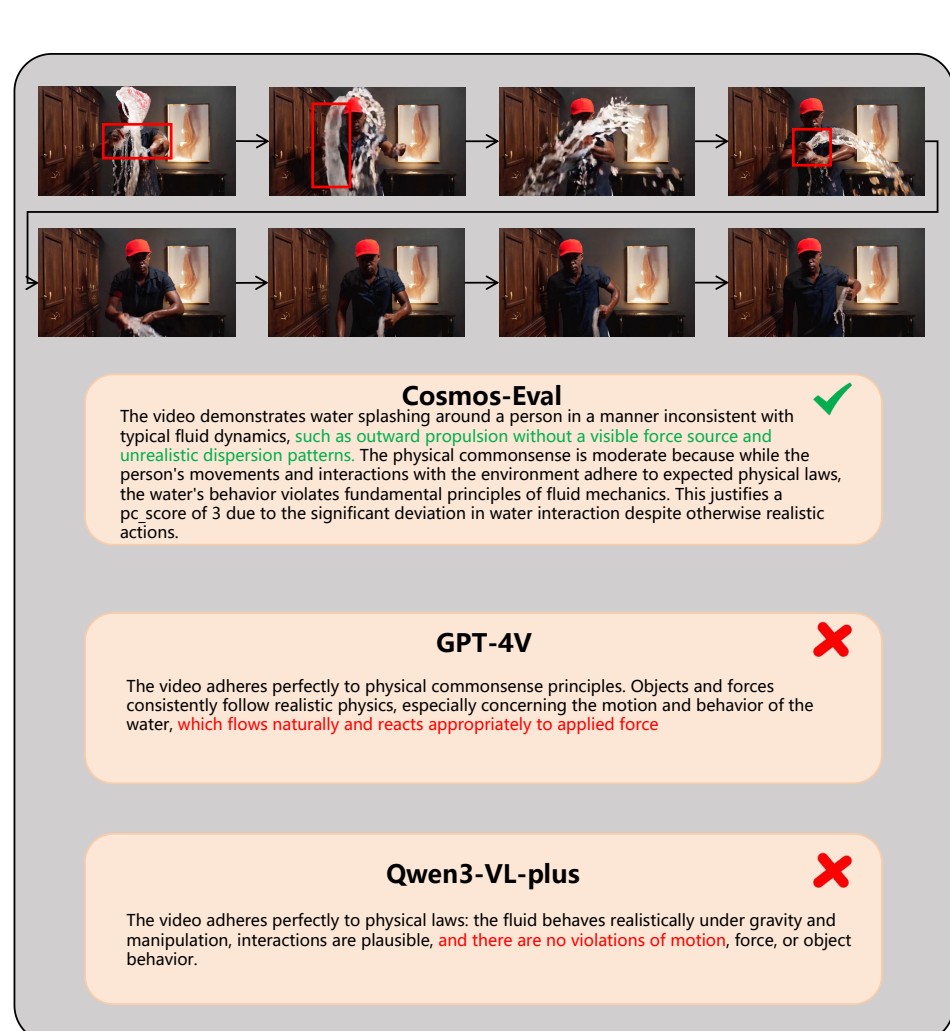

Figure 12: Example of Physical Commonsense Task 2

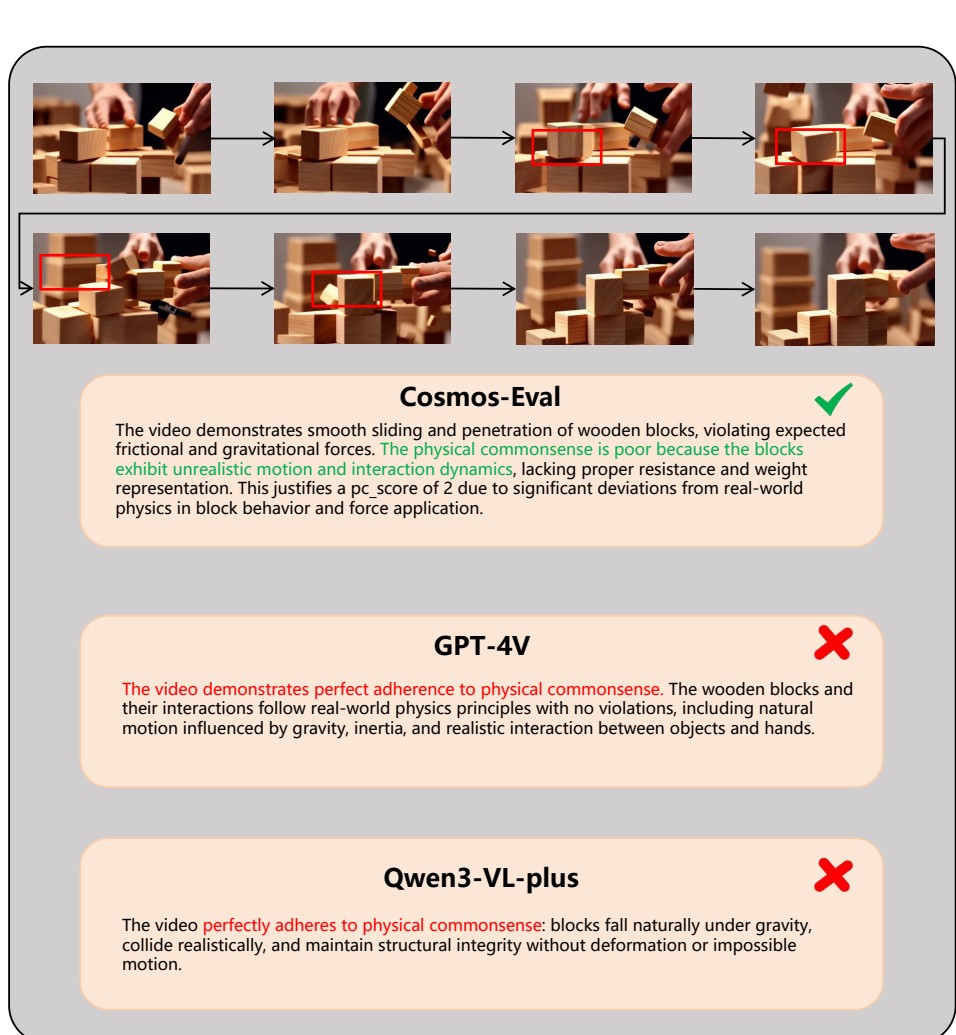

Figure 13: Example of Physical Commonsense Task 3

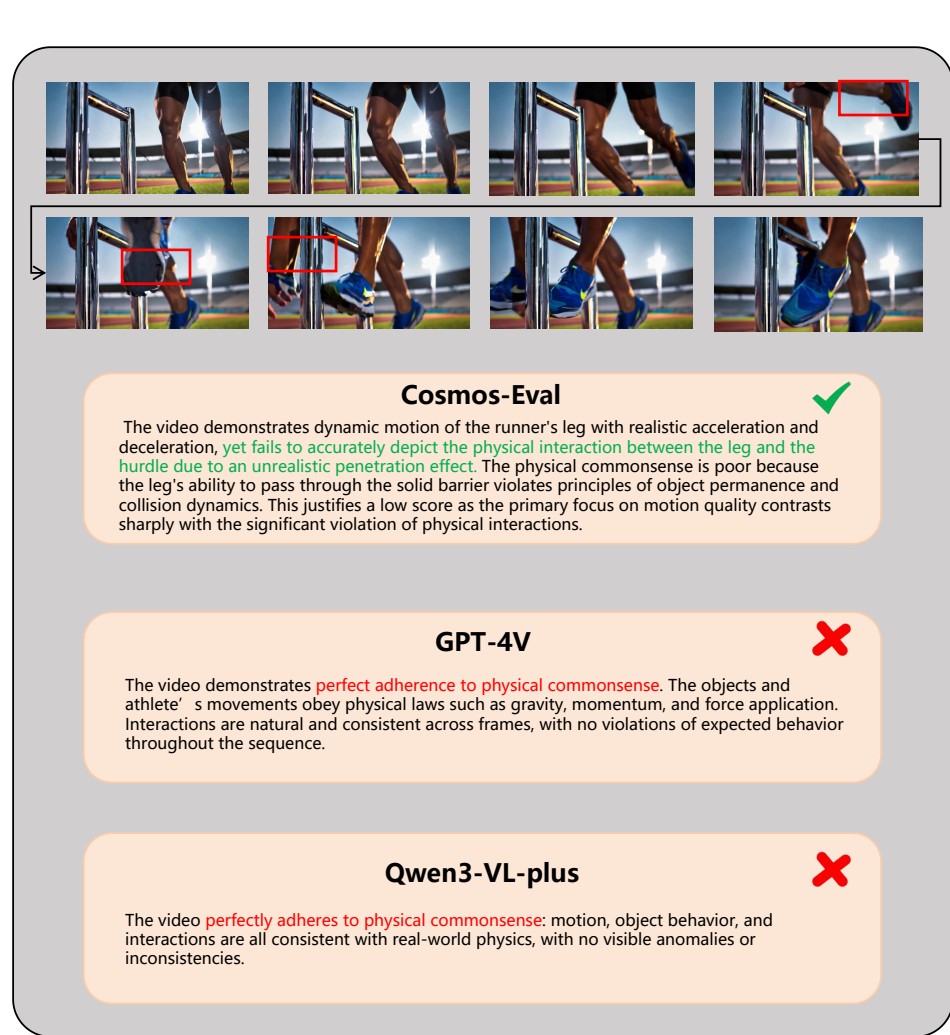

Figure 14: Example of Physical Commonsense Task 4

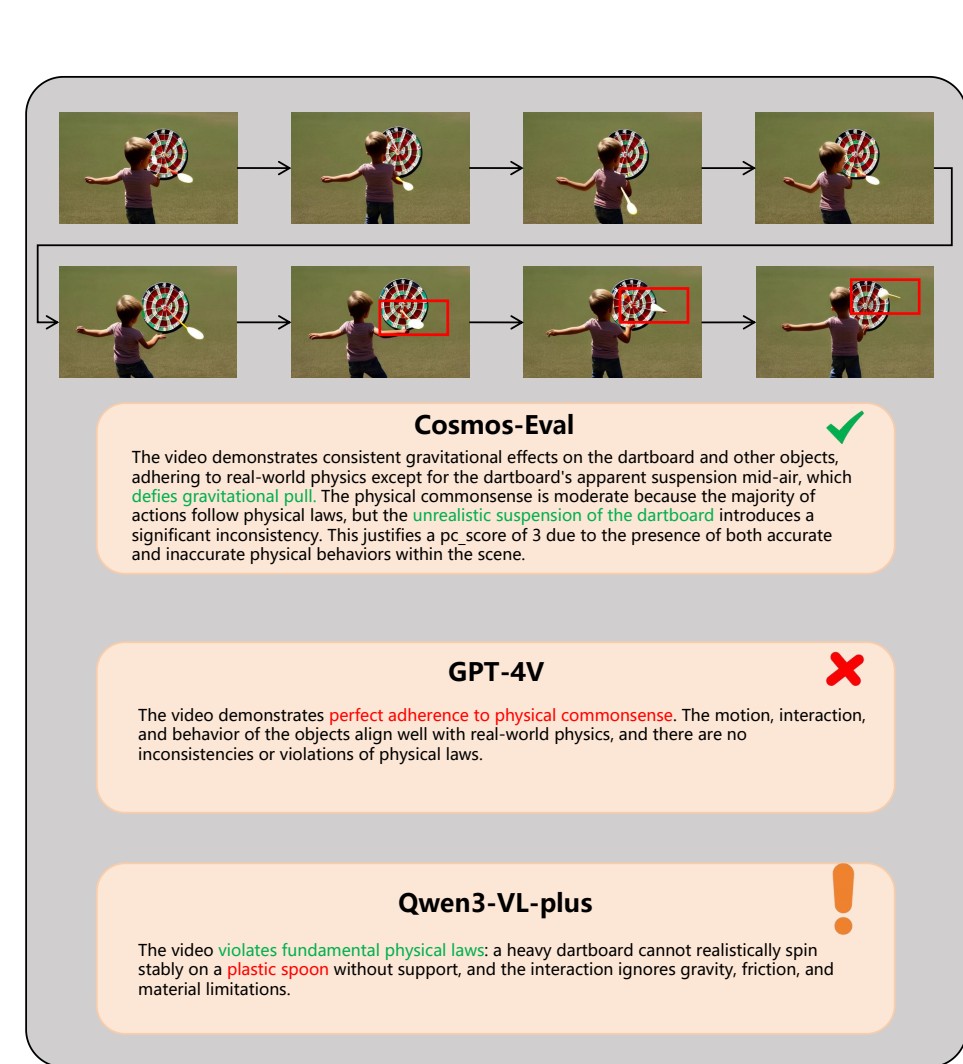

Figure 15: Example of Physical Commonsense Task 5

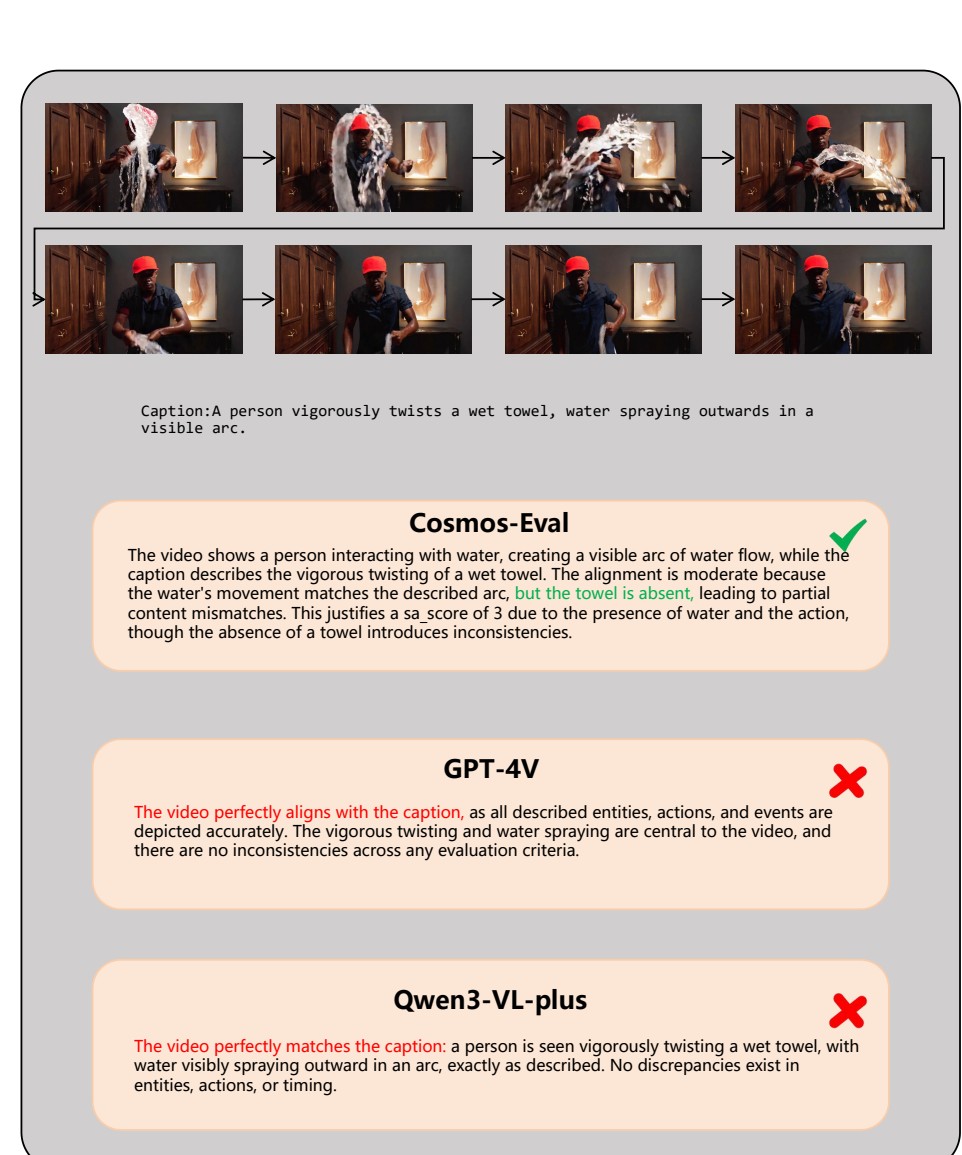

Figure 16: Example of Semantic Alignment Task 1

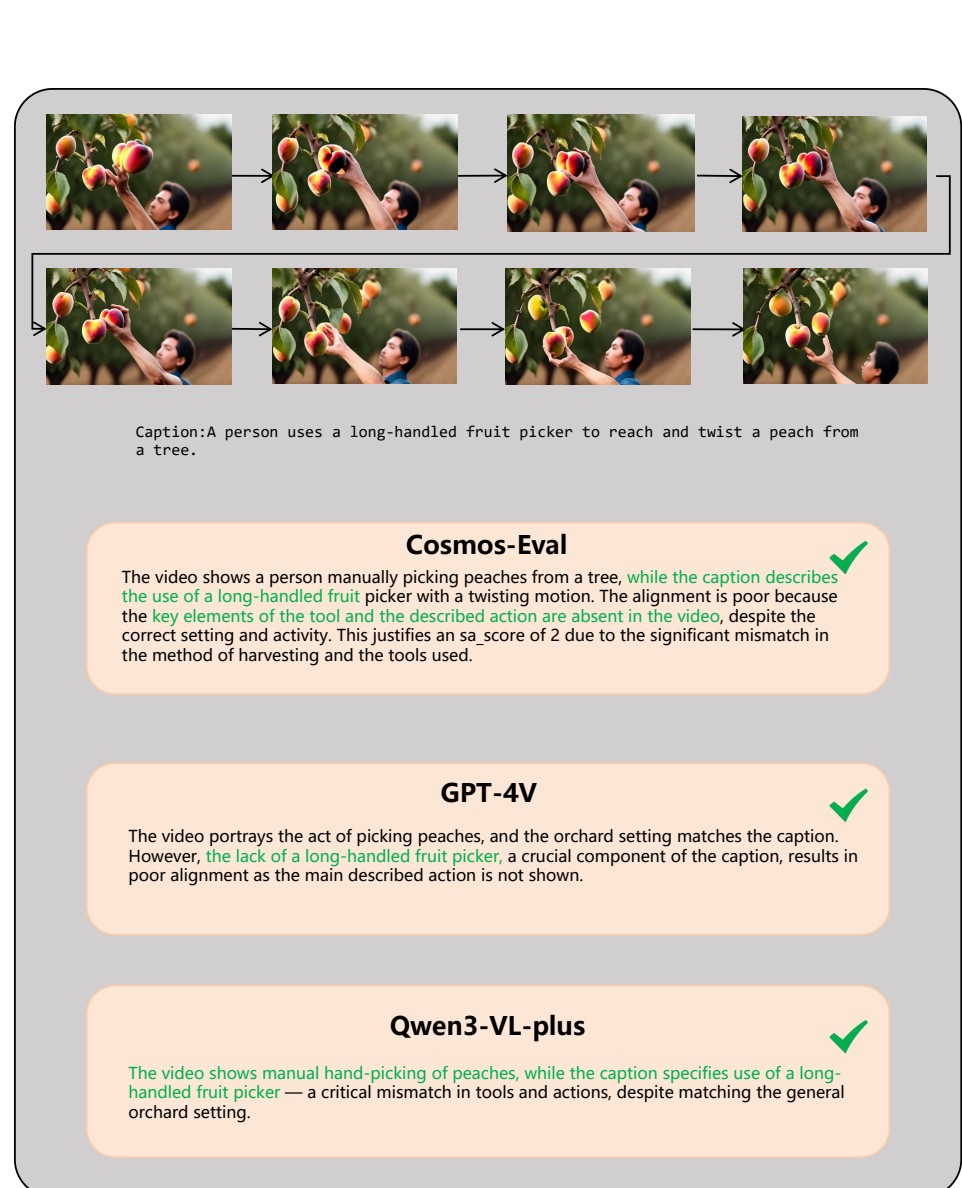

Figure 17: Example of Semantic Alignment Task 2

Figure 18: Example of Semantic Alignment Task 3

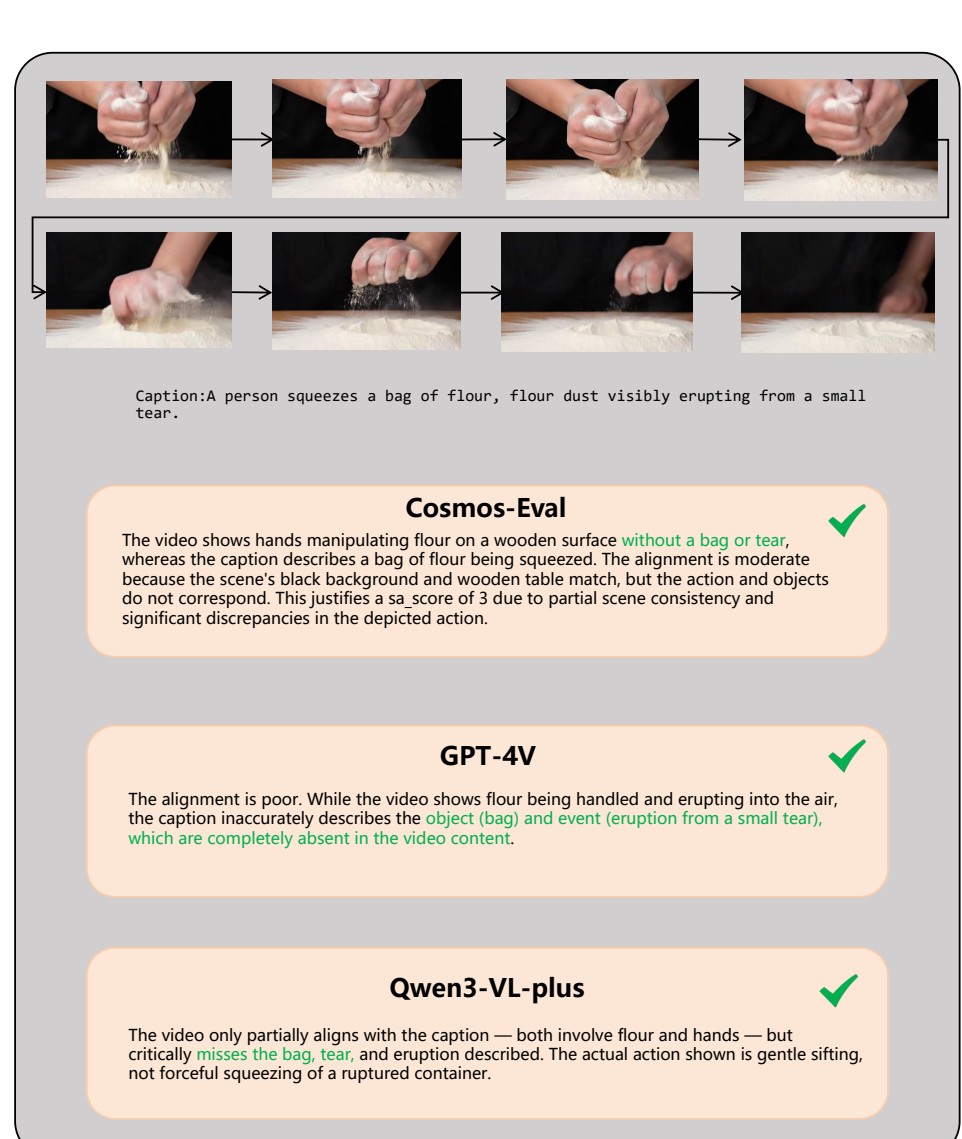

Figure 19: Example of Semantic Alignment Task 4

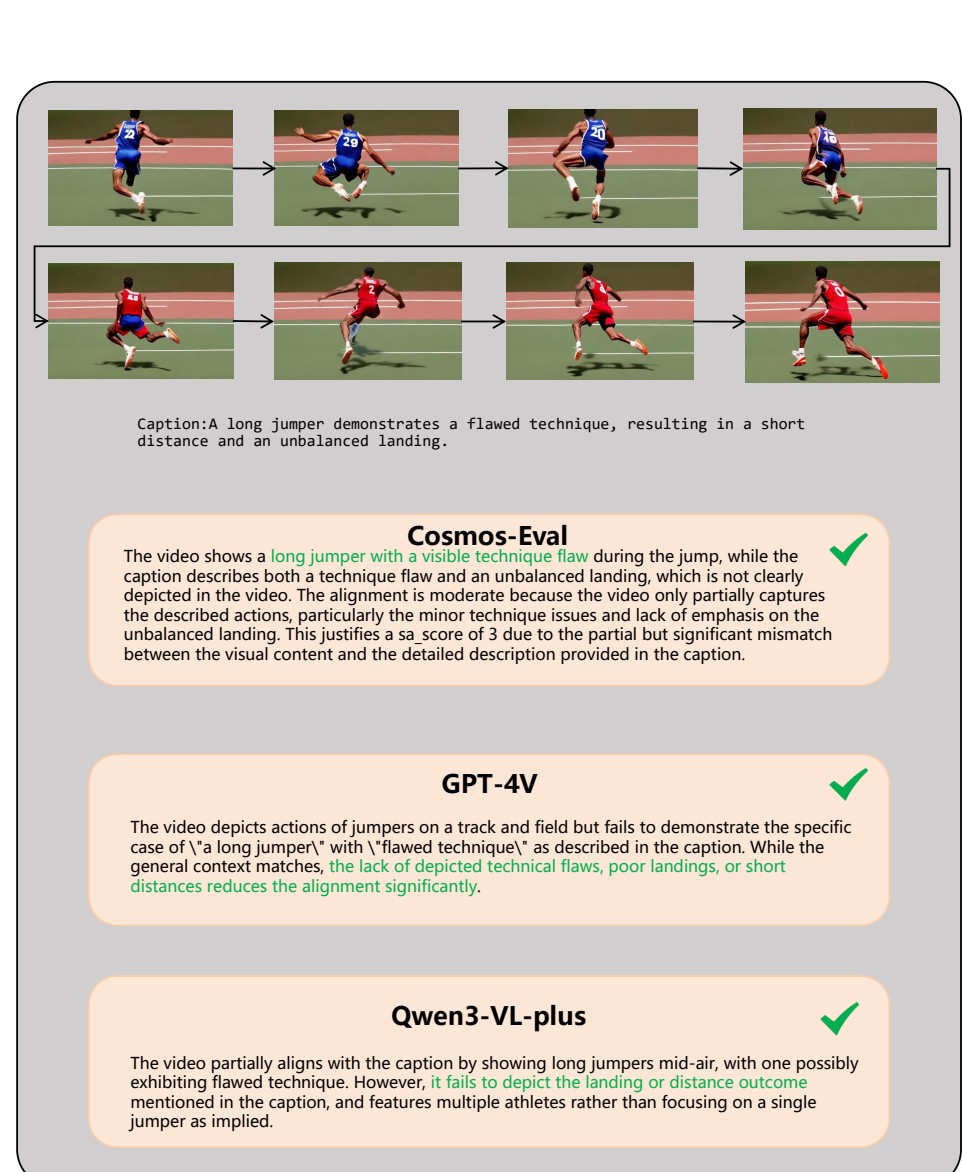

Figure 20: Example of Semantic Alignment Task 5

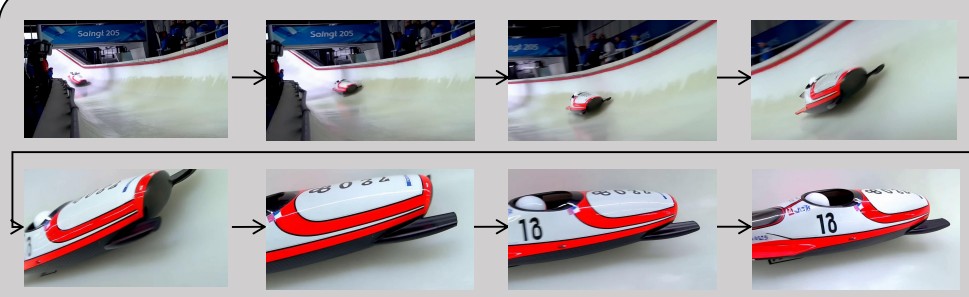

**Cosmos-Eval**

<think>\nOkay, let me watch this video carefully... I notice this sled moving down the track, and it looks like it's gliding really smoothly. The way it's moving, it seems to be following the curves of the track nicely. Wait, let me look closer at how it interacts with the surface. Hmm, this seems to be maintaining a consistent speed, which makes sense because of the smoothness of the track. Actually, I don't see any kind of sudden acceleration or deceleration, so the forces must be balanced well.\n\nI can see the lighting is pretty bright, making everything clear. The sled is maintaining its shape as it moves along, which tells me it's a rigid object as it should be. Oh wait, I should also pay attention to how it's interacting with the video around it. Yes, that confirms there are no weird penetrations or anything like that. But then again, I need to think about the track itself. It looks like it's made of some kind of smooth material, maybe ice, and the sled is sliding over it without any issues.\n\nLet me think about this for a moment. The sled's interaction with the track and its smooth motion through it are really convincing. This looks like a well-thought-out design. Actually, now that I'm thinking about it, the sled's design probably plays a big role in its ability to slide so smoothly. It's streamlined, which helps minimize air resistance, allowing it to glide faster.\n\nHmm, this seems consistent throughout the video. The sled keeps behaving the same way across the frames, without any abrupt changes. I'm really not seeing anything that would make me doubt the physics here. Yes, that confirms my earlier thoughts. The sled continues to interact with the track in a plausible manner, maintaining its consistent speed and shape.\n\nSo, putting all of this together, the video seems to adhere to physical commonsense pretty well. The sled's behavior, motion, and interactions all seem to be following the expected physical laws. It's smooth, consistent, and the design elements support that. I don't see any violations or inconsistencies. Therefore, I'm going to give this a high score, probably a 4. It's good, but maybe there could be a tiny bit more detail or complexity, but overall it's very convincing.\n</think>\n\n<answer>\nThe video demonstrates smooth sliding motion of the sled along a curved track, with consistent speed and lack of penetration, adhering closely to expected physical behavior under gravitational and aerodynamic influences. The physical commonsense is good because the sled's interaction with the track maintains realistic dynamics, with no observed violations of physical laws. This justifies a pc_score of 4 due to the high level of adherence to physical expectations with minimal potential for refinement.\n</answer>

Figure 21: Example of Physical Commonsense Task with CoT (Case 1)

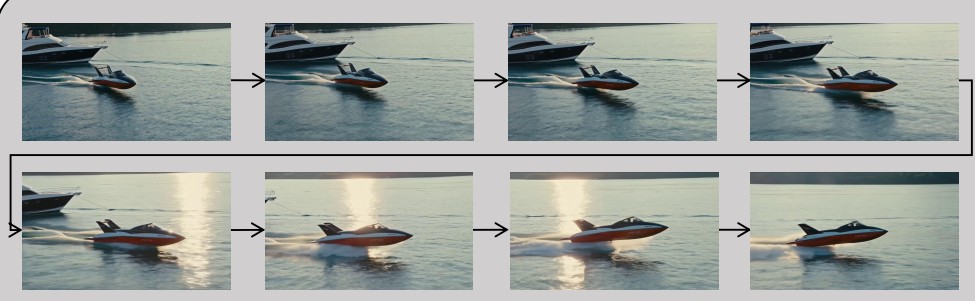

**Cosmos-Eval**

<think>\nOkay, let me watch this video carefully... I can see a speedboat gliding across the water. The lighting is quite beautiful, reflecting off the water's surface. The boat looks sleek, and it seems to be moving pretty fast. I notice it's maintaining a steady path, which is good. The wake behind it is smooth, suggesting the propulsion is strong and efficient. Wait, let me look closer at the water interaction. The ripples are forming just right around the boat, no weird splashes or anything that would defy physics. This looks like a normal speedboat ride.\n\nHmm, this seems right so far. Let's consider the motion. The boat's movement is smooth, indicating a continuous force from the motor. There's no sudden jerking or stopping, which is what you'd expect when the motor is functioning properly. Oh wait, I should check if there are any inconsistencies in the wake patterns or the way the boat interacts with the water. Actually, everything looks consistent with real-world physics here.\n\nNow, focusing on how the boat interacts with the water, it's creating that nice trail behind it. It doesn't sink or float weirdly, which is important for its buoyancy. Yes, that confirms the boat is behaving as expected. But then again, I should ensure there's nothing unusual happening with the cable. The cable is taut and straight, suggesting it's pulling the boat without any slack or slackness. This looks like a controlled environment.\n\nAs I keep watching, the video maintains a consistent state throughout. The boat keeps moving at a constant speed, and the lighting remains the same, indicating no abrupt changes in the scene. This looks like a perfectly seamless continuation of the action. I'm trying to find any anomalies, but really, everything feels right.\n\nSo, after all this, the video seems to be adhering well to physical commonsense. The boat's behavior is consistent with real-world physics, from its smooth motion and interaction with the water to the wake patterns and cable tension. Yes, that confirms my initial impression. There are no major violations or inconsistencies, and it feels very natural. The pc_score of 4 fits perfectly here, showing good adherence to physical laws with only minor issues that don't detract from the overall realism.\n</think>\n\n<answer>\nThe video demonstrates consistent hydrodynamic interactions with a smooth wake trail and a taut cable, adhering closely to real-world physics. The physical commonsense is good because the boat's motion, wake patterns, and buoyancy exhibit no major violations of physical laws, maintaining a realistic depiction of fluid dynamics and propulsion forces. This justifies a pc_score of 4 due to the high level of adherence to physical principles with only minor potential issues.\n</answer>

Figure 22: Example of Physical Commonsense Task with CoT (Case 2)

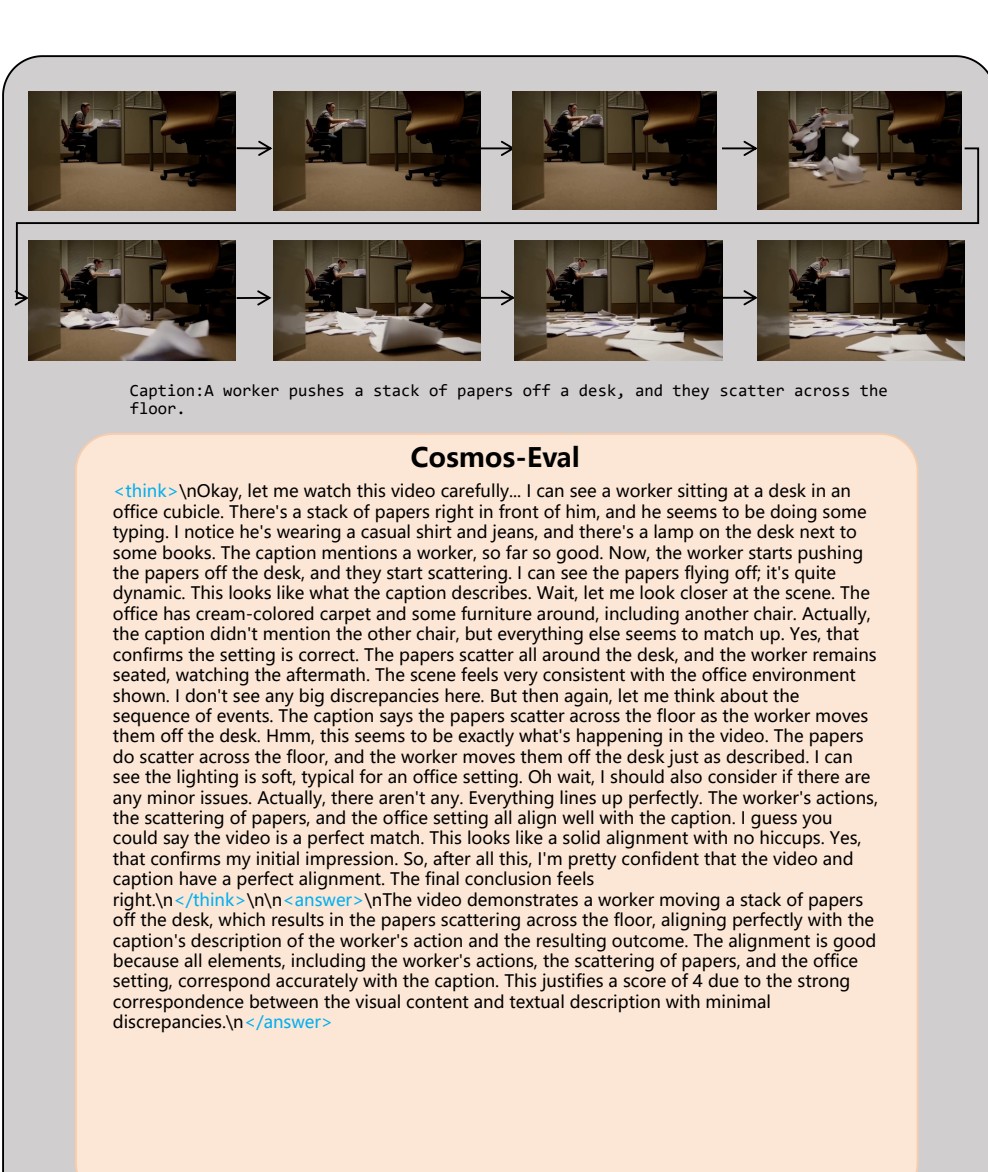

Figure 23: Example of Semantic Alignment Task with CoT (Case 1)

Caption:Someone pulls apart a tightly wound rubber band that separates with audible snap.

## Cosmos-Eval

<think>\nOkay, let me watch this video carefully... I can see a pair of hands holding a red, coiled object that looks like a toy or maybe a stress ball. The caption mentions a 'rubber band', and at first glance, this red object seems flexible, like a rubber band might be. But wait, let me look closer... Actually, it's more than just a band; it's coiled tightly and has a distinct shape. Hmm, this seems like it could be some kind of toy rather than just a simple rubber band. I notice the hands are pulling it apart, and it starts to uncoil. The caption says it 'separates with an audible snap', but I don't hear any snapping sound here. Oh wait, maybe it's just not captured in the video? Or perhaps the action is subtle. The hands are moving, and the object is stretching and twisting. This looks like a slow, deliberate motion, not the quick snap implied by the caption. But then again, the scene is quite plain, almost like a studio setup for focusing on the object. The lighting is even, and the background is neutral, which fits well with highlighting the red toy against it. Yes, that confirms the scene is simple and focused. The overall action is continuous, showing the object being stretched and manipulated without any jumps or cuts. This looks like it's showing the transformation of the toy over time, which aligns with the caption suggesting a transformation. So, putting it all together, the video partially matches the caption. It shows the hands manipulating the toy, but it doesn't capture the exact nature of the transformation or the separation as described. The scene is simple, and the motion is smooth, which supports a good alignment. I think the score should reflect this as a moderate alignment with some discrepancies, like the lack of snapping sound and the focus on transformation rather than separation. Overall, it's a decent match, but there are still noticeable differences that prevent it from being perfect.\n</think>\n\n<answer>\nThe video shows hands manipulating a coiled, red flexible object, while the caption describes a 'rubber band' that separates with an audible snap. The alignment is moderate because the video demonstrates continuous manipulation without revealing the snapping sound or the separation process, indicating a transformation rather than a simple tear. This justifies a sa_score of 3 due to the depiction of a gradual action and the absence of the described auditory and separation events.\n</answer>

Figure 24: Example of Semantic Alignment Task with CoT (Case 2)

---

**Title :Stage 1 (SA) rationale prompt — ensemble**

You are a video-text alignment expert. Analyze the alignment between video content and text descriptions.
Your response must be a valid JSON object with exactly these two fields:
{
    "score_explanation": "Based on X-point scoring basis: [explanation]",
    "mismatches": ["Caption requires ... but video shows ...", "Caption specifies ... but video shows ..."]
}

Scoring Basis:
sa points: {5=Perfect alignment | 4=Minor deviations | 3=Partial match | 2=Key omissions | 1=Completely unrelated}

Analysis Dimensions:
1. Entity presence: Objects mentioned in caption
2. Action accuracy: Faithfulness to described actions
3. Temporal order: Sequence consistency
4. Scene context: Environmental match

### Requirements:
1. Identify key alignment issues
2. Use contrastive phrasing (Caption requires... but video shows...)
3. Use specific, concise language
Explain why this video received sa={sa} score based on caption: " {caption}"

Figure 25: **Stage 1 (SA) prompt.** The SA score $s_{\mathrm{SA}}$ used in this prompt is provided by Eq. 1. This prompt forms the ensemble pool in Eq. 3; placeholders {sa} and {caption} are highlighted in blue for clarity.

2700
2701
2702
2703
2704
2705
2706
2707
2708
2709
2710
2711
2712
2713
2714
2715
2716
2717
2718
2719
2720
2721
2722
2723
2724
2725
2726
2727
2728
2729
2730
2731
2732
2733
2734
2735
2736
2737
2738
2739
2740
2741
2742
2743
2744
2745
2746
2747
2748
2749
2750
2751
2752
2753

---

### Title:Stage 1 (SA) consensus prompt — aggregator ($M = 2$)

You are a video-text alignment evaluation expert. Given two semantic alignment (SA) analyses of the same video-caption pair, use chain-of-thought reasoning to extract ONLY the error points that are SEMANTICALLY IDENTICAL in both analyses.

### Input Analysis:
**Analysis 1:**
{sa_reason1_str}

**Analysis 2:**
{sa_reason2_str}

### Reasoning Steps (Execute Strictly):
1. **Semantic Parsing**: Extract core claims and negation relationships from each analysis
2. **Proposition Decomposition**: Break each analysis into minimal verifiable proposition units
3. **Bidirectional Entailment Check**: For each proposition unit, verify:
   a) Analysis 1 entails this proposition in Analysis 2
   b) Analysis 2 entails this proposition in Analysis 1
4. **Common Proposition Filtering**: Retain only propositions that pass bidirectional entailment
5. **Evidence Fusion**: Integrate video evidence supporting common propositions from both analyses
6. **Contradiction Detection**: Check for any logical contradictions

### Output Requirements:
1. **Strict Commonality**:
   - Include ONLY semantically identical parts from both analyses
   - Use neutral video evidence: "The video shows..." NOT "Analysis1 states..."

2. **Output Format**:
   {{
      "sa_reason": "Coherent paragraph describing common errors",
      "error": "Specific contradiction reason OR empty string"
   }}

3. **Contradiction Handling Rules**:
   - Return error ONLY for logical conflicts (e.g., A claims X exists, B claims X doesn't exist)
   - Expression differences with same semantics are NOT contradictions
   - Automatic error when either analysis is empty

### Special Case Guidance (Your Bottle Example):
Input Example:
  Analysis1: "caption states the bottle will wobble and fall but video shows no wobbling or falling"
  Analysis2: "caption states the bottle will wobble and fall but video is static"
Correct Output:
  sa_reason: "The caption claims the bottle wobbles and falls, but the video shows no such dynamic process"
  error: ""

Now process the following analyses using this reasoning:

---

Figure 26: **Stage 1 (SA) consensus prompt** ($M{=}2$). This template aggregates two SA rationales into the consensus $r_{\text{ref}}^{\text{sa}}$ as defined in Eq. 4. The SA score $s_{\text{SA}}$ used upstream is obtained from Eq. 1. Placeholders {sa_reason1_str} and {sa_reason2_str} are highlighted in blue.

---

### Title:Stage 1: PC reason generation (base, $K = 5$)

Task Description: Evaluate whether the video follows physical commonsense. This judgment is based solely on the video itself and does not depend on the caption.

Evaluation Criteria:
1. **Object Behavior:** Do objects behave according to their expected physical properties (e.g., rigid objects do not deform unnaturally, fluids flow naturally)?
2. **Motion and Forces:** Are motions and forces depicted in the video consistent with real-world physics (e.g., gravity, inertia, conservation of momentum)?
3. **Interactions:** Do objects interact with each other and their environment in a plausible manner (e.g., no unnatural penetration, appropriate reactions on impact)?
4. **Consistency Over Time:** Does the video maintain consistency across frames without abrupt, unexplainable changes in object behavior or motion?

Scoring Scale:
- **1:** No adherence to physical commonsense. The video contains numerous violations of fundamental physical laws.
- **2:** Poor adherence. Some elements follow physics, but major violations are present.
- **3:** Moderate adherence. The video follows physics for the most part but contains noticeable inconsistencies.
- **4:** Good adherence. Most elements in the video follow physical laws, with only minor issues.
- **5:** Perfect adherence. The video demonstrates a strong understanding of physical commonsense with no violations.

The video has been assigned a PC score of {pc_score} Please provide 5 different detailed explanations for this score based on what you observe in the video. Each explanation should focus on different aspects or provide different perspectives on the physical commonsense evaluation.

### Output Format:
Strictly follow the JSON structure below.

```json
{{
   "explanations": [
      {{
         "explanation_id": 1,
         "explanation": "First detailed explanation focusing on specific physical aspects that justify this score"
      }},
      {{
         "explanation_id": 2,
         "explanation": "Second detailed explanation with a different perspective or focus"
      }},
      {{
         "explanation_id": 3,
         "explanation": "Third detailed explanation highlighting different physical aspects"
      }},
      {{
         "explanation_id": 4,
         "explanation": "Fourth detailed explanation with another viewpoint"
      }},
      {{
         "explanation_id": 5,
         "explanation": "Fifth detailed explanation providing additional insights"
      }}
   ]
}}
```

Figure 27: **Stage 1 (PC) candidate-generation prompt** ($K$=5). This template queries the base VLM to produce the pool $\mathcal{R}_{\text{pool}}^{\text{pc}}$ in Eq. 5, instantiated with $K$=5 samples. The upstream PC score token $s_{\text{PC}}$ conditions the prompt; the placeholder {pc_score} is highlighted in blue.

2808
2809
2810
2811
2812
2813
2814
2815

---

**Title:Stage 1: PC explanation selection (judge, K=5)**

You are an expert in evaluating physical commonsense in videos. You have been provided with 5 different explanations for why a video received a Physical Commonsense (PC) score of {pc_score}. Your task is to select the most reasonable and accurate explanation.

**Task Description:** Evaluate whether the video follows physical commonsense. This judgment is based solely on the video itself and does not depend on the caption.

**Evaluation Criteria:**
1. **Object Behavior:** Do objects behave according to their expected physical properties (e.g., rigid objects do not deform unnaturally, fluids flow naturally)?
2. **Motion and Forces:** Are motions and forces depicted in the video consistent with real-world physics (e.g., gravity, inertia, conservation of momentum)?
3. **Interactions:** Do objects interact with each other and their environment in a plausible manner (e.g., no unnatural penetration, appropriate reactions on impact)?
4. **Consistency Over Time:** Does the video maintain consistency across frames without abrupt, unexplainable changes in object behavior or motion?

**Scoring Scale:**
- **1:** No adherence to physical commonsense. The video contains numerous violations of fundamental physical laws.
- **2:** Poor adherence. Some elements follow physics, but major violations are present.
- **3:** Moderate adherence. The video follows physics for the most part but contains noticeable inconsistencies.
- **4:** Good adherence. Most elements in the video follow physical laws, with only minor issues.
- **5:** Perfect adherence. The video demonstrates a strong understanding of physical commonsense with no violations.

**The video has been assigned a PC score of {pc_score}.**

**Generated Explanations:**
{explanations_text}

**Your Task:**
1. Watch the video carefully
2. Evaluate each explanation based on how well it matches what you observe in the video
3. Select the explanation that most accurately describes the physical aspects justifying the PC score of {pc_score}
4. Consider factors like accuracy, specificity, and relevance to the observed physics

### Output Format:
Strictly follow the JSON structure below.

```json
{{
    "selected_explanation_id": [1-5],
    "reasoning": "Your detailed reasoning for why this explanation is the best, including specific observations from the video that support your choice",
    "selected_explanation_text": "The full text of the selected explanation"
}}
```

---

Figure 28: **PC explanation selection prompt** used by the LLM judge in Eq. 6. The placeholder {explanations_text} denotes the five candidates produced by Fig. 27; {pc_score} and {explanations_text} are highlighted in blue for clarity.

---

**Title:Stage 2 (SA seed): reference-conditioned reasoning**

Analyze the alignment between a video and its corresponding caption, then explain why the given alignment score (sa_score) is appropriate.
</task>

<caption>
{caption}
</caption>

<reference_reason>
{reference_reason}
</reference_reason>

<sa_score>
{sa_score}
</sa_score>

<scoring_rules>
- **1:** No alignment. The video does not match the caption at all (e.g., different objects, events, or scene).
- **2:** Poor alignment. Only a few elements of the caption are depicted, but key objects or events are missing or incorrect.
- **3:** Moderate alignment. The video matches the caption partially, but there are inconsistencies or omissions.
- **4:** Good alignment. Most elements of the caption are depicted correctly in the video, with minor issues.
- **5:** Perfect alignment. The video fully adheres to the caption with no inconsistencies.
</scoring_rules>

<evaluation_criteria>
Use these criteria for detailed analysis:
1. **Entities and Objects:**
  - Do objects/entities in the caption appear in the video?
  - Are there missing or extra objects?
2. **Actions and Events:**
  - Are described actions/events clearly depicted?
  - Is the intensity/direction of actions consistent?
3. **Temporal Consistency:**
  - Does the video follow the event sequence in the caption?
  - Are durations and timing relationships preserved?
4. **Scene and Context:**
  - Does the overall setting match (location, time period, etc)?
  - Are contextual elements consistent (lighting, weather, atmosphere)?
</evaluation_criteria>

Please respond to the above task using the Chain of Thought (CoT) reasoning method. Your response should consist of multiple steps, each of which includes three types of actions: **"Inner Thinking"**, **"Final Conclusion"**, and **"Verification"**:

- **"Inner Thinking"**: Perform step-by-step analysis using the 4 evaluation criteria. For each criterion:
  1. Identify relevant elements in the caption
  2. Check their presence/accuracy in the video
  3. Note any discrepancies
  Each step should have a brief title indicating the criterion.

- **"Final Conclusion"**: Summarize the correct reasoning from all previous "Inner Thinking" steps and provide the detailed justification for why this specific sa_score was assigned to the video-caption pair. No title is needed.

- **"Verification"**: Verify the conclusion from the "Final Conclusion" step. If the conclusion is correct, end the reasoning process. If not, return to "Inner Thinking" for further analysis. No title is needed.

### Output Format:
Strictly follow the JSON structure below.

```json
{{
 "CoT": [
   {{"action": "Inner Thinking", "title": "...", "content": "..."}},
   ...,
   {{"action": "Final Conclusion", "content": "..."}},
   {{"action": "Verification", "content": "..."}}
 ]
}}
```

Figure 29: **SA:seed-ref prompt** used in Stage 2 for Eq. 8. The placeholders `{caption}`, `{reference_reason}`, and `{sa_score}` are shown in monospace. The reference rationale is produced by Stage 1 (see Fig. 3); the JSON output follows the specified CoT schema.

---

### Title:Stage 2 (judge): reference-equivalence verification

<Task>
Verify if the model-generated reason accurately aligns with the reference reason for the given SA score.
</Task>

<Model-Generated Reason>
{Model-Generated Reason}
</Model-Generated Reason>

<Reference Reason>
{Reference Reason}
</Reference Reason>

<Verification Criteria>
Output "True" ONLY if the meanings are substantially equivalent:

1. **Core Logic Consistency** (REQUIRED):
   - Both reasons focus on similar fundamental issues (missing objects, temporal misalignment, etc.)
   - Both reach the same conclusion about alignment quality
   - No major contradictions in evidence or assessment

2. **Key Assessment Coverage** (REQUIRED):
   - Both identify similar specific elements (objects, actions, scenes, timing)
   - Both note comparable discrepancies or matches
   - Both provide similar level of analytical depth

3. **Score Justification Alignment** (REQUIRED):
   - Both reasons logically support the same SA score level
   - Both assess severity of alignment issues similarly
   - Both demonstrate comparable evaluation standards

Output "False" if ANY of the following occur:
- Contradictory evidence (one says match, other says mismatch)
- Different fundamental reasoning approaches
- Would logically support different SA scores
- Major differences in identified issues or assessment depth

 CRITICAL OUTPUT REQUIREMENTS:
- Your response MUST be EXACTLY one word: either "True" or "False"
- Do NOT include any explanations, reasoning, or additional text
- Do NOT use quotes, punctuation, or formatting
- Do NOT provide any other response format

EXAMPLES OF CORRECT OUTPUT:
True
False

</Verification Criteria>

Figure 30: **SA:Judge prompt** used in Stage 2 by $\mathcal{V}_\tau$ for Eq. 9, Eq. 12, and Eq. 14. The placeholders { } are shown in monospace and highlighted in blue.

---

## Title:Stage~2 (backtracking): verification-guided CoT refinement

<task>Analyze the alignment between a video and its corresponding caption, then explain why the given alignment score (sa_score) is appropriate.
</task>

<caption>
{caption}
</caption>

<sa_score>
{sa_score}
</sa_score>

<scoring_rules>
- **1:** No alignment. The video does not match the caption at all (e.g., different objects, events, or scene).
- **2:** Poor alignment. Only a few elements of the caption are depicted, but key objects or events are missing or incorrect.
- **3:** Moderate alignment. The video matches the caption partially, but there are inconsistencies or omissions.
- **4:** Good alignment. Most elements of the caption are depicted correctly in the video, with minor issues.
- **5:** Perfect alignment. The video fully adheres to the caption with no inconsistencies.
</scoring_rules>

<evaluation_criteria>
Use these criteria for detailed analysis:
1. **Entities and Objects:**
   - Do objects/entities in the caption appear in the video?
   - Are there missing or extra objects?
2. **Actions and Events:**
   - Are described actions/events clearly depicted?
   - Is the intensity/direction of actions consistent?
3. **Temporal Consistency:**
   - Does the video follow the event sequence in the caption?
   - Are durations and timing relationships preserved?
4. **Scene and Context:**
   - Does the overall setting match (location, time period, etc)?
   - Are contextual elements consistent (lighting, weather, atmosphere)?
</evaluation_criteria>

<previous reasoning>
{previous_reason}
</previous reasoning>

<response requirements>
Please respond to the above task using the Chain of Thought (CoT) reasoning method. Your response should consist of multiple steps, each of which includes three types of actions: **"Inner Thinking"**, **"Final Conclusion"**, and **"Verification"**:

- **"Inner Thinking"**: Perform step-by-step analysis using the 4 evaluation criteria. For each criterion:
  1. Identify relevant elements in the caption
  2. Check their presence/accuracy in the video
  3. Note any discrepancies
  Each step should have a brief title indicating the criterion.

- **"Final Conclusion"**: Summarize the correct reasoning from all previous "Inner Thinking" steps and provide the detailed justification for why this specific sa_score was assigned to the video-caption pair. No title is needed.

- **"Verification"**: Verify the conclusion from the "Final Conclusion" step. If the conclusion is correct, end the reasoning process. If not, return to "Inner Thinking" for further analysis. No title is needed.

</response requirements>

<task> Analyze the alignment between a video and its corresponding caption, then explain why the given alignment score (sa_score) is appropriate.<previous reasoning> contains your prior reasoning. Your task is to continue from the current 'Verification' step. I have manually reviewed the reasoning and determined that the **Final Conclusion** is false. Your 'Verification' results must align with mine. Proceed to refine the reasoning using **backtracking** to revisit earlier points of reasoning and construct a new Final Conclusion.

### Output Format
Strictly follow the JSON structure below. You do not need to repeat your previous reasoning. Begin directly from the next 'Verification' stage.

```json
{{
"CoT": [
    {{"action": "Verification", "content": "..."}},
    {{"action": "Inner Thinking", "title": "...", "content": "..."}},
    ...,
    {{"action": "Final Conclusion", "content": "..."}},
    {{"action": "Verification", "content": "..."}}
]
}}
```

Figure 31: **SA:Backtracking prompt** used in Stage 2 within the CoT strategy set $\mathcal{C}$ (Eq. 7). This prompt resumes at `Verification`, treats the prior `Final Conclusion` as false, and directs a validation-driven backtrack to earlier reasoning before constructing a new conclusion. The JSON output begins with `Verification`, proceeds through `Inner Thinking`, and ends with a new `Final Conclusion` and `Verification`. Placeholders {caption}, {sa_score}, {reference_reason}, and {previous_reason} are shown in monospace.

---

### Title:Stage~2 (ExploringNewPaths): exploration-guided CoT refinement

```
<task>
Analyze the alignment between a video and its corresponding caption, then explain why the given alignment score (sa_score) is appropriate.
</task>

<caption>
{caption}
</caption>

<sa_score>
{sa_score}
</sa_score>

<scoring_rules>
- **1:** No alignment. The video does not match the caption at all (e.g., different objects, events, or scene).
- **2:** Poor alignment. Only a few elements of the caption are depicted, but key objects or events are missing or incorrect.
- **3:** Moderate alignment. The video matches the caption partially, but there are inconsistencies or omissions.
- **4:** Good alignment. Most elements of the caption are depicted correctly in the video, with minor issues.
- **5:** Perfect alignment. The video fully adheres to the caption with no inconsistencies.
</scoring_rules>

<evaluation_criteria>
Use these criteria for detailed analysis:
1. **Entities and Objects:**
   - Do objects/entities in the caption appear in the video?
   - Are there missing or extra objects?
2. **Actions and Events:**
   - Are described actions/events clearly depicted?
   - Is the intensity/direction of actions consistent?
3. **Temporal Consistency:**
   - Does the video follow the event sequence in the caption?
   - Are durations and timing relationships preserved?
4. **Scene and Context:**
   - Does the overall setting match (location, time period, etc)?
   - Are contextual elements consistent (lighting, weather, atmosphere)?
</evaluation_criteria>

<previous reasoning>
{previous_reasoning}
</previous reasoning>

<response requirements>
Please respond to the above task using the Chain of Thought (CoT) reasoning method. Your response should consist of multiple steps, each of which includes three types of actions:
**"Inner Thinking"**, **"Final Conclusion"**, and **"Verification"**:

- **"Inner Thinking"**: Perform step-by-step analysis using the 4 evaluation criteria. For each criterion:
   1. Identify relevant elements in the caption
   2. Check their presence/accuracy in the video
   3. Note any discrepancies
   Each step should have a brief title indicating the criterion.

- **"Final Conclusion"**: Summarize the correct reasoning from all previous "Inner Thinking" steps and provide the detailed justification for why this specific sa_score was
assigned to the video-caption pair. No title is needed.

- **"Verification"**: Verify the conclusion from the "Final Conclusion" step. If the conclusion is correct, end the reasoning process. If not, return to "Inner Thinking" for further
analysis. No title is needed.

</response requirements>

<task> Analyze the alignment between a video and its corresponding caption, then explain why the given alignment score (sa_score) is appropriate.<previous reasoning> contains
your prior reasoning. Your task is to continue from the current 'Verification' step. I have manually reviewed the reasoning and determined that the **Final Conclusion** is false.
Your 'Verification' results must align with mine. Proceed to refine the reasoning by **exploring new approaches** to analyzing the video-caption alignment and construct a new
Final Conclusion.

### Output Format
Strictly follow the JSON structure below. You do not need to repeat your previous reasoning. Begin directly from the next 'Verification' stage.

```json
{{
"CoT": [
    {{"action": "Verification", "content": "..."}},
    {{"action": "Inner Thinking", "title": "...", "content": "..."}},
    ...,
    {{"action": "Final Conclusion", "content": "..."}},
    {{"action": "Verification", "content": "..."}}
]
}}
```

Figure 32: **SA:ExploringNewPaths prompt** used in Stage 2 within the CoT strategy set $\mathcal{C}$ (Eq. 7).
This prompt resumes at `Verification`, treats the prior `Final Conclusion` as false, and
instructs the model to explore new analytical approaches before forming a new conclusion. The
JSON output begins with `Verification`, proceeds through `Inner Thinking`, and ends with
a new `Final Conclusion` and `Verification`. Placeholders `{caption}`, `{sa_score}`,
`{reference_reason}`, and `{previous_reasoning}` are shown in monospace.

---

## Title:Stage~2 (Correction): correction-guided CoT refinement

\<task>

\<task>Analyze the alignment between a video and its corresponding caption, then explain why the given alignment score (sa_score) is appropriate.
\</task>

\<caption>
{caption}
\</caption>

\<sa_score>
{sa_score}
\</sa_score>

\<scoring_rules>
- **1:** No alignment. The video does not match the caption at all (e.g., different objects, events, or scene).
- **2:** Poor alignment. Only a few elements of the caption are depicted, but key objects or events are missing or incorrect.
- **3:** Moderate alignment. The video matches the caption partially, but there are inconsistencies or omissions.
- **4:** Good alignment. Most elements of the caption are depicted correctly in the video, with minor issues.
- **5:** Perfect alignment. The video fully adheres to the caption with no inconsistencies.
\</scoring_rules>

\<evaluation_criteria>
Use these criteria for detailed analysis:
1. **Entities and Objects:**
   - Do objects/entities in the caption appear in the video?
   - Are there missing or extra objects?
2. **Actions and Events:**
   - Are described actions/events clearly depicted?
   - Is the intensity/direction of actions consistent?
3. **Temporal Consistency:**
   - Does the video follow the event sequence in the caption?
   - Are durations and timing relationships preserved?
4. **Scene and Context:**
   - Does the overall setting match (location, time period, etc)?
   - Are contextual elements consistent (lighting, weather, atmosphere)?
\</evaluation_criteria>

\<previous reasoning>
{previous_reasoning}
\</previous reasoning>

\<response requirements>
Please respond to the above task using the Chain of Thought (CoT) reasoning method. Your response should consist of multiple steps, each of which includes three types of actions: **"Inner Thinking"**, **"Final Conclusion"**, and **"Verification"**:

- **"Inner Thinking"**: Perform step-by-step analysis using the 4 evaluation criteria. For each criterion:
  1. Identify relevant elements in the caption
  2. Check their presence/accuracy in the video
  3. Note any discrepancies
  Each step should have a brief title indicating the criterion.

- **"Final Conclusion"**: Summarize the correct reasoning from all previous "Inner Thinking" steps and provide the detailed justification for why this specific sa_score was assigned to the video-caption pair. No title is needed.

- **"Verification"**: Verify the conclusion from the "Final Conclusion" step. If the conclusion is correct, end the reasoning process. If not, return to "Inner Thinking" for further analysis. No title is needed.

\</response requirements>

\<task> Analyze the alignment between a video and its corresponding caption, then explain why the given alignment score (sa_score) is appropriate.\<previous reasoning> contains your prior reasoning. Your task is to continue from the current 'Verification' step. I have manually reviewed the reasoning and determined that the **Final Conclusion** is false. Your 'Verification' results must align with mine. Proceed to refine the reasoning by making precise **corrections** to address prior flaws in your analysis and construct a new Final Conclusion.

### Output Format
Strictly follow the JSON structure below. You do not need to repeat your previous reasoning. Begin directly from the next 'Verification' stage.

```json
{{
"CoT": [
    {{"action": "Verification", "content": "..."}},
    {{"action": "Inner Thinking", "title": "...", "content": "..."}},
    ...,
    {{"action": "Final Conclusion", "content": "..."}},
    {{"action": "Verification", "content": "..."}}
]
}}
```

Figure 33: **SA:Correction prompt** used in Stage 2 within the CoT strategy set $\mathcal{C}$ (Eq. 7). This prompt resumes at `Verification`, assumes the prior `Final Conclusion` is false, and instructs precise corrections to earlier analysis before forming a new conclusion. The JSON output begins with `Verification`, proceeds through `Inner Thinking`, and ends with a new `Final Conclusion` and `Verification`. Placeholders {caption}, {sa_score}, {refrence_reason}, and {previous_reasoning} are shown in monospace.

---

### Title:Stage~2 (Verification): verification-guided CoT refinement

```
<task>
Analyze the alignment between a video and its corresponding caption, then explain why the given alignment score (sa_score) is appropriate.
</task>

<caption>
{caption}
</caption>

<sa_score>
{sa_score}
</sa_score>

<scoring_rules>
- **1:** No alignment. The video does not match the caption at all (e.g., different objects, events, or scene).
- **2:** Poor alignment. Only a few elements of the caption are depicted, but key objects or events are missing or incorrect.
- **3:** Moderate alignment. The video matches the caption partially, but there are inconsistencies or omissions.
- **4:** Good alignment. Most elements of the caption are depicted correctly in the video, with minor issues.
- **5:** Perfect alignment. The video fully adheres to the caption with no inconsistencies.
</scoring_rules>

<evaluation_criteria>
Use these criteria for detailed analysis:
1. **Entities and Objects:**
   - Do objects/entities in the caption appear in the video?
   - Are there missing or extra objects?
2. **Actions and Events:**
   - Are described actions/events clearly depicted?
   - Is the intensity/direction of actions consistent?
3. **Temporal Consistency:**
   - Does the video follow the event sequence in the caption?
   - Are durations and timing relationships preserved?
4. **Scene and Context:**
   - Does the overall setting match (location, time period, etc)?
   - Are contextual elements consistent (lighting, weather, atmosphere)?
</evaluation_criteria>

<previous reasoning>
{previous_reasoning}
</previous reasoning>

<response requirements>
Please respond to the above task using the Chain of Thought (CoT) reasoning method. Your response should consist of multiple steps, each of which includes three types of actions:
**"Inner Thinking"**, **"Final Conclusion"**, and **"Verification"**:

- **"Inner Thinking"**: Perform step-by-step analysis using the 4 evaluation criteria. For each criterion:
  1. Identify relevant elements in the caption
  2. Check their presence/accuracy in the video
  3. Note any discrepancies
  Each step should have a brief title indicating the criterion.

- **"Final Conclusion"**: Summarize the correct reasoning from all previous "Inner Thinking" steps and provide the detailed justification for why this specific sa_score was assigned to the video-caption pair. No title is needed.

- **"Verification"**: Verify the conclusion from the "Final Conclusion" step. If the conclusion is correct, end the reasoning process. If not, return to "Inner Thinking" for further analysis. No title is needed.

</response requirements>

<task> Analyze the alignment between a video and its corresponding caption, then explain why the given alignment score (sa_score) is appropriate.<previous reasoning> contains your prior reasoning. Your task is to continue from the current 'Verification' step. I have manually reviewed the reasoning and determined that the **Final Conclusion** is false. Your 'Verification' results must align with mine. Proceed to refine the reasoning by conducting a thorough **validation** process to ensure the accuracy of your analysis and construct a new Final Conclusion.

### Output Format
Strictly follow the JSON structure below. You do not need to repeat your previous reasoning. Begin directly from the next 'Verification' stage.

```json
{{
"CoT": [
    {{"action": "Verification", "content": "..."}},
    {{"action": "Inner Thinking", "title": "...", "content": "..."}},
    ...,
    {{"action": "Final Conclusion", "content": "..."}},
    {{"action": "Verification", "content": "..."}}
]
}}
```

Figure 34: **SA:Verification prompt** used in Stage 2 within the CoT strategy set $\mathcal{C}$ (Eq. 7). This prompt resumes at `Verification`, treats the prior `Final Conclusion` as false, and instructs a thorough validation process before forming a new conclusion. The JSON output begins with `Verification`, proceeds through `Inner Thinking`, and ends with a new `Final Conclusion` and `Verification`. Placeholders {caption}, {sa_score}, {reference_reason}, and {previous} are shown in monospace.

---

### Title:Stage~2 (rethink): LabelRethink reasoning

<task>
Analyze the alignment between a video and its corresponding caption, then explain why the given alignment score (sa_score) is appropriate.
</task>

<previous reasoning>
{previous_reasoning}
</previous reasoning>

<caption>
{caption}
</caption>

<sa_score>
{sa_score}
</sa_score>

<scoring_rules>
- **1:** No alignment. The video does not match the caption at all (e.g., different objects, events, or scene).
- **2:** Poor alignment. Only a few elements of the caption are depicted, but key objects or events are missing or incorrect.
- **3:** Moderate alignment. The video matches the caption partially, but there are inconsistencies or omissions.
- **4:** Good alignment. Most elements of the caption are depicted correctly in the video, with minor issues.
- **5:** Perfect alignment. The video fully adheres to the caption with no inconsistencies.
</scoring_rules>

<response requirements>
Please refer to the reference reason I provided and generate an appropriate thought process. Your response must include the following steps, each composed of three types of actions:
**"Inner Thinking"**, **"Final Conclusion"**, and **"Verification"**:

1. **Inner Thinking**: Break down the reasoning process into multiple concise steps. Each step should start with a brief title to clarify its purpose.
2. **Final Conclusion**: Summarize the correct reasoning from all previous 'Inner Thinking' steps and provide the detailed justification for the sa_score. No title is needed.
3. **Verification**: Verify the accuracy of the "Final Conclusion". If it holds, conclude the process. Otherwise, return to "Inner Thinking" for further refinement.

</response requirements>

<task> Analyze the alignment between a video and its corresponding caption, then explain why the given alignment score (sa_score) is appropriate.<previous reasoning> contains your prior reasoning. Your task is to continue from the current 'Verification' step. Now, I'll tell you that the correct reason is "{reference_reasoning}", please reorganize your thought process based on the reference reason to generate a final justification that matches the reference reason. Your 'Verification' requires careful consideration, and if incorrect, you need to provide new Inner Thinking steps and a new Final Conclusion to ensure the final reason aligns with the correct one.

### Output Format
Strictly follow the JSON structure below. You do not need to repeat your previous reasoning. Begin directly from the next 'Verification' stage.

```json
{{
"CoT": [
    {{"action": "Verification", "content": "..."}},
    {{"action": "Inner Thinking", "title": "...", "content": "..."}},
    ...,
    {{"action": "Final Conclusion", "content": "..."}},
    {{"action": "Verification", "content": "..."}}
]
}}
```

Figure 35: **SA:LabelRethink prompt** used in Stage 2 for Eq. 13, instantiated with $\mathbf{P}^{\tau}_{\text{rethink}}$, $x^{\tau}$, $r^{\tau}_{\text{ref}}$, and history $\mathcal{H}^{\tau}_{N}$. This prompt resumes from `Verification`, consumes prior reasoning and a provided correct reason, and instructs a rethink to produce a justification aligned with the reference. The JSON output begins with `Verification`, proceeds through `Inner Thinking`, and ends with a new `Final Conclusion` and `Verification`. Placeholders {caption}, {sa_score}, {previous_reasoning}, and {reference_reasoning} are shown in monospace and highlighted in blue.

---

**Title: Stage2 (verify): Video--Text Alignment Assessment for sa score**

\<Internal Thinking\>
{COT}
\</Internal Thinking\>

\<reference_reason\>
{reference_reason}
\</reference_reason\>

Based on the internal thinking process above, generate a **professional video-text alignment assessment** that explains why the sa_score is appropriate.

Your response should be a **concise, objective evaluation** (2-3 sentences) that:

1. **Identifies key alignment factors**: Mention specific entities, actions, temporal aspects, or scene elements
2. **Explains alignment issues**: Point out what matches well and what doesn't match
3. **Justifies the score**: Clearly state why this specific sa_score is appropriate
4. **Uses professional tone**: Academic/formal language, not conversational

**Example format**: "The video shows [specific observations] while the caption describes [specific elements]. The alignment is [good/moderate/poor] because [specific reasons]. This justifies an sa_score of X due to [key factors]."

**Scoring reference**:
- **Score 1**: No alignment at all
- **Score 2**: Poor alignment, major elements missing
- **Score 3**: Moderate alignment, some inconsistencies
- **Score 4**: Good alignment, minor issues
- **Score 5**: Perfect alignment, no inconsistencies

**Output Requirements**:
- Output ONLY the assessment text (no headers/formatting)
- 2-3 sentences maximum
- Professional, objective tone
- Clear justification for the score
- Focus on observable video-caption differences/similarities

Figure 36: **SA:Assessment prompt** used in Stage 2 to produce a professional video–text alignment assessment for task $\tau$ conditioned on prior reasoning and a reference rationale. Instantiated with {COT} inside <Internal Thinking> and {reference_reason} inside <reference_reason>, the prompt asks for a concise (2–3 sentences), objective justification of the appropriateness of the given *sa_score*, explicitly identifying key entities/actions/temporal cues, calling out mismatches, and stating the rationale for the score. The output must be *text only* (no headers/formatting), focus on observable video–caption similarities and differences, and follow the 1–5 scoring reference provided in the template. Placeholders {COT} and {reference_reason} are shown in monospace and highlighted in blue.



### Title:Stage 2 (PC seed): reference-conditioned reasoning

<task>
Evaluate whether the video follows physical commonsense, then explain why the given physical commonsense score (pc_score) is appropriate.
</task>

<reference_reason>
{reference_reason}
</reference_reason>

<pc_score>
{pc_score}
</pc_score>

<scoring_rules>
- **1:** No adherence to physical commonsense. The video contains numerous violations of fundamental physical laws.
- **2:** Poor adherence. Some elements follow physics, but major violations are present.
- **3:** Moderate adherence. The video follows physics for the most part but contains noticeable inconsistencies.
- **4:** Good adherence. Most elements in the video follow physical laws, with only minor issues.
- **5:** Perfect adherence. The video demonstrates a strong understanding of physical commonsense with no violations.
</scoring_rules>

<evaluation_criteria>
Use these criteria for detailed analysis:
1. **Object Behavior:**
   - Do objects behave according to their expected physical properties?
   - Are rigid objects deforming unnaturally or fluids flowing naturally?
2. **Motion and Forces:**
   - Are motions and forces depicted consistently with real-world physics?
   - Do gravity, inertia, and conservation of momentum apply correctly?
3. **Interactions:**
   - Do objects interact with each other and their environment plausibly?
   - Are there unnatural penetrations or inappropriate reactions on impact?
4. **Consistency Over Time:**
   - Does the video maintain consistency across frames?
   - Are there abrupt, unexplainable changes in object behavior or motion?
</evaluation_criteria>

Please respond to the above task using the Chain of Thought (CoT) reasoning method. Your response should consist of multiple steps, each of which includes three types of actions: **"Inner Thinking"**, **"Final Conclusion"**, and **"Verification"**:

- **"Inner Thinking"**: Perform step-by-step analysis using the 4 evaluation criteria. For each criterion:
  1. Observe the physical behaviors in the video
  2. Check their consistency with physical laws
  3. Note any violations or inconsistencies
  Each step should have a brief title indicating the criterion.

- **"Final Conclusion"**: Summarize the correct reasoning from all previous "Inner Thinking" steps and provide the detailed justification for why this specific pc_score was assigned to the video. No title is needed.

- **"Verification"**: Verify the conclusion from the "Final Conclusion" step. If the conclusion is correct, end the reasoning process. If not, return to "Inner Thinking" for further analysis. No title is needed.

### Output Format:
Strictly follow the JSON structure below.

```json
{{
  "CoT": [
    {{"action": "Inner Thinking", "title": "...", "content": "..."}},
    ...,
    {{"action": "Final Conclusion", "content": "..."}},
    {{"action": "Verification", "content": "..."}}
  ]
}}
```



Figure 37: **PC:seed-ref prompt** used in Stage 2 for Eq. 8. The placeholders {caption}, {reference_reason}, and {pc_score} are shown in monospace. The reference rationale is produced by Stage 1 (see Fig. 28); the JSON output follows the specified CoT schema.

---

**Title:Stage 2 (pc:judge): reference-equivalence verification**

<Task>
Verify if the model-generated reason accurately aligns with the reference reason for the given PC score.
</Task>

<Model-Generated Reason>
{Model-Generated Reason}
</Model-Generated Reason>

<Reference Reason>
{Reference Reason}
</Reference Reason>

<Verification Criteria>
Output "True" ONLY if the meanings are substantially equivalent:

1. **Core Logic Consistency** (REQUIRED):
   - Both reasons focus on similar fundamental physics issues (object behavior, motion laws, etc.)
   - Both reach the same conclusion about physical commonsense adherence
   - No major contradictions in evidence or assessment

2. **Key Assessment Coverage** (REQUIRED):
   - Both identify similar specific physical elements (forces, interactions, behaviors)
   - Both note comparable physics violations or correct behaviors
   - Both provide similar level of analytical depth

3. **Score Justification Alignment** (REQUIRED):
   - Both reasons logically support the same PC score level
   - Both assess severity of physics violations similarly
   - Both demonstrate comparable evaluation standards

Output "False" if ANY of the following occur:
- Contradictory evidence (one says physics violation, other says correct)
- Different fundamental reasoning approaches
- Would logically support different PC scores
- Major differences in identified issues or assessment depth

 CRITICAL OUTPUT REQUIREMENTS:
- Your response MUST be EXACTLY one word: either "True" or "False"
- Do NOT include any explanations, reasoning, or additional text
- Do NOT use quotes, punctuation, or formatting
- Do NOT provide any other response format

EXAMPLES OF CORRECT OUTPUT:
True
False

</Verification Criteria>"""

---

Figure 38: **PC:Judge prompt** used in Stage 2 by $\mathcal{V}_\tau$ for Eq. 9, Eq. 12, and Eq. 14. The placeholders { } are shown in monospace and highlighted in blue.

3402
3403
3404
3405
3406
3407
3408
3409
3410
3411
3412
3413
3414
3415
3416
3417
3418
3419
3420
3421
3422
3423
3424
3425
3426
3427
3428
3429
3430
3431
3432
3433
3434
3435
3436
3437
3438
3439
3440
3441
3442
3443
3444
3445
3446
3447
3448
3449
3450
3451
3452
3453
3454
3455

---

### Title:Stage~2 (pc:backtracking): verification-guided CoT refinement

<task>
Evaluate whether the video follows physical commonsense, then explain why the given physical commonsense score (pc_score) is appropriate.
</task>

<pc_score>
{pc_score}
</pc_score>

<scoring_rules>
- **1:** No adherence to physical commonsense. The video contains numerous violations of fundamental physical laws.
- **2:** Poor adherence. Some elements follow physics, but major violations are present.
- **3:** Moderate adherence. The video follows physics for the most part but contains noticeable inconsistencies.
- **4:** Good adherence. Most elements in the video follow physical laws, with only minor issues.
- **5:** Perfect adherence. The video demonstrates a strong understanding of physical commonsense with no violations.
</scoring_rules>

<evaluation_criteria>
Use these criteria for detailed analysis:
1. **Object Behavior:**
  - Do objects behave according to their expected physical properties?
  - Are rigid objects deforming unnaturally or fluids flowing naturally?
2. **Motion and Forces:**
  - Are motions and forces depicted consistently with real-world physics?
  - Do gravity, inertia, and conservation of momentum apply correctly?
3. **Interactions:**
  - Do objects interact with each other and their environment plausibly?
  - Are there unnatural penetrations or inappropriate reactions on impact?
4. **Consistency Over Time:**
  - Does the video maintain consistency across frames?
  - Are there abrupt, unexplainable changes in object behavior or motion?
</evaluation_criteria>

<previous reasoning>
{previous reasoning}
</previous reasoning>

<response requirements>
Please respond to the above task using the Chain of Thought (CoT) reasoning method. Your response should consist of multiple steps, each of which includes three types of actions:
**"Inner Thinking"**, **"Final Conclusion"**, and **"Verification"**:

- **"Inner Thinking"**: Perform step-by-step analysis using the 4 evaluation criteria. For each criterion:
  1. Observe the physical behaviors in the video
  2. Check their consistency with physical laws
  3. Note any violations or inconsistencies
  Each step should have a brief title indicating the criterion.

- **"Final Conclusion"**: Summarize the correct reasoning from all previous "Inner Thinking" steps and provide the detailed justification for why this specific pc_score was assigned to the video. No title is needed.

- **"Verification"**: Verify the conclusion from the "Final Conclusion" step. If the conclusion is correct, end the reasoning process. If not, return to "Inner Thinking" for further analysis. No title is needed.

</response requirements>

<task> Evaluate whether the video follows physical commonsense, then explain why the given physical commonsense score (pc_score) is appropriate.<previous reasoning>
contains your prior reasoning. Your task is to continue from the current 'Verification' step. I have manually reviewed the reasoning and determined that the **Final Conclusion** is false. Your 'Verification' results must align with mine. Proceed to refine the reasoning by conducting a thorough **backtracking** process to ensure the accuracy of your analysis and construct a new Final Conclusion.

### Output Format
Strictly follow the JSON structure below. You do not need to repeat your previous reasoning. Begin directly from the next 'Verification' stage.

```json
{{
"CoT": [
   {{"action": "Verification", "content": "..."}},
   {{"action": "Inner Thinking", "title": "...", "content": "..."}},
   ...,
   {{"action": "Final Conclusion", "content": "..."}},
   {{"action": "Verification", "content": "..."}}
]
}}
```

Figure 39: **PC:Backtracking prompt** used in Stage 2 within the CoT strategy set $\mathcal{C}$ (Eq. 7). This prompt resumes at `Verification`, treats the prior `Final Conclusion` as false, and directs a validation-driven backtrack to earlier reasoning before constructing a new conclusion. The JSON output begins with `Verification`, proceeds through `Inner Thinking`, and ends with a new `Final Conclusion` and `Verification`. Placeholders {caption}, {pc_score}, {reference_reason}, and {previous_reason} are shown in monospace.

3456
3457
3458
3459
3460
3461
3462
3463
3464
3465
3466
3467
3468
3469
3470
3471
3472
3473
3474
3475
3476
3477
3478
3479
3480
3481
3482
3483
3484
3485
3486
3487
3488
3489
3490
3491
3492
3493
3494
3495
3496
3497
3498
3499
3500
3501
3502
3503
3504
3505
3506
3507
3508
3509

### Title:Stage~2 (pc:ExploringNewPaths): exploration-guided CoT refinement

<task>
Evaluate whether the video follows physical commonsense, then explain why the given physical commonsense score (pc_score) is appropriate.
</task>

<pc_score>
{pc_score}
</pc_score>

<scoring_rules>
- **1:** No adherence to physical commonsense. The video contains numerous violations of fundamental physical laws.
- **2:** Poor adherence. Some elements follow physics, but major violations are present.
- **3:** Moderate adherence. The video follows physics for the most part but contains noticeable inconsistencies.
- **4:** Good adherence. Most elements in the video follow physical laws, with only minor issues.
- **5:** Perfect adherence. The video demonstrates a strong understanding of physical commonsense with no violations.
</scoring_rules>

<evaluation_criteria>
Use these criteria for detailed analysis:
1. **Object Behavior:**
   - Do objects behave according to their expected physical properties?
   - Are rigid objects deforming unnaturally or fluids flowing naturally?
2. **Motion and Forces:**
   - Are motions and forces depicted consistently with real-world physics?
   - Do gravity, inertia, and conservation of momentum apply correctly?
3. **Interactions:**
   - Do objects interact with each other and their environment plausibly?
   - Are there unnatural penetrations or inappropriate reactions on impact?
4. **Consistency Over Time:**
   - Does the video maintain consistency across frames?
   - Are there abrupt, unexplainable changes in object behavior or motion?
</evaluation_criteria>

<previous reasoning>
{previous reasoning}
</previous reasoning>

<response requirements>
Please respond to the above task using the Chain of Thought (CoT) reasoning method. Your response should consist of multiple steps, each of which includes three types of actions: **"Inner Thinking"**, **"Final Conclusion"**, and **"Verification"**:

- **"Inner Thinking"**: Perform step-by-step analysis using the 4 evaluation criteria. For each criterion:
  1. Observe the physical behaviors in the video
  2. Check their consistency with physical laws
  3. Note any violations or inconsistencies
  Each step should have a brief title indicating the criterion.

- **"Final Conclusion"**: Summarize the correct reasoning from all previous "Inner Thinking" steps and provide the detailed justification for why this specific pc_score was assigned to the video. No title is needed.

- **"Verification"**: Verify the conclusion from the "Final Conclusion" step. If the conclusion is correct, end the reasoning process. If not, return to "Inner Thinking" for further analysis. No title is needed.

</response requirements>

<task> Evaluate whether the video follows physical commonsense, then explain why the given physical commonsense score (pc_score) is appropriate.<previous reasoning> contains your prior reasoning. Your task is to continue from the current 'Verification' step. I have manually reviewed the reasoning and determined that the **Final Conclusion** is false. Your 'Verification' results must align with mine. Proceed to refine the reasoning by **exploring new approaches** to analyzing the video's physical commonsense and construct a new Final Conclusion.

### Output Format
Strictly follow the JSON structure below. You do not need to repeat your previous reasoning. Begin directly from the next 'Verification' stage.

```json
{{
"CoT": [
   {{"action": "Verification", "content": "..."}},
   {{"action": "Inner Thinking", "title": "...", "content": "..."}},
   ...,
   {{"action": "Final Conclusion", "content": "..."}},
   {{"action": "Verification", "content": "..."}}
]
}}
```

Figure 40: **PC:ExploringNewPaths prompt** used in Stage 2 within the CoT strategy set $\mathcal{C}$ (Eq. 7). This prompt resumes at `Verification`, treats the prior `Final Conclusion` as false, and instructs the model to explore new analytical approaches before forming a new conclusion. The JSON output begins with `Verification`, proceeds through `Inner Thinking`, and ends with a new `Final Conclusion` and `Verification`. Placeholders {caption}, {pc_score}, {reference_reason}, and {previous_reasoning} are shown in monospace.

---

### Title:Stage~2 (pc:Correction): correction-guided CoT refinement

```
<task>
Evaluate whether the video follows physical commonsense, then explain why the given physical commonsense score (pc_score) is appropriate.
</task>

<pc_score>
{pc_score}
</pc_score>

<scoring_rules>
- **1:** No adherence to physical commonsense. The video contains numerous violations of fundamental physical laws.
- **2:** Poor adherence. Some elements follow physics, but major violations are present.
- **3:** Moderate adherence. The video follows physics for the most part but contains noticeable inconsistencies.
- **4:** Good adherence. Most elements in the video follow physical laws, with only minor issues.
- **5:** Perfect adherence. The video demonstrates a strong understanding of physical commonsense with no violations.
</scoring_rules>

<evaluation_criteria>
Use these criteria for detailed analysis:
1. **Object Behavior:**
   - Do objects behave according to their expected physical properties?
   - Are rigid objects deforming unnaturally or fluids flowing naturally?
2. **Motion and Forces:**
   - Are motions and forces depicted consistently with real-world physics?
   - Do gravity, inertia, and conservation of momentum apply correctly?
3. **Interactions:**
   - Do objects interact with each other and their environment plausibly?
   - Are there unnatural penetrations or inappropriate reactions on impact?
4. **Consistency Over Time:**
   - Does the video maintain consistency across frames?
   - Are there abrupt, unexplainable changes in object behavior or motion?
</evaluation_criteria>

<previous reasoning>
{previous_reasoning}
</previous reasoning>

<response requirements>
Please respond to the above task using the Chain of Thought (CoT) reasoning method. Your response should consist of multiple steps, each of which includes three types of actions:
**"Inner Thinking"**, **"Final Conclusion"**, and **"Verification"**:

- **"Inner Thinking"**: Perform step-by-step analysis using the 4 evaluation criteria. For each criterion:
   1. Observe the physical behaviors in the video
   2. Check their consistency with physical laws
   3. Note any violations or inconsistencies
   Each step should have a brief title indicating the criterion.

- **"Final Conclusion"**: Summarize the correct reasoning from all previous "Inner Thinking" steps and provide the detailed justification for why this specific pc_score was
assigned to the video. No title is needed.

- **"Verification"**: Verify the conclusion from the "Final Conclusion" step. If the conclusion is correct, end the reasoning process. If not, return to "Inner Thinking" for further
analysis. No title is needed.

</response requirements>

<task> Evaluate whether the video follows physical commonsense, then explain why the given physical commonsense score (pc_score) is appropriate.<previous reasoning>
contains your prior reasoning. Your task is to continue from the current 'Verification' step. I have manually reviewed the reasoning and determined that the **Final Conclusion** is
false. Your 'Verification' results must align with mine. Proceed to refine the reasoning by making precise **corrections** to address prior flaws in your analysis and construct a new
Final Conclusion.

### Output Format
Strictly follow the JSON structure below. You do not need to repeat your previous reasoning. Begin directly from the next 'Verification' stage.

```json
{{
"CoT": [
   {{"action": "Verification", "content": "..."}},
   {{"action": "Inner Thinking", "title": "...", "content": "..."}},
   ...,
   {{"action": "Final Conclusion", "content": "..."}},
   {{"action": "Verification", "content": "..."}}
]
}}
```
```

Figure 41: **PC:Correction prompt** used in Stage 2 within the CoT strategy set $\mathcal{C}$ (Eq. 7). This prompt resumes at `Verification`, assumes the prior `Final Conclusion` is false, and instructs precise corrections to earlier analysis before forming a new conclusion. The JSON output begins with `Verification`, proceeds through `Inner Thinking`, and ends with a new `Final Conclusion` and `Verification`. Placeholders {caption}, {pc_score}, {refrence_reason}, and {previous_reasoning} are shown in monospace.

---

### Title:Stage~2 (pc:Verification): verification-guided CoT refinement

---

<task>
Evaluate whether the video follows physical commonsense, then explain why the given physical commonsense score (pc_score) is appropriate.
</task>

<pc_score>
{pc_score}
</pc_score>

<scoring_rules>
- **1:** No adherence to physical commonsense. The video contains numerous violations of fundamental physical laws.
- **2:** Poor adherence. Some elements follow physics, but major violations are present.
- **3:** Moderate adherence. The video follows physics for the most part but contains noticeable inconsistencies.
- **4:** Good adherence. Most elements in the video follow physical laws, with only minor issues.
- **5:** Perfect adherence. The video demonstrates a strong understanding of physical commonsense with no violations.
</scoring_rules>

<evaluation_criteria>
Use these criteria for detailed analysis:
1. **Object Behavior:**
   - Do objects behave according to their expected physical properties?
   - Are rigid objects deforming unnaturally or fluids flowing naturally?
2. **Motion and Forces:**
   - Are motions and forces depicted consistently with real-world physics?
   - Do gravity, inertia, and conservation of momentum apply correctly?
3. **Interactions:**
   - Do objects interact with each other and their environment plausibly?
   - Are there unnatural penetrations or inappropriate reactions on impact?
4. **Consistency Over Time:**
   - Does the video maintain consistency across frames?
   - Are there abrupt, unexplainable changes in object behavior or motion?
</evaluation_criteria>

<previous reasoning>
{previous_reasoning}
</previous reasoning>

<response requirements>
Please respond to the above task using the Chain of Thought (CoT) reasoning method. Your response should consist of multiple steps, each of which includes three types of actions:
**"Inner Thinking"**, **"Final Conclusion"**, and **"Verification"**:

- **"Inner Thinking"**: Perform step-by-step analysis using the 4 evaluation criteria. For each criterion:
   1. Observe the physical behaviors in the video
   2. Check their consistency with physical laws
   3. Note any violations or inconsistencies
   Each step should have a brief title indicating the criterion.

- **"Final Conclusion"**: Summarize the correct reasoning from all previous "Inner Thinking" steps and provide the detailed justification for why this specific pc_score was assigned to the video. No title is needed.

- **"Verification"**: Verify the conclusion from the "Final Conclusion" step. If the conclusion is correct, end the reasoning process. If not, return to "Inner Thinking" for further analysis. No title is needed.

</response requirements>

<task> Evaluate whether the video follows physical commonsense, then explain why the given physical commonsense score (pc_score) is appropriate.<previous reasoning>
contains your prior reasoning. Your task is to continue from the current 'Verification' step. I have manually reviewed the reasoning and determined that the **Final Conclusion** is false. Your 'Verification' results must align with mine. Proceed to refine the reasoning by conducting a thorough **validation** process to ensure the accuracy of your analysis and construct a new Final Conclusion.

### Output Format
Strictly follow the JSON structure below. You do not need to repeat your previous reasoning. Begin directly from the next 'Verification' stage.

```json
{{
"CoT": [
   {{"action": "Verification", "content": "..."}},
   {{"action": "Inner Thinking", "title": "...", "content": "..."}},
   ...,
   {{"action": "Final Conclusion", "content": "..."}},
   {{"action": "Verification", "content": "..."}}
]
}}
```

Figure 42: **PC:Verification prompt** used in Stage 2 within the CoT strategy set $\mathcal{C}$ (Eq. 7). This prompt resumes at `Verification`, treats the prior `Final Conclusion` as false, and instructs a thorough validation process before forming a new conclusion. The JSON output begins with `Verification`, proceeds through `Inner Thinking`, and ends with a new `Final Conclusion` and `Verification`. Placeholders {caption}, {pc_score}, {reference_reason}, and {previous} are shown in monospace.

---

**Title:Stage~2 (pc:rethink): LabelRethink reasoning**

<task>
Evaluate whether the video follows physical commonsense, then explain why the given physical commonsense score (pc_score) is appropriate.
</task>

<previous reasoning>
{previous_reasoning}
</previous reasoning>

<pc_score>
{pc_score}
</pc_score>

<scoring_rules>
- **1:** No adherence to physical commonsense. The video contains numerous violations of fundamental physical laws.
- **2:** Poor adherence. Some elements follow physics, but major violations are present.
- **3:** Moderate adherence. The video follows physics for the most part but contains noticeable inconsistencies.
- **4:** Good adherence. Most elements in the video follow physical laws, with only minor issues.
- **5:** Perfect adherence. The video demonstrates a strong understanding of physical commonsense with no violations.
</scoring_rules>

<response requirements>
Please refer to the reference reason I provided and generate an appropriate thought process. Your response must include the following steps, each composed of three types of actions: **"Inner Thinking"**, **"Final Conclusion"**, and **"Verification"**:

1. **Inner Thinking**: Break down the reasoning process into multiple concise steps. Each step should start with a brief title to clarify its purpose.
2. **Final Conclusion**: Summarize the correct reasoning from all previous 'Inner Thinking' steps and provide the detailed justification for the pc_score. No title is needed.
3. **Verification**: Verify the accuracy of the "Final Conclusion". If it holds, conclude the process. Otherwise, return to "Inner Thinking" for further refinement.

</response requirements>

<task> Evaluate whether the video follows physical commonsense, then explain why the given physical commonsense score (pc_score) is appropriate.<previous reasoning>
contains your prior reasoning. Your task is to continue from the current 'Verification' step. Now, I'll tell you that the correct reason is "{reference_reasoning}", please reorganize your thought process based on the reference reason to generate a final justification that matches the reference reason. Your 'Verification' requires careful consideration, and if incorrect, you need to provide new Inner Thinking steps and a new Final Conclusion to ensure the final reason aligns with the correct one.

### Output Format
Strictly follow the JSON structure below. You do not need to repeat your previous reasoning. Begin directly from the next 'Verification' stage.

```json
{{
"CoT": [
    {{"action": "Verification", "content": "..."}},
    {{"action": "Inner Thinking", "title": "...", "content": "..."}},
    ...,
    {{"action": "Final Conclusion", "content": "..."}},
    {{"action": "Verification", "content": "..."}}
]
}}
```

Figure 43: **PC:LabelRethink prompt** used in Stage 2 for Eq. 13, instantiated with $\mathbf{P}^\tau_{\text{rethink}}$, $x^\tau$, $r^\tau_{\text{ref}}$, and history $\mathcal{H}^\tau_N$. This prompt resumes from `Verification`, consumes prior reasoning and a provided correct reason, and instructs a rethink to produce a justification aligned with the reference. The JSON output begins with `Verification`, proceeds through `Inner Thinking`, and ends with a new `Final Conclusion` and `Verification`. Placeholders `{caption}`, `{pc_score}`, `{previous_reasoning}`, and `{reference_reasoning}` are shown in monospace and highlighted in blue.

---

**Title: Stage2 (pc:verify): Video--Text Alignment Assessment for pc score**

\<Internal Thinking>
{previous_thinking}
\</Internal Thinking>

\<reference_reason>
{reference_reason}
\</reference_reason>

Based on the internal thinking process above, generate a **professional physical commonsense assessment** that explains why the pc_score is appropriate.

Your response should be a **concise, objective evaluation** (2-3 sentences) that:

1. **Identifies key physics factors**: Mention specific object behaviors, forces, interactions, or physical laws
2. **Explains physics adherence**: Point out what follows physics correctly and what violates physical laws
3. **Justifies the score**: Clearly state why this specific pc_score is appropriate
4. **Uses professional tone**: Academic/formal language, not conversational

**Example format**: "The video demonstrates [specific physical behaviors] with [physics adherence level]. The physical commonsense is [good/moderate/poor] because [specific physics reasons]. This justifies a pc_score of X due to [key physical factors]."

**Scoring reference**:
- **Score 1**: No physics adherence, numerous violations
- **Score 2**: Poor adherence, major violations present
- **Score 3**: Moderate adherence, noticeable inconsistencies
- **Score 4**: Good adherence, minor physics issues
- **Score 5**: Perfect adherence, no violations

**Output Requirements**:
- Output ONLY the assessment text (no headers/formatting)
- 2-3 sentences maximum
- Professional, objective tone
- Clear justification for the score
- Focus on observable physics behaviors and laws
"""

---

Figure 44: **PC:Assessment prompt** used in Stage 2 to produce a professional video–text alignment assessment for task $\tau$ conditioned on prior reasoning and a reference rationale. Instantiated with {COT} inside `<Internal Thinking>` and {reference_reason} inside `<reference_reason>`, the prompt asks for a concise (2–3 sentences), objective justification of the appropriateness of the given *pc_score*, explicitly identifying key entities/actions/temporal cues, calling out mismatches, and stating the rationale for the score. The output must be *text only* (no headers/formatting), focus on observable video–caption similarities and differences, and follow the 1–5 scoring reference provided in the template. Placeholders {COT} and {reference_reason} are shown in monospace and highlighted in blue.

---

Title: Stage2 (post): NaturalReasoning — Convert Structured Analysis to Stream-of-Consciousness

```
<Thought Process>
{previous_reasoning}
</Thought Process>

Your task: Convert the structured analysis above into a natural, stream-of-consciousness thinking process, as if an expert is thinking out loud while watching the video.

**Required Elements:**
1. **Internal monologue style**: Use first-person thoughts like "I notice...", "Wait, let me look closer...", "Hmm, this seems..."
2. **Natural transitions**: Include hesitations, corrections, and discoveries like "Actually...", "Oh wait...", "But then again..."
3. **Sensory observations**: Describe what you're seeing in real-time: "The coin starts spinning...", "I can see the lighting..."
4. **Uncertainty and confirmation**: Show the thinking process: "This looks like...", "Yes, that confirms..."
5. **Step-by-step discovery**: Build up the analysis naturally, not as a pre-planned structure

**Example Style:**
"Let me watch this video carefully... I can see a coin on what appears to be a wooden surface. The caption says it spins 'rapidly' - let me check that. Hmm, it's definitely spinning, but actually the pace seems more moderate than rapid to me. Wait, I should also notice the background and setting..."

**Format Requirements:**
- Use natural paragraph breaks (not forced line breaks)
- Include thinking transitions and self-corrections
- Show the discovery process as it unfolds
- Make it sound conversational and authentic

### Output Format:
Strictly follow the JSON structure below.

```json
{{
  "NaturalReasoning": "..."
}}
```
"""
```

Figure 45: **NaturalReasoning prompt** used in Stage 2 to convert structured analysis into a stream-of-consciousness narration for task $\tau$. Instantiated with the accepted structured analysis serialized as {previous_reasoning}, this prompt instructs a natural, first-person internal monologue with hesitations, sensory observations, and step-by-step discovery, and requires the JSON output to strictly follow the schema with a single key "NaturalReasoning". The specification lists required elements (internal monologue style, natural transitions, sensory cues, uncertainty/confirmation, progressive discovery) and formatting constraints (natural paragraphs, no forced line breaks). Placeholders such as {previous_reasoning} are shown in monospace and highlighted in blue.

---

Title: SA (score remapping): DeepSeek-R1 evaluator prompt

```
You are an expert in semantic alignment evaluation. Based on the alignment analysis provided below, determine what alignment score (1-5) this analysis would correspond to according to the scoring rules.

Semantic Alignment Scoring Rules (1-5):

- **1:** No alignment. The video does not match the caption at all (e.g., different objects, events, or scene).
- **2:** Poor alignment. Only a few elements of the caption are depicted, but key objects or events are missing or incorrect.
- **3:** Moderate alignment. The video matches the caption partially, but there are inconsistencies or omissions.
- **4:** Good alignment. Most elements of the caption are depicted correctly, with minor issues.
- **5:** Perfect alignment. Fully adheres with no inconsistencies.

Evaluation Criteria:
1. **Entities and Objects:** Are the described objects/entities said to appear (no obvious missing/extra)?
2. **Actions and Events:** Are the described actions/events said to match (direction/intensity included)?
3. **Temporal Consistency:** Is the claimed event order/duration consistent?
4. **Scene and Context:** Is the claimed setting consistent (location/time/weather/lighting)?
"""

Alignment Analysis:
{reason_text}

Based on the analysis above, what semantic alignment score (1-5) does this analysis indicate? Consider:
- Which caption elements are claimed present/missing
- Whether actions/events (and their directions/intensities) are claimed to match
- Whether temporal order/duration are claimed to match
- Whether scene/context is claimed to match
- The severity of any mismatches described

IMPORTANT: Respond with ONLY the integer score (1, 2, 3, 4, or 5). Do not include any explanations or additional text.
```

Figure 46: **DeepSeek-R1 remapping prompt** used to convert a final *semantic-alignment* rationale into a scalar score $s_{\text{SA}} \in \{1, \ldots, 5\}$ for the SA ablations in Sec. 3.4. The template presents the *Semantic Alignment Scoring Rules (1–5)* and alignment-oriented *Evaluation Criteria*, and asks the model (Guo et al., 2025a) to read the provided Semantic Alignment Analysis (placeholder {final_response} shown in monospace/blue in the figure) and output *only* the integer score (no explanations). We run this prompt with temperature 0 and strict output checking.

---

**Title: SA (reason-quality): Qwen-VL-Max VLM-as-judge prompt**

You are a strict, no-nonsense judge. You will see a VIDEO, a CAPTION, and ONE generated explanation ("REASON").
Judge the REASON's quality for *Semantic Alignment (SA)* between CAPTION and VIDEO. Score ONLY from what is visible in the video and what is stated in the caption; do not guess or rely on outside knowledge. Do not produce chain-of-thought.

INPUTS
- CAPTION: {caption}
- VIDEO
- REASON: {reason}

SCALE
For every dimension use {0, 0.5, 1}. Be conservative:
- 1 = fully satisfied with *concrete, checkable* evidence that ties CAPTION ↔ VIDEO.
- 0.5 = partially satisfied, generic, or uncertain.
- 0 = contradicted by CAPTION/VIDEO, invented, or missing.

DIMENSIONS (definitions + hard caps)

1) Grounding (video evidence anchoring)
- 1: Cites multiple concrete, verifiable visual details (e.g., color/region/relative position/count/motion attribute) that clearly support the alignment claims.
- 0.5: Generally matches visuals but details are vague/partial.
- 0: Conflicts with visuals or speculative.
(HARD CAP: If no concrete visual detail appears anywhere, Grounding ≤ 0.5.)

2) Temporal Alignment (ordering/duration/frequency/causality vs. CAPTION)
- 1: Key temporal relations claimed vs. CAPTION (before/after/while/repeated/caused-by) are correct AND at least one is described concretely.
- 0.5: Temporal gist roughly right but generic/unclear OR not applicable/uncertain.
- 0: Temporal claims are wrong, reversed, invented, or unsupported.

3) Consistency (internal coherence & no hallucination vs. CAPTION/VIDEO)
- 1: Internally consistent; no contradictions; no invented key objects/events; no conflict with CAPTION or VIDEO.
- 0.5: Minor inconsistency or questionable mention that does not undermine the main claim.
- 0: Clear contradiction OR hallucinated key object/event.

4) Alignment Justification (explicit SA criterion/decision and evidence-based application)
- 1: Clearly states an alignment judgment (e.g., numeric/ordinal or explicit rule) AND applies it consistently to this VIDEO–CAPTION pair with concrete, visible evidence; no conflict with other dimensions.
- 0.5: Mentions an alignment judgment/rule but is generic, partially applied, or weakly tied to visible evidence.
- 0: No meaningful alignment criterion/decision is stated, OR it is misapplied/contradicted by evidence.

5) Coverage & Specificity (CAPTION elements)
- 1: Covers ≥2 key CAPTION elements (entities/actions/relations) and uses specific, checkable details (e.g., counts, colors, positions, motion attributes).
- 0.5: Mentions some CAPTION elements but incompletely or generically; limited specifics.
- 0: Ignores key CAPTION elements or provides no specific, checkable detail.

Strictly output the following JSON only:
```
{
  "scores": {
    "grounding": 0 | 0.5 | 1,
    "temporal_alignment": 0 | 0.5 | 1,
    "consistency": 0 | 0.5 | 1,
    "alignment_justification": 0 | 0.5 | 1,
    "coverage_specificity": 0 | 0.5 | 1
  }
}
```

---

Figure 47: **Qwen-VL-Max reason-evaluation prompt** used for the SA ablations in Sec. 3.4. The template instructs a hosted VLM (*Qwen-VL-Max*) to judge a generated REASON strictly from the CAPTION and visible VIDEO evidence, without chain-of-thought, on five dimensions (Grounding, Temporal Alignment, Consistency, Alignment Justification, Coverage&Specificity) with 3-point anchors {0, 0.5, 1} and a hard cap on Grounding. The prompt enforces a *strict JSON* schema for outputs and is run with temperature 0.1. Placeholders such as {reason} and {caption} are shown in monospace and highlighted in blue.

---

### Title: PC (score remapping): DeepSeek-R1 evaluator prompt

You are an expert in physical commonsense evaluation. Based on the physical commonsense analysis provided below, determine what score (1-5) this analysis would correspond to according to the scoring rules.

Physical Commonsense Scoring Rules (1-5):

- **1:** No adherence to physical commonsense. The video contains numerous violations of fundamental physical laws.
- **2:** Poor adherence. Some elements follow physics, but major violations are present.
- **3:** Moderate adherence. The video follows physics for the most part but contains noticeable inconsistencies.
- **4:** Good adherence. Most elements in the video follow physical laws, with only minor issues.
- **5:** Perfect adherence. The video demonstrates a strong understanding of physical commonsense with no violations.

Evaluation Criteria:
1. **Object Behavior:** Do objects behave according to their expected physical properties?
2. **Motion and Forces:** Are motions and forces depicted consistently with real-world physics?
3. **Interactions:** Do objects interact with each other and their environment plausibly?
4. **Consistency Over Time:** Does the video maintain consistency across frames?

Physical Commonsense Analysis:
{final_response}

Based on the analysis above, what physical commonsense score (1-5) does this analysis indicate? Consider:
- What level of physics adherence is described
- What types of violations or correct behaviors are mentioned
- How severe any physics issues are described to be
- Overall assessment of physical realism

IMPORTANT: Respond with ONLY the integer score (1, 2, 3, 4, or 5). Do not include any explanations or additional text."""

Figure 48: **DeepSeek-R1 remapping prompt** used to convert a final physical-commonsense rationale into a scalar score $s_{\mathrm{PC}} \in \{1, \ldots, 5\}$ for the PC ablations in Sec. 3.4. The template presents the *Physical Commonsense Scoring Rules (1–5)* and four *Evaluation Criteria* (Object Behavior, Motion & Forces, Interactions, Consistency Over Time) and asks the model (Guo et al., 2025a) to read the provided `Physical Commonsense Analysis` (placeholder `{final_response}` shown in monospace/blue in the figure) and output *only* the integer score (no explanations). We run this prompt with temperature 0 and strict output checking.

---

### Title: PC (reason-quality): Qwen-VL-Max VLM-as-judge prompt

You are a strict, no-nonsense judge. You will see a video (or frames) and ONE generated explanation ("reason").
Score ONLY from visible evidence; do not guess or use outside knowledge. Do not produce chain-of-thought.

INPUTS
- VIDEO
- REASON: {reason}

SCALE
For every dimension use {0, 0.5, 1}. Be conservative:
- 1 = fully satisfied with concrete, checkable details inside the reason.
- 0.5 = partially satisfied, generic, or uncertain.
- 0 = contradicted by the visuals, invented, or missing.

DIMENSIONS (definitions + hard caps)

1) Grounding (evidence anchoring)
- 1: Cites multiple concrete, verifiable visual details (e.g., color/region/relative position/count/motion attribute) that clearly support the claims.
- 0.5: Generally matches visuals but details are vague/partial.
- 0: Conflicts with visuals or speculative.
(HARD CAP: If no concrete visual detail appears anywhere, Grounding $\leq$ 0.5.)

2) Temporal (ordering/duration/frequency/causality)
- 1: Key temporal relations (before/after/while/repeated/caused-by) are correct AND at least one is described concretely.
- 0.5: Temporal gist roughly right but generic/unclear OR not applicable/uncertain.
- 0: Temporal claims are wrong, reversed, invented, or unsupported.

3) Consistency (internal coherence & no hallucination)
- 1: Internally consistent; no contradictions; no invented key objects/events; no conflict with the visuals (and caption/task if given).
- 0.5: Minor inconsistency or questionable mention that does not undermine the main claim.
- 0: Clear contradiction OR hallucinated key object/event.

4) Criteria & Justification (explicit evaluation rule/score and its evidence-based application)
- 1: Clearly states an evaluation criterion (e.g., numeric/ordinal score or explicit rule for judging) AND applies it consistently to this video with concrete, visible evidence; no conflict with other dimensions.
- 0.5: Mentions a criterion/score/rule but is generic, only partially applied, or weakly tied to visible evidence.
- 0: No meaningful criterion/score/rule is stated, OR it is misapplied/contradicted by the evidence.

5) Video Quality Assessment (clear judgment of whether the video itself is good or bad, grounded in what is visible)
- 1: Gives an explicit good/bad (or degree) judgment about the video's visual quality and backs it with concrete indicators (e.g., sharpness/blur, lighting/exposure, occlusion, framing/stability, scale/visibility of key entities).
- 0.5: Mentions quality in general terms (e.g., "clear/unclear") without concrete indicators, or uncertain.
- 0: No quality judgment, or the judgment contradicts what is visible.

Strictly output the following JSON only:
{
  "scores": {
    "grounding": 0 | 0.5 | 1,
    "temporal": 0 | 0.5 | 1,
    "consistency": 0 | 0.5 | 1,
    "criteria_justification": 0 | 0.5 | 1,
    "video_quality_assessment": 0 | 0.5 | 1
  }
}

Figure 49: **Qwen-VL-Max reason-evaluation prompt** used for the PC ablations in Sec. 3.4. The template instructs a hosted VLM (*Qwen-VL-Max*) to judge a generated REASON strictly from visible evidence, without chain-of-thought, on five dimensions (Grounding, Temporal, Consistency, Criteria&Justification, VideoQuality) with 3-point anchors $\{0, 0.5, 1\}$ and a hard cap on Grounding. The prompt enforces a *strict JSON* schema for outputs and is run with temperature 0.1. Placeholders such as {reason} are shown in monospace and highlighted in blue.

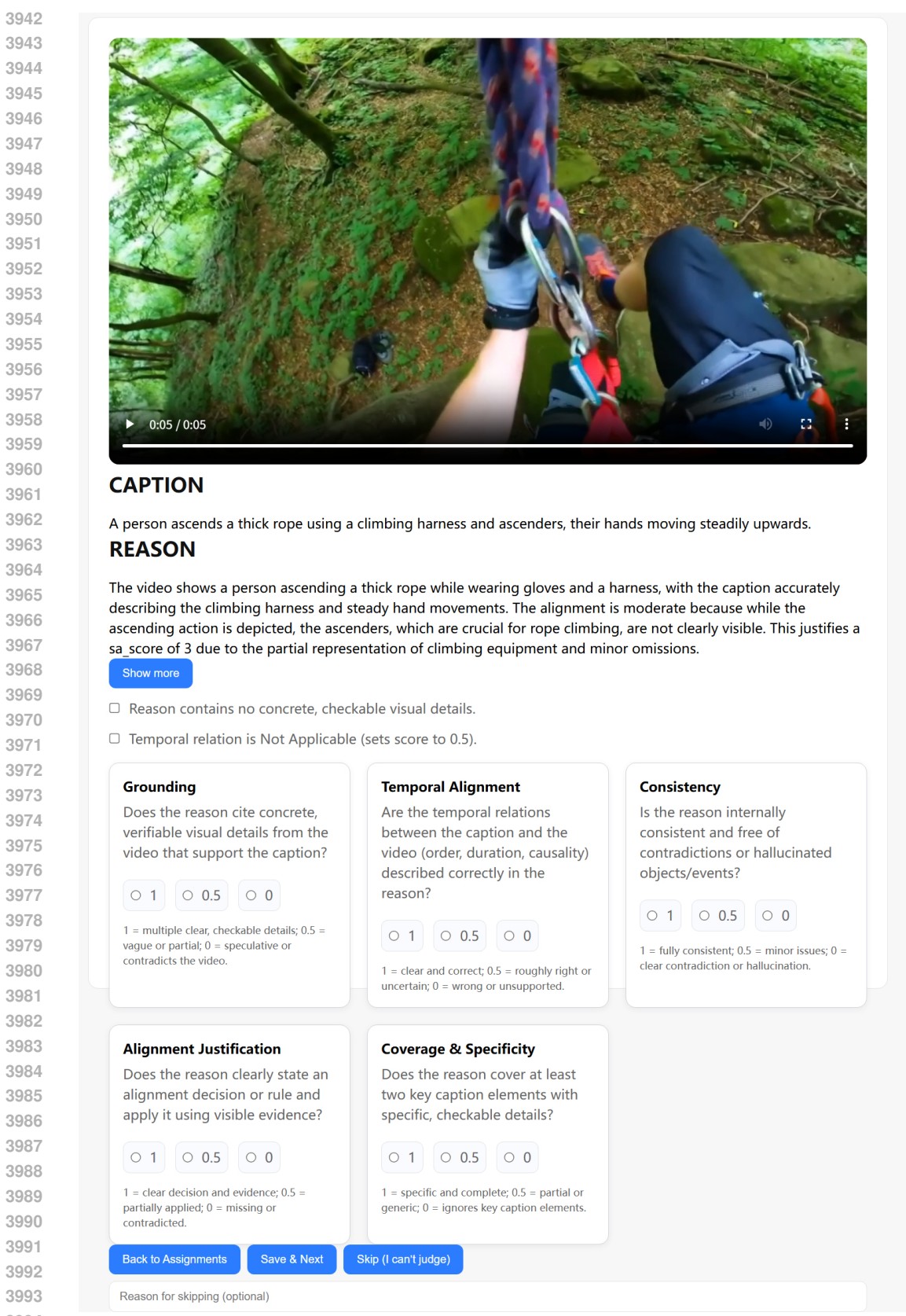

Figure 50: **Web interface for human evaluation of SA/PC rationales.** Annotators watch the video, read the caption (for SA), and assign 0/0.5/1 scores to each rubric dimension defined in Tables 7 and 8.

