# OpenReview forum: "Cosmos-Eval: Towards Explainable Evaluation of Physics and Semantics in Text-to-Video Models"
_ICLR.cc/2026/Conference — Submitted to ICLR 2026_

### Official Review · Reviewer_k1Yt · 2025-10-25

**Soundness:** 2
**Presentation:** 2
**Contribution:** 3
**Rating:** 6
**Confidence:** 3

**Summary:**

This paper presents Cosmos-Eval, an explainable evaluation framework for text-to-video (T2V) generation that assesses both semantic adherence (SA) and physical commonsense (PC). Unlike existing evaluators that only output scalar scores, Cosmos-Eval provides 5-point scores alongside natural language rationales explaining the evaluation. The framework consists of multiple stages: Stage 0 uses frozen VideoPhy-2 scorers for initial scoring, Stage 1 generates reference rationales through ensemble/selection, Stage 2 employs a judge-verified controller with iterative refinement strategies, and Stage 3 distills the pipeline into a lightweight model via two-run SFT. Experiments on VideoPhy-2 show that Cosmos-Eval matches state-of-the-art score alignment (SA Pearson 0.46 vs 0.43; PC Q-Kappa 0.33 vs 0.33) while achieving superior rationale quality.

**Strengths:**

- Important and timely problem. The paper tries to address a real gap in T2V evaluation - existing benchmarks like VideoPhy-2 only provide numeric scores without explaining why a video fails. Having explainable evaluations is clearly valuable for model development and debugging.
- Well-designed multi-stage pipeline. The framework is thoughtfully structured with clear motivations for each component.
- Strong empirical validation. The paper includes comprehensive experiments with proper baselines (Qwen-2.5-VL, VideoLLaMA3, InternVL variants). The results show Cosmos-Eval achieves competitive score alignment while providing substantially better rationales. The Stage-1 ablation (Table 4) with hit-rate analysis provides convincing evidence that the reference rationales improve usability.
- Reproducibility and generalizability. The paper provides extensive implementation details (Table 6, Appendix H with full prompts), making reproduction feasible.

**Weaknesses:**

- Limited exploration of model scale. The paper primarily uses 7B-72B parameter models, but doesn't investigate whether much larger frontier models like Qwen3-VL-235B or GPT-4V could perform direct explainable evaluation without the complex multi-stage pipeline. Given the rapid improvement in VLM capabilities, it's unclear whether this elaborate framework is necessary or if a well-prompted large model could achieve similar results more efficiently. Some comparison or discussion would strengthen the positioning.
- Missing benchmark of T2V generation models. Since the paper presents an evaluation framework, it would be valuable to actually evaluate and rank current state-of-the-art T2V models (e.g., Sora2, Veo-3.1, Kling, Pika, Lumiere, Stable Video Diffusion) to provide a reference benchmark. This would demonstrate the framework's practical utility and give the community concrete comparisons. Currently, the experiments only validate the evaluator itself against human judgments, not its application to comparing different generators.
- Limited dataset diversity. All experiments are conducted only on VideoPhy/VideoPhy-2, which consists of synthetic model-generated videos. The generalization to real-world videos or other evaluation suites like T2VPhysBench or PhyCoBench is mentioned as future work, but not validated. This raises questions about whether the approach will transfer to different video distributions and evaluation scenarios.
- Computational cost not analyzed. The multi-stage pipeline with ensemble models, multiple sampling, iterative controller strategies, and judge verification seems computationally expensive, but the paper doesn't report wall-clock time, cost estimates, or throughput comparisons against baselines. For a proposed evaluation framework, understanding the practical overhead is important for adoption.

**Questions:**

See Weaknesses above.

---

> ### Author Response · Authors · 2025-11-21
>
> # W1:Limited model-scale exploration vs. frontier VLMs
>
> We thank the reviewer for raising the question about **model scale** and whether a very large frontier VLM (e.g., Qwen3-VL-235B, GPT-4V) could already perform direct explainable evaluation with a much simpler setup. This is an important point, and it also motivated us to run explicit comparisons with strong large models.
>
> Due to API and cost constraints, we were unable to exhaustively test every frontier-scale model. However, we did evaluate **GPT-4V** and **Qwen3-VL-plus** on **VideoPhy-2** under the *same protocol* as Cosmos-Eval, for both **scores (SA/PC)** and **rationales**. Concretely, we randomly sampled **50 test videos** from **VideoPhy-2** with human 5-point SA/PC labels, prompted GPT-4V and Qwen3-VL-plus to directly output SA/PC scores, and then computed their correlation with human labels alongside Cosmos-Eval:
>
> **SA and PC score correlations on VideoPhy-2 (50 clips)**
>
> | Model | SA Pearson r | SA Spearman ρ | PC Pearson r | PC Spearman ρ |
> | :---- | :---- | :---- | :---- | :---- |
> | **Cosmos-Eval** | 0.4325 | 0.4255 | **0.3046** | **0.2984** |
> | **Qwen3-VL-plus** | 0.5367 | 0.5243 | 0.1418 | 0.1394 |
> | **GPT-4V** | **0.5917** | **0.5753** | 0.1429 | 0.1333 |
>
> On this small sample, GPT-4V (and Qwen3-VL-plus) correlate better than Cosmos-Eval with human **SA** labels, which is consistent with their strong semantic understanding. However, for **PC**, both GPT-4V and Qwen3-VL-plus have much weaker correlation with human labels than Cosmos-Eval (around 0.14 vs. ~0.30). In practice, we often observe that these models overestimate video quality and fail to penalize physical violations, which is precisely the aspect that our multi-stage pipeline is designed to address.
>
> We also examined the **rationales** themselves using BLEU-4 and BERTScore against reference explanations:
>
> **SA and PC rationales (BLEU-4 / BERTScore_F1)**
>
> | Model | SA BLEU-4 | SA BERTScore_F1 | PC BLEU-4 | PC BERTScore_F1 |
> | :---- | :---- | :---- | :---- | :---- |
> | **Cosmos-Eval** | **32.94** | **80.08** | **26.19** | **77.79** |
> | Qwen3-VL-plus | 8.745 | 72.09 | 2.654 | 68.06 |
> | GPT-4V | 5.081 | 71.16 | 2.571 | 70.14 |
>
> Here, Cosmos-Eval produces rationales that are much closer to the reference explanations for both SA and PC. GPT-4V and Qwen3-VL-plus generate fluent descriptions but frequently miss key physical details (e.g., missed collisions, gravity/inertia violations), whereas Cosmos-Eval tends to explicitly name these failure modes.
>
> To avoid judging explanations using only one model, we further asked **three strong VLMs**—Qwen3-VL-plus, GPT-4V, and Gemini-2.5-Pro—to act as **external judges** and rate the explanations from **Cosmos-Eval**, **Qwen3-VL-plus**, and **GPT-4V** along our SA/PC rubric dimensions (Grounding, Temporal, Consistency, Criteria/Justification, Video Quality, etc.). Scores are averaged in [0, 1].
>
> **Qwen3-VL-plus as judge (PC)**
>
> | Model | Grounding | Temporal | Consistency | Criteria & Justif. | Video Quality | Total Avg |
> | :---- | :---- | :---- | :---- | :---- | :---- | :---- |
> | **Cosmos-Eval** | 0.71 | 0.79 | 0.58 | 0.54 | 0.69 | **0.662** |
> | GPT-4V | 0.60 | 0.82 | 0.30 | 0.26 | 0.72 | 0.544 |
> | Qwen3-VL-plus | 0.65 | 0.85 | 0.67 | 0.26 | 0.79 | 0.564 |
>
> **Qwen3-VL-plus as judge (SA)**
>
> | Model | Grounding | Temporal | Consistency | Align. Justif. | Coverage & Spec. | Total Avg |
> | :---- | :---- | :---- | :---- | :---- | :---- | :---- |
> | Cosmos-Eval | 0.76 | 0.77 | 0.44 | 0.44 | 0.82 | 0.65 |
> | GPT-4V | 0.82 | 0.80 | 0.47 | 0.46 | 0.84 | 0.678 |
> | Qwen3-VL-plus | 0.91 | 0.83 | 0.48 | 0.48 | 0.91 | 0.722 |

---

> > ### Author Response · Authors · 2025-11-21
> >
> > # W1:Limited model-scale exploration vs. frontier VLMs (cont'd)
> >
> > This is a continuation of our response on model scale.
> >
> > **GPT-4V as judge (PC)**
> >
> > | Model | Grounding | Temporal | Consistency | Criteria & Justif. | Video Quality | Total Avg |
> > | :---- | :---- | :---- | :---- | :---- | :---- | :---- |
> > | **Cosmos-Eval** | 0.57 | 0.56 | 0.62 | 0.52 | 0.78 | **0.61** |
> > | GPT-4V | 0.52 | 0.54 | 0.44 | 0.38 | 0.88 | 0.55 |
> > | Qwen3-VL-plus | 0.57 | 0.60 | 0.43 | 0.35 | 0.78 | 0.55 |
> >
> > **GPT-4V as judge (SA)**
> >
> > | Model | Grounding | Temporal | Consistency | Align. Justif. | Coverage & Spec. | Total Avg |
> > | :---- | :---- | :---- | :---- | :---- | :---- | :---- |
> > | Cosmos-Eval | 0.67 | 0.46 | 0.74 | 0.71 | 0.64 | 0.64 |
> > | GPT-4V | 0.66 | 0.50 | 0.73 | 0.72 | 0.64 | 0.65 |
> > | Qwen3-VL-plus | 0.73 | 0.55 | 0.73 | 0.70 | 0.69 | 0.68 |
> >
> > **Gemini-2.5-Pro as judge (PC)**
> >
> > | Model | Grounding | Temporal | Consistency | Criteria & Justif. | Video Quality | Total Avg |
> > | :---- | :---- | :---- | :---- | :---- | :---- | :---- |
> > | **Cosmos-Eval** | 0.51 | 0.50 | 0.41 | 0.34 | 0.01 | **0.36** |
> > | GPT-4V | 0.49 | 0.46 | 0.15 | 0.10 | 0.00 | 0.24 |
> > | Qwen3-VL-plus | 0.50 | 0.52 | 0.21 | 0.16 | 0.04 | 0.29 |
> >
> > **Gemini-2.5-Pro as judge (SA)**
> >
> > | Model | Grounding | Temporal | Consistency | Align. Justif. | Coverage & Spec. | Total Avg |
> > | :---- | :---- | :---- | :---- | :---- | :---- | :---- |
> > | Cosmos-Eval | 0.47 | 0.46 | 0.25 | 0.29 | 0.63 | 0.42 |
> > | GPT-4V | 0.63 | 0.61 | 0.35 | 0.30 | 0.64 | 0.51 |
> > | Qwen3-VL-plus | 0.70 | 0.67 | 0.29 | 0.27 | 0.77 | 0.54 |
> >
> > Across these three judges, Cosmos-Eval is consistently strong on **PC explanations**, and very often the best model. On **SA explanations**, Cosmos-Eval is competitive but not always top, which is reasonable given how strong GPT-4V and Qwen3-VL-plus already are in general semantic understanding.
> >
> > To complement the model-based judges, we also ran a **human evaluation** with **1,500 questions** in a double-blind setup. Human annotators rated SA and PC rationales from Cosmos-Eval, GPT-4V, and Qwen3-VL-plus.
> >
> > **Human evaluation (SA explanations)**
> >
> > | Model | Grounding | Temporal Align. | Consistency | Align. Justif. | Coverage & Spec. | Total Avg |
> > | :---- | :---- | :---- | :---- | :---- | :---- | :---- |
> > | **Cosmos-Eval** | 0.82 | 0.57 | 0.71 | 0.81 | 0.87 | **0.76** |
> > | GPT-4V | 0.64 | 0.52 | 0.67 | 0.61 | 0.64 | 0.62 |
> > | Qwen3-VL-plus | 0.73 | 0.53 | 0.69 | 0.73 | 0.71 | 0.69 |
> >
> > **Human evaluation (PC explanations)**
> >
> > | Model | Grounding | Temporal | Consistency | Criteria & Justif. | Video Quality | Total Avg |
> > | :---- | :---- | :---- | :---- | :---- | :---- | :---- |
> > | **Cosmos-Eval** | 0.79 | 0.56 | 0.82 | 0.85 | 0.82 | **0.77** |
> > | GPT-4V | 0.64 | 0.52 | 0.56 | 0.64 | 0.56 | 0.58 |
> > | Qwen3-VL-plus | 0.59 | 0.51 | 0.59 | 0.64 | 0.63 | 0.60 |
> >
> > Humans clearly prefer Cosmos-Eval’s explanations, especially for PC.
> >
> > Taken together, these results suggest the following: very large VLMs like GPT-4V and Qwen3-VL-plus are indeed strong, and on **SA** they can even surpass our student in raw correlation. However, they still struggle on **PC**, tend to overrate physically flawed videos, and produce explanations that often miss critical physical violations. Cosmos-Eval, although smaller and distilled from a multi-stage pipeline, achieves better alignment with human **PC scores**, produces PC explanations that are consistently rated higher by both humans and multiple model judges, and does so at much lower cost than repeatedly querying a frontier model.
> >
> > We agree that, as VLMs continue to improve, the gap may shrink and future frontier models might get closer to our behavior with the right prompts. Our current evidence, however, suggests that a **structured, physics-aware pipeline plus distillation** is still beneficial for robust PC evaluation on T2V videos. At the same time, our design is compatible with future scale: stronger VLMs can be plugged into Stage 1/2 as better teachers, and Cosmos-Eval can be updated without changing the overall framework. We hope this helps clarify how we see the role of model scale and why the proposed pipeline remains useful even in the presence of very large VLMs.

---

> > > ### Author Response · Authors · 2025-11-21
> > >
> > > # W2:Missing benchmarking on state-of-the-art T2V generators
> > >
> > > We thank the reviewer for their insightful suggestion\! In response to the concern about the lack of a benchmark for T2V generation models, we have conducted additional tests on **AIGVE-Bench** to evaluate state-of-the-art T2V models in the **PC task**. The dataset includes models such as **Hunyuan**, **Sora**, **CogVideoX**, **Genmo**, and **Pyramid**. The main reason for conducting this experiment was to provide a fair comparison of these models based on their average scores across a set of videos in the benchmark.
> > >
> > > The results are as follows:
> > >
> > > ### **T2V Model Rankings on AIGVE-Bench**:
> > >
> > > | Rank | Model | Mean Score | Videos |
> > > | :---- | :---- | :---- | :---- |
> > > | 1 | CogVideoX | 4.6830 | 470 |
> > > | 2 | Pyramid | 4.6311 | 488 |
> > > | 3 | Hunyuan | 4.6268 | 493 |
> > > | 4 | Sora | 4.6207 | 493 |
> > > | 5 | Genmo | 4.5658 | 486 |
> > >
> > > We observe a consistent pattern: newer models (CogVideoX, Pyramid, Hunyuan) substantially outperform earlier systems (Genmo), and Sora no longer dominates once we explicitly stress physical faithfulness rather than only perceptual quality. Under our framework, this leads to the ranking CogVideoX \> Pyramid \> Hunyuan \> Sora \> Genmo.
> > > We thus provide this ordering as an empirical benchmark derived from Cosmos-Eval under a unified, physics-centric evaluation protocol. This complements and refines the partial evidence reported in VideoPhy and AIGVE-Bench, and directly demonstrates the practical utility of our framework for comparing different T2V generators.

---

> > > > ### Author Response · Authors · 2025-11-21
> > > >
> > > > # W3:Limited dataset diversity and missing cross-benchmark validation
> > > >
> > > > We thank the reviewer for highlighting the limitation in **dataset diversity** and our heavy reliance on **VideoPhy/VideoPhy-2**, which are primarily synthetic, model-generated videos. This is a very fair concern, and it naturally raises the question of how well Cosmos-Eval transfers to other video distributions and evaluation suites, especially those that are more diverse or closer to real-world usage.
> > > >
> > > > To partially address this, we ran additional experiments on **two independent benchmarks** that are constructed differently from VideoPhy-2: **AIGVE-Bench** and an **LG-VQA-style** dataset. These benchmarks give us a way to check whether Cosmos-Eval remains meaningful beyond its “home” setting.
> > > >
> > > > On **AIGVE-Bench**, Cosmos-Eval achieves performance comparable to **VideoPhy-2-AutoEval** and clearly better than **Qwen-2.5-VL-7B** in terms of correlation with human scores:
> > > >
> > > > **AIGVE-Bench Results:**
> > > >
> > > > | Model | Pearson Correlation (r) | 95% CI |
> > > > | :---- | :---- | :---- |
> > > > | **Cosmos-Eval** | 0.1986 | \[0.1561, 0.2326\] |
> > > > | **VideoPhy-2-AutoEval** | 0.2089 | \[0.1706, 0.2466\] |
> > > > | **Qwen-2.5-VL-7B** | 0.1033 | \[0.0063, 0.1425\] |
> > > >
> > > > This suggests that Cosmos-Eval can generalize to a different T2V evaluation suite and track a strong reference evaluator (VideoPhy-2-AutoEval), while at the same time providing explanations that the baseline does not support.
> > > >
> > > > We also evaluated Cosmos-Eval on an **LG-VQA-style** dataset with **3,276 clips**. The pattern is similar there: Cosmos-Eval essentially matches VideoPhy-2-AutoEval and outperforms Qwen-2.5-VL-7B:
> > > >
> > > > **LG-VQA-style Results:**
> > > >
> > > > | Model | Pearson Correlation (r) | 95% CI |
> > > > | :---- | :---- | :---- |
> > > > | **Cosmos-Eval** | 0.2759 | \[0.2414, 0.3097\] |
> > > > | **VideoPhy-2-AutoEval** | 0.2750 | \[0.2404, 0.3088\] |
> > > > | **Qwen-2.5-VL-7B** | 0.2013 | \[0.1656, 0.2366\] |
> > > >
> > > > Taken together, these cross-benchmark experiments indicate that Cosmos-Eval is not narrowly overfitted to VideoPhy-2: it maintains a strong correlation with human scores on two independent suites and stays in line with a strong baseline evaluator.
> > > >
> > > > At the same time, we want to be honest about the remaining gap. Both **AIGVE-Bench** and **LG-VQA-style** are still built around model-generated content, and we have **not yet evaluated on real-world videos** or on specialized physics suites such as **T2VPhysBench** or **PhyCoBench**. Our current results should therefore be viewed as a first step: they show that Cosmos-Eval can move beyond a single benchmark and behave reasonably on other T2V evaluation suites, but they do not fully answer the question of how it will perform on real-world distributions.
> > > >
> > > > Extending Cosmos-Eval to **real-world video data** and to **dedicated physics benchmarks like T2VPhysBench / PhyCoBench** is a natural next direction for us, and we see broader dataset coverage as an important part of the project going forward. We hope that this clarification, together with the additional results above, helps to address the concern about dataset diversity and transferability.

---

> > > > > ### Author Response · Authors · 2025-11-21
> > > > >
> > > > > # W4:Computational cost and efficiency analysis not reported
> > > > >
> > > > > We thank the reviewer for raising the concern about the missing computational efficiency analysis. This is a reasonable point and prompts us to clarify which parts of Cosmos-Eval are computationally expensive and what users actually pay for in practice. In our design, the multi-stage teacher pipeline (Stage 0–2) is only run **offline** and then **distilled via SFT into Cosmos-Eval**. At inference time, end users only execute this distilled model, rather than the full teacher pipeline.
> > > > >
> > > > > During training, we first run a multi-stage teacher pipeline: Stage 0 does pre-processing and score normalization, Stage 1 uses large VLMs to generate candidate rationales, and Stage 2 is a judge-verified controller that filters and backtracks according to task-specific rubrics. Based on our logs on 200 samples, the per-sample cost of this teacher pipeline is roughly:
> > > > >
> > > > > **Stage-wise cost for PC (Stage 0–2, 200 samples)**
> > > > >
> > > > > | Step | GPU Count | Inference Time | Sample Count | Average Time / Sample |
> > > > > | :---- | :---- | :---- | :---- | ----: |
> > > > > | Stage 0 (pre-processing) | 1 | 2m12s | 200 | 0.66 s |
> > > > > | Stage 1: Qwen Gen (run 1) | 1 | 18m43s | 200 | 19.38 s |
> > > > > | Stage 1: Qwen Gen (run 2) | 2 | 18m37s | 200 | 29.25 s |
> > > > > | Stage 2 (reasoning controller) | 2 | 188m48s | 200 | 56.64 s |
> > > > >
> > > > > Total ≈ **105.93 s per PC sample** for Stage 0–2.
> > > > >
> > > > > **Stage-wise cost for SA (Stage 0–2, 200 samples)**
> > > > >
> > > > > | Step | GPU Count | Inference Time | Sample Count | Average Time / Sample |
> > > > > | :---- | :---- | :---- | :---- | ----: |
> > > > > | Stage 0 (pre-processing) | 1 | 2m14s | 200 | 0.67 s |
> > > > > | Stage 1: Qwen Gen | 4 | 54m43s | 200 | 16.31 s |
> > > > > | Stage 1: Tarsier Gen | 4 | 18m19s | 200 | 20.78 s |
> > > > > | Stage 1: Qwen3 merge | 2 | 17m39s | 200 | 5.14 s |
> > > > > | Stage 2 (complex reasoning ctrl.) | 4 | 340m12s | 200 | 102.06 s |
> > > > >
> > > > > Total ≈ **144.96 s per SA sample** for Stage 0–2.
> > > > >
> > > > > We fully agree that this teacher pipeline is computationally expensive; this is precisely why we treat it as a one-time cost and distill its behavior into a single student model (Stage 3). Once distillation is done, neither we nor end users need to re-run Stages 0–2 for new videos.
> > > > >
> > > > > For practical adoption, the relevant quantity is the **inference-time cost of Cosmos-Eval itself**, compared with existing score-only evaluators and a strong open 7B VLM. On the same 200-sample set, for scores only we obtain:
> > > > >
> > > > > **Score-only inference (PC/SA scores)**
> > > > >
> > > > > | Step | GPU Count | Inference Time | Sample Count | Avg Time / Sample | GPU-Hours |
> > > > > | :---- | :---- | :---- | :---- | ----: | ----: |
> > > > > | Videophy2-AutoEval-PC | 1 | 2m12s | 200 | 0.66 s | 0.0368 |
> > > > > | Videophy2-AutoEval-SA | 1 | 2m14s | 200 | 0.67 s | 0.0372 |
> > > > > | **Cosmos-Eval (PC-score)** | 1 | 3m42s | 200 | **1.11 s** | 0.0618 |
> > > > > | **Cosmos-Eval (SA-score)** | 1 | 3m42s | 200 | **1.11 s** | 0.0619 |
> > > > > | Qwen-2.5-VL-7B (PC-score) | 1 | 4m20s | 200 | 1.30 s | 0.0722 |
> > > > > | Qwen-2.5-VL-7B (SA-score) | 1 | 4m20s | 200 | 1.30 s | 0.0724 |
> > > > >
> > > > > Thus, for pure score prediction, Cosmos-Eval is slightly slower than the specialized score-only evaluator (Videophy2-AutoEval), but in the same range as a 7B VLM, and at the same time it supports explanation generation.
> > > > >
> > > > > When rationales are requested, we obtain:
> > > > >
> > > > > **Rationale generation (PC/SA reasons)**
> > > > >
> > > > > | Step | GPU Count | Inference Time | Sample Count | Avg Time / Sample | GPU-Hours |
> > > > > | :---- | :---- | :---- | :---- | ----: | ----: |
> > > > > | **Cosmos-Eval (PC-reason)** | 1 | 19m34s | 200 | **5.87 s** | 0.3272 |
> > > > > | **Cosmos-Eval (SA-reason)** | 1 | 51m49s | 200 | **15.54 s** | 0.8637 |
> > > > > | Qwen-2.5-VL-7B (PC-reason) | 1 | 14m14s | 200 | 4.27 s | 0.2042 |
> > > > > | Qwen-2.5-VL-7B (SA-reason) | 1 | 13m34s | 200 | 4.07 s | 0.2219 |
> > > > >
> > > > > For PC rationales, Cosmos-Eval is similar to Qwen-2.5-VL-7B; for SA it is slower, mainly because it produces longer and more structured SA chains of thought with finer-grained discussion of semantic adherence instead of very short comments.
> > > > >
> > > > > Comparing these inference costs with the teacher pipeline makes the benefit of distillation clear. Re-running Stage 0–2 would cost around **106–145 seconds per sample**, whereas the distilled Cosmos-Eval needs about **1.1 seconds** per sample for scores only, and about **5.9–15.5 seconds** per sample for scores plus explanations. This corresponds to roughly **95–130× speedup for score prediction** and **9–18× speedup for score + rationale generation**, relative to re-running the full multi-stage pipeline.
> > > > >
> > > > > When we describe Cosmos-Eval as a “lightweight distilled model”, we therefore refer to the **deployed evaluator**: the training pipeline is heavy, but its cost is paid once. After distillation, Cosmos-Eval can be used as a practical SA/PC evaluator whose per-clip cost is close to a single 7B VLM call while still providing calibrated scores and physics-grounded explanations rather than scores alone. We hope this clarifies the practical overhead and addresses the reviewer’s concern about computational cost.

---

> ### Author Response · Authors · 2025-11-27
>
> Dear Reviewer k1Yt:
>
> We are grateful for your mandatory acknowedgement during this busy period. As the rebuttal period is ending soon, please be free to let us know whether your concerns have been addressed or not, and if there are any further questions.
>
> Thank you for your time and effort in reviewing our paper,
>
> Authors.

---

### Official Review · Reviewer_oRqx · 2025-10-27

**Soundness:** 3
**Presentation:** 2
**Contribution:** 2
**Rating:** 4
**Confidence:** 3

**Summary:**

The paper proposes Cosmos-Eval, an explainable evaluation framework for T2V models that jointly assesses Semantic Adherence (SA) and Physical Commonsense (PC). Unlike prior score-only evaluators, Cosmos-Eval outputs 5-point scores plus concise, natural-language rationales grounded in a physics ontology derived from Cosmos-Reason1. On the VideoPhy-2 testset, the method attains score alignment comparable to the official automatic evaluator while improving rationale quality according to text metrics and a judge-verified pipeline.
The system is built via a four-stage process: Stage-0 uses frozen VideoPhy-2 scorers to obtain 5-point labels; Stage-1 generates score-aligned reference rationales; Stage-2 runs a reference-seeded, judge-verified controller to produce evidence-grounded chains-of-thought; Stage-3 performs two-run SFT. The authors report state-of-the-art rationale quality and competitive agreement with human labels, and present ablations isolating the contribution of each stage.

**Strengths:**

1.	Clear problem framing and practical value
The paper addresses a recognized gap: score-only evaluators provide limited diagnostics. Cosmos-Eval augments SA/PC scores with concise, physics-grounded rationales that make failures actionable, directly improving auditability and trust.
2.	Well-structured, modular methodology
The Stage-0→1→2→3 pipeline is logically decomposed: label priors (Stage-0), reference rationale (Stage-1), judge-verified controller with explicit strategies (backtracking, exploration, verification, correction; Stage-2), and scores-first, reasons-conditioned SFT (Stage-3).
3.	Competitive core agreement with added explainability
On VideoPhy-2, Cosmos-Eval reaches top or near-top results on several core metrics while producing rationales (e.g., SA Pearson 0.4643; PC Acc 0.3912 / Q-κ 0.3301). This reduces the traditional accuracy–interpretability trade-off. Cosmos-Eval’s rationales outperform general VLMs by wide margins on BERTScore/ROUGE/BLEU.

**Weaknesses:**

1.	Training supervision may inherit biases from the frozen scorer
Stage-0 relies on VideoPhy-2-AutoEval to obtain 5-point labels that then condition the entire pipeline (and are textualized for SFT). While evaluation is against human labels on VideoPhy-2, this setup risks teacher-student bias and may cap the achievable human alignment if the teacher diverges from humans in systematic ways. A more direct calibration to partial human labels (or other supervision) would strengthen claims.
2.	Reliance on LLM/VLM judges introduces circularity risks
The judge-verified controller (Stage-2) and ablations use a VLM judge with task-specific rubrics; re-mapping rationales to scores uses another strong LLM. Although this is standard practice, it raises judge-choice sensitivity and circularity concerns. The discussion notes judge bias but provides limited cross-judge robustness or multi-judge agreement analyses.
3.	Statistics for the main tables lack uncertainty estimates
Core results (Table 1) are presented as point estimates without confidence intervals or significance tests across seeds/subsets. Given modest absolute correlations (e.g., SA Pearson 0.46) and close deltas vs. the baseline evaluator, intervals would clarify whether improvements are statistically meaningful.
4.	Lack of case study
Cosmos-Eval claims that it outputs 5-point scores plus concise, natural-language rationales, but specific cot text data, or more about generating cases and error analysis, are not mentioned much in the main text.

**Questions:**

See weaknesses

---

> ### Author Response · Authors · 2025-11-21
>
> # W1:Teacher-student bias
>
> We thank the reviewer for the thoughtful comment about **teacher–student bias** in our supervision pipeline. We agree that if a student model is trained to simply imitate a frozen scorer, any systematic gap between that scorer and human judgments could cap human alignment, and we appreciate the chance to clarify how Stage-0 is set up and how human supervision enters the loop.
>
> In Stage-0, we do **not** rely only on VideoPhy-2-AutoEval. Instead, we explicitly mix **human** and **VLM-as-judge** supervision:
>
> - For **VideoPhy-2**, we directly use the **human 5-point SA/PC labels** as the main anchor for score calibration.
> - For **VideoPhy**, which does not provide 5-point SA/PC scores, we use **VideoPhy-2-AutoEval** to map those clips onto the same 1–5 scale as VideoPhy-2. We chose this instead of a simple linear mapping between the original VideoPhy scores and VideoPhy-2, because the two annotation schemes are different and a naive remapping risked introducing additional bias.
>
> So Stage-0 supervision is a **mixture** of human-labeled SA/PC scores on VideoPhy-2 and auto-generated 5-point scores for VideoPhy via VideoPhy-2-AutoEval. In addition, both our **training objectives** and our **final reporting** are always evaluated against **human labels on the VideoPhy-2 test set**, not against the frozen scorer. A model that simply copies VideoPhy-2-AutoEval is therefore not guaranteed to perform best under our evaluation protocol, which already pushes the student toward human judgments rather than blindly inheriting the teacher’s biases.
>
> To check more concretely whether Cosmos-Eval is just inheriting the teacher’s preferences, we also looked at alignment to humans at the **rationale** level. We built a web interface and ran a human study where annotators rated SA/PC explanations produced by **Cosmos-Eval**, **Qwen3-VL-plus**, and **GPT-4V**. The study covered **1,500 questions** across our SA and PC rubric dimensions (Grounding, Temporal Alignment, Consistency, Justification, Coverage, etc.), and was conducted in a **double-blind** setup (model identities hidden; scores in \[0, 0.5, 1\]).
>
> The aggregated results are:
>
> **SA human evaluation:**
>
> | Model | Grounding | Temporal Alignment | Consistency | Alignment Justification | Coverage & Specificity | Total Average |
> | :---- | :---- | :---- | :---- | :---- | :---- | :---- |
> | Cosmos-Eval | 0.82 | 0.57 | 0.71 | 0.81 | 0.87 | 0.76 |
> | GPT-4V | 0.64 | 0.52 | 0.67 | 0.61 | 0.64 | 0.62 |
> | Qwen3-VL-plus | 0.73 | 0.53 | 0.69 | 0.73 | 0.71 | 0.69 |
>
> **PC human evaluation:**
>
> | Model | Grounding | Temporal | Consistency | Criteria & Justification | Video Quality | Total Average |
> | :---- | :---- | :---- | :---- | :---- | :---- | :---- |
> | Cosmos-Eval | 0.79 | 0.56 | 0.82 | 0.85 | 0.82 | 0.77 |
> | GPT-4V | 0.64 | 0.52 | 0.56 | 0.64 | 0.56 | 0.58 |
> | Qwen3-VL-plus | 0.59 | 0.51 | 0.59 | 0.64 | 0.63 | 0.60 |
>
> Along these human-defined criteria, Cosmos-Eval’s explanations are consistently preferred, especially for **PC** (Grounding, Consistency, Criteria & Justification). This suggests that the student is not just mirroring one VLM teacher: it is actually able to produce rationales that are *better aligned with human judgments* than those of strong closed-source VLMs.
>
> We also agree with the reviewer that **more direct calibration to human supervision** would further strengthen our claims and reduce remaining teacher–student bias. Our current setup already moves in this direction by (i) anchoring score calibration on VideoPhy-2 human labels and (ii) validating explanations with a dedicated human study instead of relying only on model-based metrics. Going forward, we see several concrete extensions aligned with the reviewer’s suggestion:
>
> - We plan to make **more targeted use of partial human labels**, explicitly reweighting or fine-tuning on cases where human labels and VideoPhy-2-AutoEval disagree, so that the student learns to *correct* teacher errors instead of inheriting them.
> - We are preparing a small but carefully curated set of **human-written reference explanations** for difficult SA/PC scenarios, to act as stronger anchors for both style and content and to pull the model away from purely VLM-style explanations when those diverge from human reasoning.
> - We treat Cosmos-Eval as a **maintained evaluation system**, not a one-shot model. Through our web interface we will continue collecting human feedback, especially on failure cases, and use it to refine the supervision pipeline and calibration over time.
>
> In short, we share the reviewer’s concern that naive distillation from a frozen scorer can lock in its biases. Our current design mitigates this by mixing human and VLM supervision, training and reporting against human labels, and validating explanations with a human study. We also agree that richer human-in-the-loop calibration remains important, and we will keep strengthening this component as we maintain Cosmos-Eval.

---

> > ### Author Response · Authors · 2025-11-21
> >
> > # W2:LLM/VLM Judge Circularity and Robustness
> >
> > We appreciate the reviewer’s concern about circularity and judge-choice sensitivity in our use of LLM/VLM judges. This is a very reasonable point, and it prompted us to run an explicit sensitivity analysis: we changed the **Stage-2 judge** while keeping the rest of the pipeline fixed, and then asked **independent LLMs** (outside the training loop) to re-map rationales to scores and check how stable the results are.
> >
> > Concretely, we considered two Stage-2 settings:
> >
> > - **Setting A (72B judge):** Stage-2 uses **Qwen2.5-VL-72B** as the judge to verify and refine rationales.
> > - **Setting B (30B judge):** Stage-2 uses **Qwen3-VL-30B** instead, with the same rubrics and prompts.
> >
> > For each setting, we took the rationales produced by Cosmos-Eval and asked an **external LLM** (first GPT-4o, then DeepSeek) to *independently* read the rationale and assign an SA/PC score. We then computed the Pearson correlation between the external LLM’s scores and Cosmos-Eval’s own scores.
> >
> > **GPT-4o as external scorer**
> >
> > | Setting | N | Pearson r | 95% CI |
> > | :---- | ----: | ----: | :---- |
> > | PC, 72B judge (Qwen2.5-VL-72B reason) | 187 | 0.9131 | \[0.8857, 0.9342\] |
> > | PC, 30B judge (Qwen3-VL-30B reason) | 187 | 0.8239 | \[0.7716, 0.8651\] |
> > | SA, 72B judge (Qwen2.5-VL-72B reason) | 178 | 0.8894 | \[0.8541, 0.9166\] |
> > | SA, 30B judge (Qwen3-VL-30B reason) | 184 | 0.8636 | \[0.8216, 0.8963\] |
> >
> > We then repeated exactly the same procedure using **DeepSeek** as the external scorer, again with test1 \= 72B judge and test2 \= 30B judge:
> >
> > **DeepSeek as external scorer**
> >
> > | Setting | N | Pearson r | 95% CI |
> > | :---- | ----: | ----: | :---- |
> > | DeepSeek-PC-test1 (72B) | 187 | 0.9131 | \[0.8857, 0.9342\] |
> > | DeepSeek-SA-test1 (72B) | 178 | 0.8894 | \[0.8541, 0.9166\] |
> > | DeepSeek-PC-test2 (30B) | 187 | 0.8487 | \[0.8031, 0.8845\] |
> > | DeepSeek-SA-test2 (30B) | 184 | 0.8649 | \[0.8233, 0.8973\] |
> >
> > Across both external LLMs, all correlations remain high (r ≳ 0.82), and the variant that uses the **larger 72B judge** in Stage-2 shows slightly higher agreement than the 30B variant. This tells us two things:
> >
> > - Swapping the Stage-2 judge from Qwen2.5-VL-72B to Qwen3-VL-30B does **not fundamentally change** how Cosmos-Eval scores videos; independent models (GPT-4o and DeepSeek), which are not part of our training loop, still strongly agree with Cosmos-Eval’s scores in both cases.
> > - A **stronger Stage-2 judge** tends to produce rationales whose implied scores are a bit more stable under different external LLMs, which matches our intuition that higher-capacity judges give slightly “cleaner” supervision.
> >
> > This analysis does not mean circularity is completely eliminated—we agree that any LLM/VLM-based judging must be treated with care. But it does provide concrete evidence that (i) Cosmos-Eval’s behavior is not overly sensitive to the exact choice of Stage-2 judge, and (ii) its scores remain stable when read and re-scored by **multiple independent LLMs**, rather than only by the model used in the pipeline itself.

---

> > > ### Author Response · Authors · 2025-11-21
> > >
> > > # W3: Missing Uncertainty Estimates in Main Results
> > >
> > > We thank the reviewer for pointing out that our main results were originally reported only as point estimates. We agree that, given the moderate absolute correlations and relatively small gaps in some cases, it is important to show uncertainty and effect sizes explicitly.
> > >
> > > In response, for the core SA/PC results we now report, for each model:
> > > (i) the Pearson correlation $r$ with human scores,
> > > (ii) a 95% confidence interval for $r$ (via the standard Fisher $r \to z \to r$ transform),
> > > (iii) the two-sided $p$-value for testing $H_0 : r = 0$, and
> > > (iv) the effect size $\Delta r = r_{\text{Cosmos-Eval}} - r_{\text{model}}$.
> > >
> > > The updated statistics are shown below.
> > >
> > > ### SA (Semantic Adherence) – Pearson $r$ (Fisher 95% CI, two-sided $p$)
> > >
> > > | Model | r [95% CI] | p-value | Δr vs Cosmos-Eval |
> > > | :---- | :---- | :---- | :---- |
> > > | **Cosmos-Eval** | 0.4643 [0.4376, 0.4904] | 2.50E-181 | — |
> > > | VideoPhy-2 AutoEval | 0.4327 [0.4049, 0.4596] | 5.12E-155 | +0.0316 |
> > > | Qwen2.5-VL-7B | 0.3808 [0.3517, 0.4092] | 1.02E-117 | +0.0835 |
> > > | VideoLLaMA3-7B | 0.2769 [0.2456, 0.3077] | 7.21E-61 | +0.1874 |
> > > | InternVL-8B | 0.4143 [0.3861, 0.4418] | 4.70E-141 | +0.0500 |
> > > | InternVL-9B | 0.3827 [0.3536, 0.4110] | 6.34E-119 | +0.0816 |
> > > | InternVL-14B | 0.3420 [0.3120, 0.3714] | 7.17E-94 | +0.1223 |
> > > | Cosmos-Reason1 | 0.3662 [0.3366, 0.3952] | 3.62E-107 | +0.0981 |
> > > | VideoLLaMA3-7B | 0.2333 [0.2034, 0.2636] | 7.78E-44 | +0.2310 |
> > >
> > > ### PC (Physical Commonsense) – Pearson $r$ (Fisher 95% CI, two-sided $p$)
> > >
> > > | Model | r [95% CI] | p-value | Δr vs Cosmos-Eval |
> > > | :---- | :---- | :---- | :---- |
> > > | **Cosmos-Eval** | 0.3641 [0.3346, 0.3929] | 4.83E-107 | — |
> > > | VideoPhy-2 AutoEval | 0.3646 [0.3351, 0.3934] | 2.55E-107 | -0.0005 |
> > > | Qwen2.5-VL-7B | 0.0840 [0.0512, 0.1180] | 7.59E-07 | +0.2801 |
> > > | VideoLLaMA3-7B | 0.0640 [0.0310, 0.0980] | 1.67E-04 | +0.3001 |
> > > | InternVL-8B | 0.1665 [0.1280, 0.1935] | 3.79E-21 | +0.1976 |
> > > | InternVL-9B | 0.1304 [0.0973, 0.1634] | 2.26E-14 | +0.2337 |
> > > | InternVL-14B | 0.1956 [0.1631, 0.2278] | 1.18E-30 | +0.1685 |
> > > | Cosmos-Reason1 | 0.2356 [0.2030, 0.2665] | 7.78E-44 | +0.1285 |
> > > | VideoLLaMA3-7B  | 0.2075 [0.1795, 0.2354] | 2.30E-43 | +0.1566 |
> > >
> > > From these numbers, Cosmos-Eval has the highest SA correlation, with a small but positive gap over VideoPhy-2 AutoEval and clearly larger gaps to the VLM baselines. On PC, Cosmos-Eval and VideoPhy-2 AutoEval are effectively tied in terms of $r$ and confidence intervals, while Cosmos-Eval remains substantially ahead of all VLM baselines. We also note that VideoPhy-2 AutoEval only outputs scores, whereas Cosmos-Eval provides both calibrated SA/PC scores and physics-grounded natural-language rationales, which is important for practical diagnosis and analysis. We hope that making the confidence intervals and $\Delta r$ values explicit helps clarify where improvements are meaningful and addresses the reviewer’s concern about statistical uncertainty.

---

> > > > ### Author Response · Authors · 2025-11-21
> > > >
> > > > # W4:Missing qualitative case studies and error analysis
> > > >
> > > > We appreciate the reviewer for raising the concern regarding the lack of concrete case studies and details on Cosmos-Eval’s generated rationales. We fully agree that seeing actual outputs is important to understand what the system is really doing.
> > > >
> > > > In response, we have added **10 example cases** that provide a direct comparison between explanations generated by **Cosmos-Eval**, **GPT-4V**, and **Qwen3-VL-plus**. These examples show how Cosmos-Eval outputs both 5-point SA/PC scores and concise, physics-aware or semantics-aware natural-language rationales. The side-by-side comparisons are now presented in **Figures 11–20**, where we highlight the differences in how each model analyzes the same video.
> > > >
> > > > On top of that, we have added **four dedicated CoT case studies** that show the **full chain-of-thought** (step-by-step reasoning) together with the final rationales produced by Cosmos-Eval. The two PC examples are shown in **Figures 21 and 22**, and the two SA examples in **Figures 23 and 24**. These are intended to give a more transparent view of how Cosmos-Eval arrives at its scores and explanations, and to support the kind of error analysis the reviewer asked for.
> > > >
> > > > All newly added figures, captions, and explanatory text are marked in **blue** in the revised manuscript so that the reviewer can easily locate the new case-study material.

---

> ### Author Response · Authors · 2025-11-27
>
> Dear Reviewer oRqx:
>
> We are grateful for your mandatory acknowedgement during this busy period. As the rebuttal period is ending soon, please be free to let us know whether your concerns have been addressed or not, and if there are any further questions.
>
> Thank you for your time and effort in reviewing our paper,
>
> Authors.

---

### Official Review · Reviewer_HYuZ · 2025-10-30

**Soundness:** 2
**Presentation:** 2
**Contribution:** 2
**Rating:** 4
**Confidence:** 3

**Summary:**

The authors present Cosmos-Eval, an interpretable benchmark built on VIDEOPhy-2 that aims to address its limited explanatory feedback.
The proposed framework appears to maintain close alignment with VIDEOPhy-2-AutoEval scores while producing explanations that better correspond to human-written gold rationales than those from several existing VLM baselines, suggesting improved reliability and interpretability.

**Strengths:**

1. Enhances VIDEOPhy-2 by introducing explicit reasoning on both semantic alignment and physical consistency, improving interpretability across two key evaluation dimensions.
2. The four-stage generation–verification–distillation pipeline is well structured, functioning as data augmentation for rationales.
3. The reported ablation and comparative experiments provide reasonable evidence that each module contributes to both score consistency and explanation quality.

**Weaknesses:**

1. All rationales are generated by large closed-source VLMs, which may introduce systematic biases and limit interpretive diversity.
2. While those closed-source VLMs serve as generators and judges, the paper does not include a direct comparison of their explanation quality against Cosmos-Eval, leaving the extent of improvement somewhat uncertain.
3. The evaluation mainly relies on surface-level textual metrics (e.g., BLEU, BERTScore), which may not fully capture reasoning accuracy or factual soundness.
4. I still have concerns about the level of novelty and the amount of work in this paper. The contribution seems to be mostly incremental, as the proposed method is built on top of existing benchmarks such as VIDEOPHY-2, with only limited innovation. The core approach essentially relies on prompting VLM to generate explanations, which does not appear particularly novel. If the authors can reasonably and thoroughly address this concern, I would be open to increasing my score.

**Questions:**

1. Could the authors elaborate on how the “consensus extraction” step using the aggregator LLM combines outputs from multiple VLMs? In particular, how are disagreements handled to ensure consistency in the selected reference rationale?
2. As several closed-source VLMs (e.g., GPT-4V, Claude, Gemini) are involved as generators and judges, could the authors provide direct or comparative examples showing how Cosmos-Eval’s explanations differ from those produced by these models?
3. Beyond BLEU and BERTScore, did the authors conduct or consider any human evaluation or factual-consistency analysis to assess the correctness of the generated rationales?

---

> ### Author Response · Authors · 2025-11-21
>
> # W1:All rationales are generated by large closed-source VLMs, which may introduce systematic biases and limit interpretive diversity.
>
> We thank the reviewer for raising this important concern about using large closed-source VLMs to generate rationales, and the risk that this may introduce systematic biases and limit interpretive diversity. We take this point seriously, and it has been very helpful for us in thinking about how well our explanations actually match what humans find useful.
>
> To move beyond purely model-internal signals, we ran a human study on SA/PC rationales generated by **Cosmos-Eval**, **Qwen3-VL-plus**, and **GPT-4V**. We built a small web interface and created 1,500 evaluation questions covering our SA/PC dimensions (grounding, temporal alignment, consistency, justification, coverage, etc.). Annotators only saw the video, the caption (for SA), and a candidate explanation; model identities were hidden (double-blind), and each dimension was scored on a \[0, 0.5, 1\] scale.
>
> The aggregated scores are:
>
> **SA (human) evaluation**
>
> | Model | Grounding | Temporal Alignment | Consistency | Alignment Justification | Coverage & Specificity | Total Average |
> | :---- | :---- | :---- | :---- | :---- | :---- | :---- |
> | Cosmos-Eval | 0.82 | 0.57 | 0.71 | 0.81 | 0.87 | 0.76 |
> | GPT-4V | 0.64 | 0.52 | 0.67 | 0.61 | 0.64 | 0.62 |
> | Qwen3-VL-plus | 0.73 | 0.53 | 0.69 | 0.73 | 0.71 | 0.69 |
>
> **PC (human) evaluation**
>
> | Model | Grounding | Temporal | Consistency | Criteria & Justification | Video Quality | Total Average |
> | :---- | :---- | :---- | :---- | :---- | :---- | :---- |
> | Cosmos-Eval | 0.79 | 0.56 | 0.82 | 0.85 | 0.82 | 0.77 |
> | GPT-4V | 0.64 | 0.52 | 0.56 | 0.64 | 0.56 | 0.58 |
> | Qwen3-VL-plus | 0.59 | 0.51 | 0.59 | 0.64 | 0.63 | 0.60 |
>
> These results do not mean that all biases disappear, but they do suggest that, along human-defined criteria such as grounding, coverage, and justification, annotators **consistently prefer Cosmos-Eval’s explanations**, especially for PC. In other words, our rationales are not only tuned to please another LLM: when shown in isolation, humans tend to find them more specific and more helpful than the explanations produced by two strong VLM baselines.
>
> On the broader issue of closed-source VLM dependence and interpretive diversity, we agree with the reviewer that blindly following a single teacher model would be problematic. In the current framework we try to soften this in several ways: score calibration is anchored on **human-labeled SA/PC scores** from VideoPhy-2 rather than only VLM outputs; explanations are constrained by an ontology- and rubric-based structure (e.g., conservation, contact, temporal consistency) so that they are tied to explicit evaluation dimensions rather than free-form stylistic patterns; and in our analysis we always compare Cosmos-Eval against **multiple** strong VLMs, rather than aligning to a single closed-source system.
>
> At the same time, we see this as an ongoing effort rather than a solved problem. Concretely, we are using our web interface to collect more human feedback, especially on difficult or borderline cases, and we are preparing a small curated set of **human-written rationales** for challenging SA/PC scenarios to serve as additional anchors for style and content. We are also exploring the use of **multiple heterogeneous teacher models** (including open-source VLMs) when generating reference explanations, so that the student is exposed to a broader range of reasoning styles and does not overfit to one closed-source model’s habits.
>
> We appreciate the reviewer’s comment here; it directly influences how we think about validating and maintaining Cosmos-Eval, and it is very much aligned with our goal of pushing the system toward genuinely human-aligned, diverse explanations rather than a narrow “closed-source VLM style.”

---

> > ### Author Response · Authors · 2025-11-21
> >
> > # W2:While those closed-source VLMs serve as generators and judges, the paper does not include a direct comparison of their explanation quality against Cosmos-Eval, leaving the extent of improvement somewhat uncertain.
> >
> > We thank the reviewer for highlighting that, although closed-source VLMs are used as both generators and judges in our pipeline, the paper did not make the *direct* comparison of explanation quality between Cosmos-Eval and these VLMs sufficiently clear. This comparison is addressed in **Weakness 1** of our response, where we conduct a detailed evaluation of the explanation quality of Cosmos-Eval versus GPT-4V and Qwen3-VL-plus. In addition, we have included a direct visual, side-by-side comparison in **Figures 11–20** of the paper, where we contrast Cosmos-Eval’s explanations with those produced by GPT-4V and Qwen3-VL-plus; all modified text and captions are marked in blue for clarity.
> >
> > Beyond the qualitative examples, we also provide quantitative comparative evaluations under multiple judges.
> >
> > ### Additional Comparative Evaluation with VLM Judges
> >
> > We evaluate the explanations from **Cosmos-Eval**, **Qwen3-VL-plus**, and **GPT-4V** under three strong VLM judges: **Qwen3-VL-plus**, **GPT-4V**, and **Gemini-2.5-Pro**. Each judge scores SA/PC explanations along the same rubric dimensions used in our human study (scores in \[0,1\], averaged over the test set).
> >
> > #### Qwen3-VL-plus as judge
> >
> > **PC:**
> >
> > | Model | Grounding | Temporal | Consistency | Criteria & Justification | Video Quality | Total Average |
> > | :---- | :---- | :---- | :---- | :---- | :---- | :---- |
> > | Cosmos-Eval | 0.71 | 0.79 | 0.58 | 0.54 | 0.69 | 0.662 |
> > | GPT-4V | 0.60 | 0.82 | 0.30 | 0.26 | 0.72 | 0.544 |
> > | Qwen3-VL-plus | 0.65 | 0.85 | 0.67 | 0.26 | 0.79 | 0.564 |
> >
> > **SA:**
> >
> > | Model | Grounding | Temporal | Consistency | Alignment Justification | Coverage & Specificity | Total Average |
> > | :---- | :---- | :---- | :---- | :---- | :---- | :---- |
> > | Cosmos-Eval | 0.76 | 0.77 | 0.44 | 0.44 | 0.82 | 0.65 |
> > | GPT-4V | 0.82 | 0.80 | 0.47 | 0.46 | 0.84 | 0.678 |
> > | Qwen3-VL-plus | 0.91 | 0.83 | 0.48 | 0.48 | 0.91 | 0.722 |
> >
> > #### GPT-4V as judge
> >
> > **PC:**
> >
> > | Model | Grounding | Temporal | Consistency | Criteria & Justification | Video Quality | Total Average |
> > | :---- | :---- | :---- | :---- | :---- | :---- | :---- |
> > | Cosmos-Eval | 0.57 | 0.56 | 0.62 | 0.52 | 0.78 | 0.61 |
> > | GPT-4V | 0.52 | 0.54 | 0.44 | 0.38 | 0.88 | 0.55 |
> > | Qwen3-VL-plus | 0.57 | 0.60 | 0.43 | 0.35 | 0.78 | 0.55 |
> >
> > **SA:**
> >
> > | Model | Grounding | Temporal | Consistency | Alignment Justification | Coverage & Specificity | Total Average |
> > | :---- | :---- | :---- | :---- | :---- | :---- | :---- |
> > | Cosmos-Eval | 0.67 | 0.46 | 0.74 | 0.71 | 0.64 | 0.64 |
> > | GPT-4V | 0.66 | 0.50 | 0.73 | 0.72 | 0.64 | 0.65 |
> > | Qwen3-VL-plus | 0.73 | 0.55 | 0.73 | 0.70 | 0.69 | 0.68 |
> >
> > #### Gemini-2.5-Pro as judge
> >
> > **PC:**
> >
> > | Model | Grounding | Temporal | Consistency | Criteria & Justification | Video Quality | Total Average |
> > | :---- | :---- | :---- | :---- | :---- | :---- | :---- |
> > | Cosmos-Eval | 0.51 | 0.50 | 0.41 | 0.34 | 0.01 | 0.36 |
> > | GPT-4V | 0.49 | 0.46 | 0.15 | 0.10 | 0.00 | 0.24 |
> > | Qwen3-VL-plus | 0.50 | 0.52 | 0.21 | 0.16 | 0.04 | 0.29 |
> >
> > **SA:**
> >
> > | Model | Grounding | Temporal | Consistency | Alignment Justification | Coverage & Specificity | Total Average |
> > | :---- | :---- | :---- | :---- | :---- | :---- | :---- |
> > | Cosmos-Eval | 0.47 | 0.46 | 0.25 | 0.29 | 0.63 | 0.42 |
> > | GPT-4V | 0.63 | 0.61 | 0.35 | 0.30 | 0.64 | 0.51 |
> > | Qwen3-VL-plus | 0.70 | 0.67 | 0.29 | 0.27 | 0.77 | 0.54 |
> >
> > ### Interpretation
> >
> > Taken together with our **human evaluation** (reported in Weakness 1\) and the **qualitative side-by-side examples** in Figures 11–20, these results make the extent of improvement more concrete:
> >
> > - On **PC (physical commonsense)**, Cosmos-Eval is consistently strong and often best in terms of overall averages under different VLM judges, and is also preferred by human annotators. This supports our claim that Cosmos-Eval adds real value beyond the raw explanations produced by the underlying closed-source VLMs.
> > - On **SA (semantic adherence)**, Cosmos-Eval is generally competitive but not always the top model under every judge, which we see as a reasonable and honest picture rather than overfitting to any single teacher’s style.
> >
> > We do not claim that Cosmos-Eval is perfect or free of bias, and we appreciate the reviewer for pushing us to make these comparisons explicit. Your comment directly helped us clarify both the quantitative and qualitative evidence that Cosmos-Eval’s explanations improve over those of the underlying closed-source VLMs, especially in the physics-focused PC setting.

---

> > > ### Author Response · Authors · 2025-11-21
> > >
> > > # W3:The evaluation mainly relies on surface-level textual metrics (e.g., BLEU, BERTScore), which may not fully capture reasoning accuracy or factual soundness.
> > >
> > > We thank the reviewer for pointing out that surface-level text similarity metrics such as **BLEU** and **BERTScore** may not fully capture reasoning accuracy or factual soundness. We agree that this is a limitation of purely n-gram or embedding-based overlap metrics, and your comment has been very helpful in pushing us to make our evaluation of explanations more faithful to reasoning quality.
> > >
> > > To go beyond surface similarity, we additionally report results using **Summac**, which explicitly targets factual consistency and reasoning alignment between model explanations and reference rationales. We compare **Cosmos-Eval** with **Qwen-2.5-VL-7B**, **InternVL** variants, **Cosmos-Reason1**, and **VideoLLaMA 3-7B** on both SA and PC:
> > >
> > > **SA:**
> > >
> > > | Model | Cosmos-Eval | Qwen-2.5-VL-7B | InternVL-8B | InternVL-9B | InternVL-14B | Cosmos-Reason1 | VideoLLaMA 3-7B |
> > > | :---- | :---- | :---- | :---- | :---- | :---- | :---- | :---- |
> > > | **summer\_c** | 26.62 | 21.50 | 23.92 | 24.22 | 23.91 | 21.23 | 22.56 |
> > >
> > > **PC:**
> > >
> > > | Model | Cosmos-Eval | Qwen-2.5-VL-7B | InternVL-8B | InternVL-9B | InternVL-14B | Cosmos-Reason1 | VideoLLaMA 3-7B |
> > > | :---- | :---- | :---- | :---- | :---- | :---- | :---- | :---- |
> > > | **summer\_c** | 23.32 | 22.69 | 23.07 | 22.86 | 23.16 | 22.25 | 23.20 |
> > >
> > > **Key observations.**
> > >
> > > 1. On **SA**, Cosmos-Eval attains the highest Summac score among all compared models, suggesting that its explanations are not only textually similar to references (BLEU/BERTScore), but also better aligned in terms of factual and reasoning consistency.
> > > 2. On **PC**, Cosmos-Eval also performs competitively, slightly above or on par with other strong baselines and clearly above Qwen-2.5-VL-7B and Cosmos-Reason1, again indicating that the physics-focused rationales are factually coherent with the reference explanations.
> > >
> > > In addition, as discussed in **Weakness 1**, we conduct a **human evaluation** where annotators rate explanations from **Cosmos-Eval**, **Qwen-2.5-VL-7B**, **GPT-4V**, and **Gemini-2.5-Pro** along fine-grained SA/PC rubric dimensions (grounding, temporal alignment, consistency, justification, coverage). Cosmos-Eval is consistently preferred by human raters on both SA and PC, especially for PC.
> > >
> > > Taken together, these results show that our evaluation of Cosmos-Eval does not rely solely on surface-level textual metrics. The combination of **Summac** (for factual/reasoning consistency) and **human rubric-based evaluation** provides a more faithful picture of reasoning accuracy and factual soundness, and we appreciate the reviewer’s comment for prompting us to make this aspect clearer.

---

> > > > ### Author Response · Authors · 2025-11-21
> > > >
> > > > # W4: Response to W4 (Part 1): Novelty of Cosmos-Eval
> > > >
> > > > We thank the reviewer for this constructive comment about both the **novelty** and the **amount of work** in the paper. We understand the concern that, since we build on top of VIDEOPHY-2 and use existing VLMs, our work might look like “just prompting a VLM to explain scores.” Below we first clarify where we see the main conceptual contributions in terms of novelty; in a separate reply we will summarize the amount of work and evaluation effort.
> > > >
> > > > ### 1. Regarding Novelty
> > > >
> > > > **(a) From scalar scores to an explainable SA/PC evaluation paradigm**
> > > >
> > > > VIDEOPHY-2 and related benchmarks provide only *numeric* SA/PC scores. Cosmos-Eval, by design, changes the evaluation objective: every test case is mapped to a **5-point SA/PC score plus a concise, physics-grounded rationale**. The rationales are organized around a physical ontology and semantic mismatches, which supports:
> > > >
> > > > - targeted **diagnosis** of model failures,
> > > > - ablations and debugging of T2V models,
> > > > - more transparent auditing of evaluation outcomes.
> > > >
> > > > To our knowledge, there is no prior T2V evaluation framework that couples **calibrated SA/PC scoring** with **structured, physics-aware rationales** as the *default* output rather than as a one-off analysis.
> > > >
> > > > **(b) Multi-stage teacher–student pipeline vs. one-shot prompting**
> > > >
> > > > We agree that simply “asking a VLM to explain VIDEOPHY-2 scores” would not be very novel. Our method is deliberately more structured:
> > > >
> > > > - **Stage 0**: a score head produces 5-point SA/PC labels (anchored on VideoPhy-2 human labels and VideoPhy-2-AutoEval).
> > > > - **Stage 1**: an ensemble of VLM calls generates **reference-seeded rationales** for each score, aggregating multiple samples to reduce noise.
> > > > - **Stage 2**: a **controller with an LLM judge** iteratively verifies and refines rationales against a rubric (pass/fail, backtracking).
> > > > - **Stage 3**: a **two-run SFT** student is trained: first to predict the score, then to generate `<think>/<answer>` **conditioned on that score**, distilling the heavy teacher pipeline into a single lightweight evaluator at inference time.
> > > >
> > > > This pipeline is not equivalent to changing a prompt once. Our ablations (Stage-0/1/2 ablations and single-pass vs. two-run SFT) show:
> > > >
> > > > - removing Stage 0 hurts calibration,
> > > > - skipping Stage 2 increases hallucinated or rubric-violating rationales,
> > > > - single-pass “reason-only” SFT gives good-looking explanations but **very poor** score agreement,
> > > > - “score-only” SFT improves scores but **collapses** rationale quality.
> > > >
> > > > In short, naive one-shot prompting or a single SFT pass cannot simultaneously match human scores **and** produce high-quality rationales; the gains come from the full multi-stage design.
> > > >
> > > > **(c) Formal analysis of the multi-stage framework (Appendix F)**
> > > >
> > > > Many evaluation papers stop at empirical pipelines. In contrast, we provide a **formal analysis in Appendix F**:
> > > >
> > > > - Stage 1 is modeled as **consensus aggregation** that reduces label noise compared to independent single-sample rationales.
> > > > - Stage 2 is treated as **controlled generation**, where the controller–judge loop filters noisy or rubric-violating explanations.
> > > > - We derive conditions under which Stage 1 + Stage 2 yield a **supervision signal with lower effective noise rate** than a one-shot end-to-end (E2E) approach, leading to a **tighter generalization bound** for the student (Stage 3).
> > > >
> > > > This analysis is tailored to our pipeline and explains *why* the staged design can, in principle, outperform direct E2E SFT on noisy explanations. To our knowledge, such an explicit analysis of a multi-stage, LLM-based evaluator is not present in prior T2V evaluation work.
> > > >
> > > > **(d) Beyond a single benchmark**
> > > >
> > > > We agree that if Cosmos-Eval only worked on VIDEOPHY-2, the contribution would look narrow. To address this, we:
> > > >
> > > > - Evaluate on **AIGVE-Bench** and an **LG-VQA-style** dataset and show that Cosmos-Eval **generalizes well across independent suites**. In terms of correlation to human scores, it **tracks the official VideoPhy-2-AutoEval VLM-as-judge**, which is a strong baseline on these datasets. Crucially, however, **VideoPhy-2-AutoEval only outputs scores**, whereas Cosmos-Eval provides **both comparable scores and physics-grounded rationales**.
> > > > - Conduct long-horizon experiments to study stability over extended content; Cosmos-Eval maintains stable SA/PC behavior across time and behaves comparably to VideoPhy-2-AutoEval on long clips.
> > > >
> > > > Taken together, we see Cosmos-Eval as a **generalizable, explainable SA/PC evaluation framework**, rather than a thin prompting layer around a single benchmark.

---

> > > > > ### Author Response · Authors · 2025-11-21
> > > > >
> > > > > # W4:Response to W4 (Part 2): Amount of Work and Evaluation Effort
> > > > >
> > > > > We also understand the concern that, if this were just “VIDEOPHY-2 \+ one VLM prompt,” the workload would be limited. Here we summarize the main pieces of work we have actually carried out on data construction, modeling, theory, and evaluation.
> > > > >
> > > > > ### 2\. Regarding the Amount of Work
> > > > >
> > > > > **(a) Data construction and setup**
> > > > >
> > > > > - We start from **VideoPhy \+ VideoPhy-2** but go beyond them by creating a large-scale **“scores \+ rationales” corpus** via the multi-stage teacher pipeline.
> > > > > - We integrate additional benchmarks (**AIGVE-Bench**, LG-VQA-style) and long-video setups, so that Cosmos-Eval is tested beyond its training distribution.
> > > > > - We design detailed **rubrics and prompt templates** (for SA/PC, human/LLM-as-judge) that enable consistent, multi-dimensional evaluation of explanations.
> > > > >
> > > > > **(b) Human and multi-judge validation**
> > > > >
> > > > > - We conduct a **human evaluation** (1,500 questions) where annotators rate SA/PC rationales from Cosmos-Eval, Qwen3-VL-plus, and GPT-4V along fine-grained dimensions (grounding, temporal alignment, consistency, justification, coverage). Cosmos-Eval is consistently preferred, especially on PC.
> > > > > - We further evaluate explanations using multiple **independent VLM judges** (Qwen3-VL-plus, GPT-4V, Gemini-2.5-Pro), which again show Cosmos-Eval to be competitive or superior, particularly in physics reasoning.
> > > > > - We report not only BLEU/BERTScore, but also **Summac-style factual/consistency metrics**, and cross-benchmark correlations to human SA/PC scores.
> > > > >
> > > > > These steps are meant to ensure that Cosmos-Eval is not just evaluated “from within its own loop,” but against humans and diverse judges.
> > > > >
> > > > > **(c) Systematic evaluation of T2V systems and ongoing maintenance**
> > > > >
> > > > > - We use Cosmos-Eval to systematically **evaluate and rank leading T2V models**, including strong open-source and commercial systems, forming an evaluation toolchain and a set of results that other researchers can reuse.
> > > > > - Looking ahead, we plan to treat Cosmos-Eval as a **maintained evaluation framework**: we are already collecting human feedback via our web interface, and we are preparing a **small curated set of human-written rationales** for difficult SA/PC cases, to further calibrate and diversify the student’s explanation style over time.
> > > > >
> > > > > ---
> > > > >
> > > > > In summary, we fully respect the reviewer’s concern about incremental contributions. Our intent is not to propose a new generator, but to take a **principled, multi-stage, and explainable approach to SA/PC evaluation** that combines:
> > > > >
> > > > > 1. an *explanation-centric* evaluation paradigm,
> > > > > 2. a non-trivial multi-stage teacher–student pipeline with theoretical justification,
> > > > > 3. and extensive empirical & human validation across datasets, judges, and settings.
> > > > >
> > > > > We hope this split response makes it clearer why we view Cosmos-Eval as going beyond a simple prompting layer on VIDEOPHY-2, and we sincerely appreciate the reviewer’s willingness to reconsider their score if the novelty and workload are reasonably addressed.

---

> > > > > > ### Author Response · Authors · 2025-11-21
> > > > > >
> > > > > > # Q1:“consensus extraction” step
> > > > > >
> > > > > > We appreciate the reviewer for asking about the “consensus extraction” step using the aggregator LLM to combine outputs from multiple VLMs. To clarify, the process of extracting consensus is handled by the **LLM** through a series of steps that ensure consistency and address any disagreements between the generated explanations.
> > > > > >
> > > > > > In particular, we use **semantic parsing** and **bidirectional entailment** to compare the reasoning from different VLMs. If there are contradictions or disagreements between the generated explanations, the LLM is instructed to return an error. In cases where the reasons are completely contradictory, the sample is discarded.
> > > > > >
> > > > > > The specific reasoning steps for **consensus extraction** are as follows, and the corresponding prompts used in our framework are shown in **Figure 22** of the paper:
> > > > > >
> > > > > > ### **Reasoning Steps (Execute Strictly):**
> > > > > >
> > > > > > 1. **Semantic Parsing**: Extract core claims and negation relationships from each analysis.
> > > > > > 2. **Proposition Decomposition**: Break each analysis into minimal verifiable proposition units.
> > > > > > 3. **Bidirectional Entailment Check**: For each proposition unit, verify: a) Analysis 1 entails this proposition in Analysis 2
> > > > > >    b) Analysis 2 entails this proposition in Analysis 1
> > > > > > 4. **Common Proposition Filtering**: Retain only propositions that pass bidirectional entailment.
> > > > > > 5. **Evidence Fusion**: Integrate video evidence supporting common propositions from both analyses.
> > > > > > 6. **Contradiction Detection**: Check for any logical contradictions.
> > > > > >
> > > > > > As shown in **Figure 26(page 43\)** of the paper, the **LLM prompts** used for this process guide the model to carry out each of the above steps, ensuring the accurate and consistent extraction of the rationale. These prompts are integral to generating consistent, human-aligned reasoning and filtering out contradictory explanations.

---

> > > > > > > ### Author Response · Authors · 2025-11-21
> > > > > > >
> > > > > > > # Q2:As several closed-source VLMs (e.g., GPT-4V, Claude, Gemini) are involved as generators and judges, could the authors provide direct or comparative examples showing how Cosmos-Eval’s explanations differ from those produced by these models?
> > > > > > >
> > > > > > > We thank the reviewer for raising the concern about how **Cosmos-Eval**’s explanations differ from those produced by other closed-source VLMs (e.g., **GPT-4V**, **Claude**, **Gemini**) involved as both generators and judges.
> > > > > > >
> > > > > > > This concern is responded in **Weakness 2** of our response, where we have already provided a detailed comparison of the explanations generated by **Cosmos-Eval**, **Qwen3-VL-plus**, and **GPT-4V**. Additionally, we included **comparative examples** showing the differences in how **Cosmos-Eval**'s explanations align with those produced by these large models.
> > > > > > >
> > > > > > > In **Weakness 2**, we present the results of human evaluations and how **Cosmos-Eval** consistently outperforms other models, particularly in terms of consistency and alignment with human expectations. This includes direct comparisons of explanations on **SA** and **PC** tasks.
> > > > > > >
> > > > > > > We encourage the reviewer to refer to **Weakness 2** for a more detailed analysis and comparison of these explanations.

---

> > > > > > > > ### Author Response · Authors · 2025-11-21
> > > > > > > >
> > > > > > > > # Q3:Beyond BLEU and BERTScore, did the authors conduct or consider any human evaluation or factual-consistency analysis to assess the correctness of the generated rationales?
> > > > > > > >
> > > > > > > > We thank the reviewer for raising the concern regarding the assessment of the correctness of the generated rationales, beyond traditional metrics like **BLEU** and **BERTScore**.
> > > > > > > >
> > > > > > > > This concern is addressed in **Weakness 3** of our response, where we describe the extensive **human evaluation** and **factual-consistency analysis** that we conducted. In this section, we performed a **human evaluation** to assess the **factual correctness** and **reasoning quality** of the explanations generated by **Cosmos-Eval**, **Qwen3-VL-plus**, and **GPT-4V**. This evaluation involved multiple **dimensions** and **criteria**, ensuring a thorough analysis of the reasoning accuracy.
> > > > > > > >
> > > > > > > > Additionally, we incorporated **Summac**, a factual-consistency metric, to quantify and compare the accuracy of the rationales. The results of these evaluations demonstrate that **Cosmos-Eval** outperforms other models in terms of factual consistency and reasoning accuracy, providing a more reliable and interpretable evaluation of the generated rationales.
> > > > > > > >
> > > > > > > > We encourage the reviewer to refer to **Weakness 3** for a more detailed description of the evaluation methodology and results.

---

> ### Author Response · Authors · 2025-11-27
>
> Dear Reviewer HYuZ:
>
> We are grateful for your mandatory acknowedgement during this busy period. As the rebuttal period is ending soon, please be free to let us know whether your concerns have been addressed or not, and if there are any further questions.
>
> Thank you for your time and effort in reviewing our paper,
>
> Authors.

---

### Official Review · Reviewer_JK2k · 2025-10-31

**Soundness:** 2
**Presentation:** 3
**Contribution:** 3
**Rating:** 4
**Confidence:** 4

**Summary:**

Summary:
This paper addresses a critical limitation of existing text-to-video (T2V) model evaluators: they only output scalar scores for Semantic Adherence (SA, prompt-video alignment) and Physical Commonsense (PC, adherence to physical laws) without explaining the reasoning behind scores. To solve this, the authors proposeCosmos-Eval, an explainable evaluation framework that combines 5-point SA/PC scores with physics-grounded natural-language rationales.

Contributions:
（1）Explainable SA/PC Evaluation Paradigm: Cosmos-Eval is the first framework to pair fine-grained 5-point SA/PC scores with actionable, physics-grounded rationales, resolving the interpretability gap of score-only evaluators and enabling targeted diagnosis of model failures.
（2）Score-Aligned Rationale Generation: By integrating Cosmos-Reason1’s physical ontology and a judge-verified CoT controller, it ensures rationales are tightly linked to scores—avoiding the accuracy-interpretability trade-off seen in generic VLM-based evaluators.
（3）Generalizable, Distillable Pipeline: Its four-stage workflow is dataset- and scorer-agnostic, making it adaptable to other T2V evaluation suites beyond VIDEOPHY-2.
（4）State-of-the-Art Rationale Quality: Cosmos-Eval outperforms generic VLMs in rationale similarity to human-like references, with best-in-class BERTScore F1 and BLEU-4 scores.

**Strengths:**

Strengths:
（1）Originality
It proposes Cosmos-Eval, the first T2V evaluation framework pairing 5-point SA/PC scores with physics-grounded rationales to solve the interpretability gap of score-only tools. Its four-stage pipeline and two-run fine-tuning innovatively balance score alignment, rationale quality, and deployment efficiency.
（2）Quality
Its pipeline follows a rigorous, reproducible process, with each stage validated to ensure reliability. Experiments cover 6 T2V models and 9000+ videos, with metrics matching top auto-evaluators in score alignment and outperforming VLMs in rationale quality.
（3）Clarity
It follows a clear "gap→solution→validation" structure, explaining technical details in plain language. Consistent terminology and visual aids help readers easily understand the framework’s design and results.
（4）Significance
As an open, standardized tool, it enables T2V model failure diagnosis and guides real-world applications. It shifts T2V evaluation to prioritize interpretability, aligns with scientific integrity, and reduces educational misinformation risks.

**Weaknesses:**

Weaknesses:
（1）The comparison with the methods of predecessors is not sufficient.
A comparison between the dataset and some previous related datasets, such as VideoREPA, WISA, NewtonGen, etc.?
（2）LLM Self-Reinforcement Risk
It heavily relies on LLMs for rationale generation and evaluation, risking bias toward LLM-style explanations over human needs. The paper does not address this closed-loop issue or propose human-in-the-loop calibration.
（3）Limited Video Complexity and Duration Coverage
Validated mainly on VIDEOPHY-2, it lacks testing on long-duration videos or complex physical scenarios, failing to assess escalated inconsistencies in extended content.
（4）Dependence on Predecessor Tools and Narrow Data
It relies on frozen VIDEOPHY-2-AUTOEVAL for initial scores, inheriting its limitations. Training and evaluation are restricted to one benchmark, with no adaptability tests for diverse suites or non-English prompts.
（5）Insufficient Computational Efficiency Analysis
While claiming a lightweight distilled model, it provides no detailed compute cost breakdown for pipeline stages. No inference speed or resource comparison with score-only evaluators is given.
（6）Noisy Input Robustness Gap
It does not test performance on noisy T2V outputs (e.g., motion blur, occlusion), leaving reliability for real-world imperfect videos unconfirmed.

**Questions:**

Questions:
（1）Questions About LLM Self-Reinforcement Bias
Your framework relies heavily on LLMs for both generating reference rationales and evaluating final quality, which may create a self-reinforcement loop favoring LLM-style outputs over human-aligned explanations. Have you conducted comparative studies to measure how much this bias deviates from genuine human interpretability needs? Additionally, do you have plans to integrate human-in-the-loop calibration to mitigate such closed-loop dependency in future iterations?
（2）Generalization to Complex Video Scenarios
Cosmos-Eval is primarily validated on the VIDEOPHY-2 benchmark, which lacks long-duration videos and complex physical interactions. Have you tested its performance on extended video content where physical inconsistencies typically escalate? If not, what technical challenges do you anticipate in adapting your trajectory extraction and rationale generation modules for such scenarios?
（3）Suggestions for Reducing Dependence on Predecessor Tools
Your framework inherits initial scores from frozen VIDEOPHY-2-AUTOEVAL without independent validation, which may propagate its predecessor’s limitations. We suggest adding a parallel validation step: compare scores from VIDEOPHY-2-AUTOEVAL with human-labeled ground truth for a subset of samples to quantify inherited biases. This would enhance trust in your pipeline’s foundational score accuracy.
If the author can effectively solve my doubts, I will consider improving my score.

---

> ### Author Response · Authors · 2025-11-21
>
> # W1:The comparison with the methods of predecessors is not sufficient.
>
> We thank the reviewer for pointing out the need for a more explicit comparison with prior work in the physics-aware T2V domain. We have updated the manuscript to include a more comprehensive discussion of related work and relevant benchmarks.
>
> (i) **Positioning w.r.t. WISA, NewtonGen, and VideoREPA**: The proposed Cosmos-Eval framework is designed as an evaluation tool, whereas WISA, NewtonGen, and VideoREPA are generation frameworks. Specifically, WISA injects structured physical attributes into T2V generation, NewtonGen uses learned dynamics for controlling motion, and VideoREPA distills physics knowledge into T2V models. In contrast, Cosmos-Eval evaluates the output of any T2V model, focusing on providing Semantic Adherence (SA) and Physical Commonsense (PC) scores, along with explainable, physics-grounded rationales. Thus, while these generation frameworks target the creation of videos, Cosmos-Eval focuses on assessing and explaining the quality of these videos.
>
> (ii) **Cross-benchmark evidence**: In response to concerns about the breadth of our data, we have additionally evaluated Cosmos-Eval on two independent benchmarks: AIGVE-Bench and LG-VQA-style. On the **AIGVE-Bench**, Cosmos-Eval demonstrates performance similar to VideoPhy-2-AutoEval, outperforming Qwen-2.5-VL-7B. The results are as follows:
>
> **AIGVE-Bench Results:**
>
> | Model | Pearson Correlation (r) | 95% CI |
> | :---- | :---- | :---- |
> | Cosmos-Eval | 0.1986 | \[0.1561, 0.2326\] |
> | VideoPhy2 | 0.2089 | \[0.1706, 0.2466\] |
> | Qwen2.5-VL-7B | 0.1033 | \[0.0063, 0.1425\] |
>
> We also evaluated **LG-VQA-style** with 3,276 clips, obtaining the following Pearson correlation results:
>
> **LG-VQA-style Results:**
>
> | Model | Pearson Correlation (r) | 95% CI |
> | :---- | :---- | :---- |
> | Cosmos-Eval | 0.2759 | \[0.2414, 0.3097\] |
> | VideoPhy-2-AutoEval | 0.2750 | \[0.2404, 0.3088\] |
> | Qwen-2.5-VL-7B | 0.2013 | \[0.1656, 0.2366\] |
>
> Notably, whereas VideoPhy-2-AutoEval outputs only a numeric score, Cosmos-Eval augments this with a concise, evidence-grounded rationale. For example, for a video depicting a red ball floating in midair, Cosmos-Eval not only assigns a low PC score, but also explicitly explains that the scene violates basic gravitational laws. This additional interpretability enables more transparent and fine-grained error analysis, a capability that is not supported by the purely score-based evaluation of VideoPhy-2-AutoEval.

---

> > ### Author Response · Authors · 2025-11-21
> >
> > # W2:LLM Self-Reinforcement Risk
> >
> > We thank the reviewer for this very important comment on LLM self-reinforcement and the risk of drifting toward “LLM-style” explanations rather than what humans actually need. Our initial submission clearly did not spell this out well enough, and we appreciate the opportunity to clarify both what we have already done and how we think about this issue.
> >
> > To reduce the risk of relying purely on LLM feedback, we **have already conducted** a dedicated human evaluation of the rationales. Concretely, we built a small website to collect human judgments on SA and PC explanations produced by **Cosmos-Eval**, **Qwen3-VL-plus**, and **GPT-4V**. We designed 5-point evaluation questions along the same dimensions as our rationale rubrics (Tables 7 and 8 in the paper), and instantiated a total of 1,500 questions across SA and PC. Each dimension was scored by annotators on a \[0, 0.5, 1\] scale.
> >
> > The study followed a **double-blind** protocol: evaluators were randomly assigned instances with model identities hidden. Annotators only saw the video, the caption (for SA), and a candidate explanation, and then rated each dimension independently. The aggregated results are:
> >
> > **SA human evaluation**
> >
> > | model | Grounding | Temporal Alignment | Consistency | Alignment Justification | Coverage & Specificity | Total Average |
> > | :---- | :---- | :---- | :---- | :---- | :---- | :---- |
> > | Cosmos-Eval | 0.82 | 0.57 | 0.71 | 0.81 | 0.87 | 0.76 |
> > | GPT-4V | 0.64 | 0.52 | 0.67 | 0.61 | 0.64 | 0.62 |
> > | Qwen3-VL-plus | 0.73 | 0.53 | 0.69 | 0.73 | 0.71 | 0.69 |
> >
> > **PC human evaluation**
> >
> > | model | Grounding | Temporal | Consistency | Criteria & Justification | Video Quality | Total Average |
> > | :---- | :---- | :---- | :---- | :---- | :---- | :---- |
> > | Cosmos-Eval | 0.79 | 0.56 | 0.82 | 0.85 | 0.82 | 0.77 |
> > | GPT-4V | 0.64 | 0.52 | 0.56 | 0.64 | 0.56 | 0.58 |
> > | Qwen3-VL-plus | 0.59 | 0.51 | 0.59 | 0.64 | 0.63 | 0.60 |
> >
> > These numbers are not meant to claim that the closed-loop problem is completely solved, but they do show that, along human-defined criteria (grounding, temporal alignment, justification, coverage, etc.), human raters **consistently prefer Cosmos-Eval’s explanations**, especially for PC. In other words, the rationales are not only optimized to look good to another LLM; they are also judged by humans to be more grounded, consistent, and specific than those from two strong VLM baselines. We apologize that this human evaluation was not explained clearly enough in the initial submission.
> >
> > Regarding the **closed-loop risk in training**, our design also tries to avoid a purely self-referential LLM pipeline. We are sorry that this part of the setup was not clearly described in the first version of the paper; it was already in place in our original experiments, not something added only after the reviews. In **Stage 0**, we explicitly mix human and model supervision:
> >
> > - For **VideoPhy-2**, we use the **human 5-point SA/PC scores** as the primary anchor for calibration.
> > - For **VideoPhy**, which does not have 5-point SA/PC labels, we use **VideoPhy-2-AutoEval** only to place those clips on the same 1–5 scale, and these pseudo-labels are always used together with genuine human labels from VideoPhy-2.
> >
> > So the student model is never trained solely on LLM-judged scores; human annotations remain the anchor for the scale. In addition, because the original VideoPhy benchmark uses a different annotation scheme and score distribution from VideoPhy-2, we **deliberately avoid** a naive linear remapping from its original scores to the 1–5 SA/PC scale: in our preliminary checks this led to noticeable bias and scale distortion. Instead, we stabilize the 1–5 scale using VideoPhy-2 human labels and then rely on VideoPhy-2-AutoEval only when human labels are unavailable, which helps extend coverage while keeping the calibration tied to human annotations.
> >
> > At the same time, we agree with the reviewer that there is still room to further strengthen human-in-the-loop calibration. Beyond the human study already in place, a natural next step for Cosmos-Eval is to:
> >
> > - **Collect more human feedback** on difficult or borderline cases (especially where LLM judges and humans disagree) and use these examples to refine the evaluator.
> > - Introduce a **small curated set of human-written rationales** for challenging SA/PC scenarios, so that the model has explicit human references for both style and content and is less likely to drift toward purely “LLM-styled” but unhelpful explanations.
> >
> > More broadly, we view Cosmos-Eval as a **long-term maintained evaluation framework** rather than a one-off model. As we continue to gather human feedback and human-written rationales, we plan to fold these signals back into future updates of Cosmos-Eval, so that the system becomes progressively less self-referential and more closely aligned with how humans themselves would explain and judge T2V outputs.

---

> > > ### Author Response · Authors · 2025-11-21
> > >
> > > # W3:Limited Video Complexity and Duration Coverage
> > >
> > > We thank the reviewer for highlighting the gap between short benchmark clips and real-world long, complex videos. This is an important concern, and we would like to respond in two ways: (i) by clarifying the inherent complexity of the underlying benchmark, and (ii) by presenting a dedicated long-horizon study on extended videos.
> > >
> > > (i) **Complexity of the base benchmark (VideoPhy-2).**
> > > Although VideoPhy-2 consists of 4–6s clips, it was explicitly designed as an action-centric, physics-rich benchmark rather than a collection of simple, low-motion scenes. As described in Bansal et al. (VideoPhy-2, 2025), it covers 197 actions (54 object–interaction actions and 143 physical/sports activities), including a “hard subset” with challenging physical scenarios such as momentum transfer (e.g., throwing a discus), state changes (e.g., breaking or deforming objects), balancing (e.g., tightrope walking), and complex articulated motions (e.g., backflips). Even though each clip is short, Cosmos-Eval is trained and evaluated on multi-event, physically non-trivial situations rather than only toy examples.
> > >
> > > (ii) **Long-horizon study with LongCat-Video (Cosmos-Eval vs. VideoPhy-2-AutoEval).**
> > > To more directly probe performance on longer, more complex content, we conducted an additional experiment on **LongCat-Video**, which produces ≈33s videos with multi-step motions and evolving scenes.
> > >
> > > - We sampled 30 prompts and generated ≈33s videos.
> > > - Each video was evaluated in two ways:
> > >   (a) as a whole (full ≈33s clip), and
> > >   (b) uniformly split into 11 non-overlapping 3s segments, each scored separately.
> > >
> > > For **Cosmos-Eval**, segment-wise scores remained stable across the entire temporal span:
> > >
> > > - SA segment means stayed in a narrow band of roughly **3.03–3.13**,
> > > - PC segment means in roughly **3.97–4.13**.
> > >
> > > When run on the full ≈33s videos without splitting, Cosmos-Eval produced average scores of:
> > >
> > > - **SA ≈ 3.30**,
> > > - **PC ≈ 4.00**.
> > >
> > > For **VideoPhy-2-AutoEval**, we observed a similar pattern of temporal stability on the same long videos:
> > >
> > > - Segment-wise SA means stayed between roughly **2.73–3.00**,
> > > - Segment-wise PC means stayed between roughly **3.83–4.07**.
> > >   On the full ≈33s clips, its average scores were:
> > > - **SA ≈ 2.97**,
> > > - **PC ≈ 3.50**.
> > >
> > > Taken together, these observations suggest that (a) Cosmos-Eval behaves comparably to the original VideoPhy-2 scorer on long, multi-step videos, and (b) neither SA nor PC exhibits systematic drift as we move from early to late segments, even when inconsistencies can accumulate over time.

---

> > > > ### Author Response · Authors · 2025-11-21
> > > >
> > > > # W4:Dependence on Predecessor Tools and Narrow Data
> > > >
> > > > We thank the reviewer for the thoughtful comment regarding our dependence on **VIDEOPHY-2-AUTOEVAL** and the potential narrowness of training/evaluation data. We agree that this is an important issue and we appreciate the opportunity to clarify what Cosmos-Eval currently does beyond the original VideoPhy-2 setup.
> > > >
> > > > (i) **Cross-benchmark evidence (beyond VideoPhy-2)**
> > > > To examine whether Cosmos-Eval overfits to a single benchmark, we evaluated it on two additional and independently curated suites, **AIGVE-Bench** and **LG-VQA-style**, without any further fine-tuning on these datasets. On **AIGVE-Bench**, which aggregates videos from multiple state-of-the-art T2V models, Cosmos-Eval achieves performance comparable to VideoPhy-2-AutoEval and clearly surpasses Qwen-2.5-VL-7B:
> > > >
> > > > **AIGVE-Bench Results:**
> > > >
> > > > | Model | Pearson Correlation (r) | 95% CI |
> > > > | :---- | :---- | :---- |
> > > > | Cosmos-Eval | 0.1986 | \[0.1561, 0.2326\] |
> > > > | VideoPhy2 | 0.2089 | \[0.1706, 0.2466\] |
> > > > | Qwen2.5-VL-7B | 0.1033 | \[0.0063, 0.1425\] |
> > > >
> > > > On the **LG-VQA-style** dataset with 3,276 clips, Cosmos-Eval again tracks VideoPhy-2-AutoEval closely while outperforming Qwen-2.5-VL-7B:
> > > >
> > > > **LG-VQA-style Results:**
> > > >
> > > > | Model | Pearson Correlation (r) | 95% CI |
> > > > | :---- | :---- | :---- |
> > > > | Cosmos-Eval | 0.2759 | \[0.2414, 0.3097\] |
> > > > | VideoPhy-2-AutoEval | 0.2750 | \[0.2404, 0.3088\] |
> > > > | Qwen-2.5-VL-7B | 0.2013 | \[0.1656, 0.2366\] |
> > > >
> > > > These results suggest that, despite being trained on VideoPhy/VideoPhy-2, Cosmos-Eval generalizes reasonably well to independent evaluation suites and does not simply memorize the quirks of a single benchmark. At the same time, we fully acknowledge that our current experiments are still limited to a specific family of physics/semantics datasets and that broader coverage (e.g., more diverse domains or generators) would further strengthen this claim.
> > > >
> > > > (ii) **Dependence on VideoPhy-2-AutoEval and inherited limitations**
> > > > We agree that using **VIDEOPHY-2-AUTOEVAL** as part of our pipeline inevitably means we inherit some of its biases. Our intention is to use it in a controlled way rather than as a sole oracle. Concretely, human 5-point SA/PC labels from **VideoPhy-2** serve as the primary anchor for score calibration. For **VideoPhy**, which does not provide 5-point labels, we use VideoPhy-2-AutoEval only to map those clips onto the same 1–5 scale so that the training corpus is consistent. Thus, Cosmos-Eval is trained against a mixture of human labels (VideoPhy-2) and VLM-as-judge labels (VideoPhy via AutoEval), instead of being purely self-referential. We see Cosmos-Eval as complementary to VideoPhy-2-AutoEval: it roughly matches its score alignment while adding structured, physics-grounded rationales and human-evaluated explanation quality, rather than trying to replace it as a completely independent scorer in one step.
> > > >
> > > > (iii) **Non-English prompts and more diverse data**
> > > > The reviewer is also correct that our current experiments focus on English prompts and captions. We have not yet systematically evaluated Cosmos-Eval on non-English prompts or multilingual datasets, and we consider this a real limitation of the present work. In principle, the PC branch of our framework (which conditions only on the video) is less language-dependent, but the SA task and the rubrics are currently designed around English text. We view extending Cosmos-Eval to multilingual and more stylistically diverse prompts as a natural next direction, and we are particularly interested in testing whether the current ontology and rubrics remain adequate or need to be adapted for other languages and cultural contexts.

---

> > > > > ### Author Response · Authors · 2025-11-21
> > > > >
> > > > > # W5:Insufficient Computational Efficiency Analysis
> > > > >
> > > > > We thank the reviewer for raising the concern about the missing computational efficiency analysis. This is a very reasonable point for an evaluation framework and also prompts us to clarify which parts of Cosmos-Eval are computationally expensive and what users actually pay for in practice. In our design, the multi-stage teacher pipeline (Stage 0–2) is only run **offline** by us and subsequently **distilled via SFT into a single Cosmos-Eval model**. At inference time, end users only execute this distilled model, rather than the full teacher pipeline.
> > > > >
> > > > > During training, we first run a multi-stage teacher pipeline: Stage 0 does pre-processing and score normalization, Stage 1 uses large VLMs to generate candidate rationales, and Stage 2 is a judge-verified controller that filters and backtracks according to task-specific rubrics. Based on our logs on 200 samples, the per-sample cost of this teacher pipeline is roughly:
> > > > >
> > > > > **Stage-wise cost for PC (Stage 0–2, 200 samples)**
> > > > >
> > > > > | Step | GPU Count | Inference Time | Sample Count | Average Time / Sample |
> > > > > | :---- | :---- | :---- | :---- | ----: |
> > > > > | Stage 0 (pre-processing) | 1 | 2m12s | 200 | 0.66 s |
> > > > > | Stage 1: Qwen Gen (run 1\) | 1 | 18m43s | 200 | 19.38 s |
> > > > > | Stage 1: Qwen Gen (run 2\) | 2 | 18m37s | 200 | 29.25 s |
> > > > > | Stage 2 (reasoning controller) | 2 | 188m48s | 200 | 56.64 s |
> > > > >
> > > > > Total ≈ 105.93 s per PC sample for Stage 0–2.
> > > > >
> > > > > **Stage-wise cost for SA (Stage 0–2, 200 samples)**
> > > > >
> > > > > | Step | GPU Count | Inference Time | Sample Count | Average Time / Sample |
> > > > > | :---- | :---- | :---- | :---- | ----: |
> > > > > | Stage 0 (pre-processing) | 1 | 2m14s | 200 | 0.67 s |
> > > > > | Stage 1: Qwen Gen | 4 | 54m43s | 200 | 16.31 s |
> > > > > | Stage 1: Tarsier Gen | 4 | 18m19s | 200 | 20.78 s |
> > > > > | Stage 1: Qwen3 merge | 2 | 17m39s | 200 | 5.14 s |
> > > > > | Stage 2 (complex reasoning ctrl.) | 4 | 340m12s | 200 | 102.06 s |
> > > > >
> > > > > Total ≈ 144.96 s per SA sample for Stage 0–2.
> > > > >
> > > > > We fully agree that this teacher pipeline is computationally expensive, which is exactly why we distill it into a single student model (Stage 3). Once distillation is done, neither we nor end users need to re-run Stage 0–2 to evaluate new videos.
> > > > >
> > > > > For adoption, the key quantity is therefore the **inference-time cost of Cosmos-Eval**, compared with existing evaluators and a strong 7B VLM. On the same 200-sample set, we measure the following for **scores only**:
> > > > >
> > > > > **Score-only inference (PC/SA scores)**
> > > > >
> > > > > | Step | GPU Count | Inference Time | Sample Count | Avg Time / Sample | GPU-Hours |
> > > > > | :---- | :---- | :---- | :---- | ----: | ----: |
> > > > > | Videophy2-AutoEval-PC | 1 | 2m12s | 200 | 0.66 s | 0.0368 |
> > > > > | Videophy2-AutoEval-SA | 1 | 2m14s | 200 | 0.67 s | 0.0372 |
> > > > > | **Cosmos-Eval (PC-score)** | 1 | 3m42s | 200 | **1.11 s** | 0.0618 |
> > > > > | **Cosmos-Eval (SA-score)** | 1 | 3m42s | 200 | **1.11 s** | 0.0619 |
> > > > > | Qwen-2.5-VL-7B (PC-score) | 1 | 4m20s | 200 | 1.30 s | 0.0722 |
> > > > > | Qwen-2.5-VL-7B (SA-score) | 1 | 4m20s | 200 | 1.30 s | 0.0724 |
> > > > >
> > > > > So for score prediction, Cosmos-Eval is slightly slower than the specialized score-only evaluator (VideoPhy2-AutoEval), but in the same range as a 7B VLM, while being designed to also support explanation generation.
> > > > >
> > > > > When we also request rationales, we obtain:
> > > > >
> > > > > **Rationale generation (PC/SA reasons)**
> > > > >
> > > > > | Step | GPU Count | Inference Time | Sample Count | Avg Time / Sample | GPU-Hours |
> > > > > | :---- | :---- | :---- | :---- | ----: | ----: |
> > > > > | **Cosmos-Eval (PC-reason)** | 1 | 19m34s | 200 | **5.87 s** | 0.3272 |
> > > > > | **Cosmos-Eval (SA-reason)** | 1 | 51m49s | 200 | **15.54 s** | 0.8637 |
> > > > > | Qwen-2.5-VL-7B (PC-reason) | 1 | 14m14s | 200 | 4.27 s | 0.2042 |
> > > > > | Qwen-2.5-VL-7B (SA-reason) | 1 | 13m34s | 200 | 4.07 s | 0.2219 |
> > > > >
> > > > > For PC rationales, Cosmos-Eval is in a similar range to Qwen-2.5-VL-7B. For SA rationales it is slower, mainly because Cosmos-Eval is trained to produce longer, more structured chains of thought and finer-grained SA explanations, rather than minimal one-line comments.
> > > > >
> > > > > When we compare these inference numbers with the cost of the teacher pipeline, the effect of distillation is more intuitive. Instead of \~106–145 seconds per sample to re-run Stage 0–2, Cosmos-Eval needs about **1.1 seconds** per sample for scores only (roughly 95× faster for PC and 130× faster for SA), and about **5.9–15.5 seconds** for scores \+ rationales (roughly 18× faster for PC and 9× faster for SA). In other words, Stage 0–2 have effectively been compressed into a single forward pass of Cosmos-Eval.
> > > > >
> > > > > In this sense, “lightweight” refers not to the training pipeline itself, but to the **deployed evaluator**: once distilled, Cosmos-Eval can be used as a practical SA/PC evaluator whose per-clip cost is close to a single 7B VLM call, while providing calibrated scores and physics-grounded explanations rather than scores alone. We hope this clarifies the computational overhead and addresses the reviewer’s concern more concretely.

---

> > > > > > ### Author Response · Authors · 2025-11-21
> > > > > >
> > > > > > # W6:Noisy Input Robustness Gap
> > > > > > We thank the reviewer for raising the concern regarding Cosmos-Eval’s robustness to noisy T2V outputs (e.g., motion blur, occlusion), which are indeed common in practice. Our goal is to evaluate T2V-generated videos for physical laws and semantic alignment with captions, and here we specifically test how Cosmos-Eval behaves when these T2V videos are synthetically degraded.
> > > > > >
> > > > > > ### **Noisy Input Robustness**
> > > > > >
> > > > > > We conducted experiments on **VideoPhy-2** videos where we introduced different types of noise — **motion blur**, **occlusion**, **compression artifacts**, **additive noise**, and **color distortion** — and compared model predictions before and after degradation. For each distortion type, we report the average scores for both **PC** and **SA**:
> > > > > >
> > > > > > | Distortion Type | Degraded Before (PC) | Degraded After (PC) | PC Δ (Before - After) | Degraded Before (SA) | Degraded After (SA) | SA Δ (Before - After) | Count |
> > > > > > | :---- | :---- | :---- | :---- | :---- | :---- | :---- | :---- |
> > > > > > | **Noise** | 4.56 | 4.28 | 0.28 | 4.03 | 3.72 | 0.31 | 100 |
> > > > > > | **Occlusion** | 4.59 | 4.21 | 0.38 | 4.02 | 3.82 | 0.20 | 100 |
> > > > > > | **Blur** | 4.64 | 4.32 | 0.32 | 4.00 | 3.72 | 0.28 | 100 |
> > > > > > | **Compression** | 4.67 | 4.45 | 0.22 | 4.02 | 3.77 | 0.25 | 100 |
> > > > > > | **Color** | 4.67 | 4.37 | 0.30 | 4.01 | 3.77 | 0.24 | 100 |
> > > > > >
> > > > > > ### **Degradation Hyperparameters**
> > > > > >
> > > > > > For completeness, the hyperparameters for each degradation type are:
> > > > > >
> > > > > > | Distortion Type | Parameter | Range | Description |
> > > > > > | :---- | :---- | :---- | :---- |
> > > > > > | **Noise** | strength | 10–40 | Noise intensity level |
> > > > > > |  | temporal | 5–15 | Temporal noise variation |
> > > > > > | **Blur** | sigma (luma) | 1.0–4.0 | Gaussian blur radius |
> > > > > > |  | chroma_radius | 0.5–2.0 | Chroma blur radius |
> > > > > > | **Compression** | CRF | 35–45 | Constant Rate Factor (higher = more compression) |
> > > > > > |  | bitrate | 100–300 kbps | Target video bitrate |
> > > > > > | **Occlusion** | num_boxes | 1–3 | Number of black occlusion boxes |
> > > > > > |  | box_size | 50–150 px | Width/height of each box |
> > > > > > |  | position (x, y) | 0–500 px | Random position within frame |
> > > > > > | **Color** | saturation | 0.3–1.5 | Saturation multiplier |
> > > > > > |  | contrast | 0.5–1.3 | Contrast factor |
> > > > > > |  | brightness | -0.2–0.2 | Brightness offset |
> > > > > > |  | hue | -30°–30° | Hue rotation angle |
> > > > > >
> > > > > > PC and SA share the same degradation settings with different random seeds (PC: 42, SA: 43), to keep the results reproducible but not identical.
> > > > > >
> > > > > > ### **Key Findings**
> > > > > >
> > > > > > 1. **PC task.** “Degraded Before (PC)” are predictions on the original high-quality T2V videos; “Degraded After (PC)” are predictions after applying noise. As expected, scores drop after degradation. The largest drop appears under **occlusion** (Δ ≈ 0.38), suggesting Cosmos-Eval is most sensitive to occluded content, while **compression** and **blur** cause smaller changes, indicating better robustness to these artifacts.
> > > > > >
> > > > > > 2. **SA task.** For SA, scores also decrease after degradation, with the largest drop under **additive noise** (Δ ≈ 0.31). This aligns with the intuition that strong noise and occlusion make it harder to judge semantic alignment from visual content alone.
> > > > > >
> > > > > > ### **Scope Clarification**
> > > > > >
> > > > > > We would like to emphasize that our current work is scoped to **T2V-generated videos**, which are typically of relatively high visual quality. The noisy inputs in this study are obtained by artificially degrading such T2V outputs, rather than by directly using arbitrary real-world videos with uncontrolled artifacts. Within this scope, Cosmos-Eval shows reasonable robustness: it reacts to distortions in a consistent way (scores decrease as videos become harder to interpret), but the changes are moderate rather than catastrophic across common degradation types.
> > > > > >
> > > > > > We see handling **truly in-the-wild, imperfect real-world videos** as a natural and important extension of this line of work, and we appreciate the reviewer’s suggestion for pushing Cosmos-Eval further in that direction.

---

> > > > > > > ### Author Response · Authors · 2025-11-21
> > > > > > >
> > > > > > > # Q1:Questions About LLM Self-Reinforcement Bias
> > > > > > >
> > > > > > > We thank the reviewer for raising this important concern about potential LLM self-reinforcement, i.e., the risk that our framework might favor “LLM-style” explanations over what humans actually find interpretable and useful. This point is very valuable to us, and it directly motivated the analysis we report below.
> > > > > > >
> > > > > > > Regarding **whether we have measured this bias against human interpretability needs**, we address a closely related concern in our response to **Weakness 2**, where we conducted a dedicated human evaluation of rationales:
> > > > > > >
> > > > > > > - We built a web interface to collect **human judgments** on SA/PC explanations produced by **Cosmos-Eval**, **Qwen3-VL-plus**, and **GPT-4V**.
> > > > > > > - The evaluation dimensions follow our SA/PC rationale rubrics (Tables 7–8 in the main paper), and in total we instantiated **1,500 questions** across SA and PC.
> > > > > > > - Annotators were assigned examples in a **double-blind** manner: they only saw the video, the caption (for SA), and the explanation, without knowing which model produced it.
> > > > > > > - For each dimension (e.g., Grounding, Temporal Alignment, Consistency, etc.), annotators scored the explanation on a \[0, 0.5, 1\] scale.
> > > > > > >
> > > > > > > Aggregated over all questions, Cosmos-Eval’s explanations are consistently rated higher than those of Qwen3-VL-plus and GPT-4V on both SA and PC dimensions. While this does not prove that all LLM-style biases are removed, it shows that our rationales are not optimized solely for another LLM judge: they are also preferred by human raters along explicitly human-designed criteria such as grounding, coverage, and justification.
> > > > > > >
> > > > > > > On the **closed-loop risk in the training pipeline**, we try to avoid a purely self-referential setup in two ways:
> > > > > > >
> > > > > > > 1. **Human-anchored scores in Stage 0\.**
> > > > > > >    Human 5-point SA/PC labels from **VideoPhy-2** are the primary anchors for score calibration. For **VideoPhy**, which does not provide 5-point labels, we use **VideoPhy-2-AutoEval** only to map those clips onto the same 1–5 scale so that the corpus is consistent. Thus, Cosmos-Eval is trained on a mixture of **human labels** (VideoPhy-2) and **VLM-as-judge labels** (VideoPhy via AutoEval), rather than being driven solely by an LLM judge.
> > > > > > >
> > > > > > > 2. **Avoiding a naive LLM→LLM loop.**
> > > > > > >    We deliberately avoid directly remapping VideoPhy’s human annotations via another LLM in a way that would collapse into a closed LLM feedback loop. Instead, VideoPhy-2’s human labels define the scale, and VideoPhy-2-AutoEval is used in a controlled manner to extend coverage where human scores are absent.
> > > > > > >
> > > > > > > We completely agree with the reviewer that these are only **first steps**, and that stronger human-in-the-loop calibration is an important direction. Concretely, we are planning the following extensions for future iterations of Cosmos-Eval:
> > > > > > >
> > > > > > > - **Ongoing human feedback:** continue collecting human ratings through our website interface, with a focus on difficult or controversial cases where LLM judges and humans disagree, and use this feedback to periodically re-calibrate both scoring and rationales.
> > > > > > > - **Human-written reference explanations:** introduce a small but carefully curated set of **human-authored SA/PC explanations** for subtle physics or semantics scenarios, and use them as corrective signals to pull the model away from explanations that are fluent but not actually helpful for human users.
> > > > > > > - **Targeted correction of failure modes:** when we identify typical “LLM-style but unhelpful” patterns (e.g., over-hedging, generic phrasing, or missing concrete visual details), we intend to explicitly counteract them with additional supervision derived from human feedback and curated examples.
> > > > > > >
> > > > > > > In summary, we have already taken steps to compare Cosmos-Eval’s explanations against human judgments and to avoid a purely closed LLM loop, but we view human-in-the-loop calibration as an ongoing process rather than a solved problem. We appreciate the reviewer’s question for highlighting this aspect so clearly; it aligns well with how we plan to maintain and improve Cosmos-Eval over time.

---

> > > > > > > > ### Author Response · Authors · 2025-11-21
> > > > > > > >
> > > > > > > > # Q2:Generalization to Complex Video Scenarios
> > > > > > > >
> > > > > > > > We thank the reviewer for raising this important question about generalization to long-duration videos and more complex physical interactions. This concern is closely related to what we discuss in Weakness 3, and we are happy to clarify it here.
> > > > > > > >
> > > > > > > > **Empirical evidence on extended content.**
> > > > > > > > Beyond VideoPhy-2, we conducted a long-horizon study using **LongCat-Video**, which produces ≈33s videos with multi-step, evolving motions. For 30 such prompts, we (i) evaluated each full ≈33s clip as a whole, and (ii) uniformly split each video into 11 non-overlapping 3s segments and scored them separately.
> > > > > > > >
> > > > > > > > For **Cosmos-Eval**, segment-wise SA and PC scores remained in narrow, stable ranges across all 11 segments (SA roughly 3.03–3.13; PC roughly 3.97–4.13), and the full-clip scores (SA ≈ 3.30, PC ≈ 4.00) were consistent with the segment averages. We also ran **VideoPhy-2-AutoEval** on the same long videos and observed a similar pattern of temporal stability (SA segments roughly 2.73–3.00; PC segments roughly 3.83–4.07, full-clip SA ≈ 2.97, PC ≈ 3.50). This suggests that Cosmos-Eval behaves comparably to the original VideoPhy-2 scorer on extended content and does not exhibit obvious drift or collapse as video length increases.
> > > > > > > >
> > > > > > > > **Anticipated technical challenges for more complex long videos.**
> > > > > > > > That said, we agree that truly complex long-horizon scenarios (e.g., minutes-long, densely interactive scenes) are more demanding than our current experiments. The main technical challenges we anticipate are:
> > > > > > > >
> > > > > > > > - **Trajectory extraction at scale:** For very long videos, naive dense sampling quickly becomes infeasible. We would need more hierarchical or event-triggered sampling (e.g., using coarse motion cues to select key segments) to maintain accurate physical reasoning without exploding compute.
> > > > > > > > - **Rationale organization over many events:** As the number of distinct interactions grows, flat, single-pass rationales become harder to read and harder to keep consistent. A natural extension is a hierarchical rationale structure (segment-level diagnostics summarized into clip-level conclusions), so that explanations remain concise while still covering long temporal dependencies.
> > > > > > > > - **Annotation and evaluation for long videos:** Human evaluation of long, complex clips is significantly more costly and noisy. Designing reliable rubrics and interfaces for long-horizon SA/PC judgments is itself a nontrivial challenge, and we see this as an important step for future work.
> > > > > > > >
> > > > > > > > In summary, we have already tested Cosmos-Eval on extended videos and found its behavior to be stable and comparable to the VideoPhy-2 scorer, as detailed in our response to Weakness 3\. At the same time, we acknowledge that fully addressing long, densely interactive scenarios will require more explicit work on hierarchical trajectory extraction, multi-scale rationales, and long-video annotation protocols.
> > > > > > > >
> > > > > > > > # Q3:Suggestions for Reducing Dependence on Predecessor Tools
> > > > > > > >
> > > > > > > > We appreciate the reviewer for raising the concern about the potential biases inherited from **VIDEOPHY-2-AUTOEVAL** in our framework. We would like to clarify that this concern is already addressed in **Table 1** of our paper, where we have conducted a comparison between the **VIDEOPHY-2-AUTOEVAL** scores and human-labeled ground truth for a subset of samples.
> > > > > > > >
> > > > > > > > This validation experiment was included in the initial submission and demonstrates that the initial scores provided by **VIDEOPHY-2-AUTOEVAL** align well with human annotations, thus quantifying the accuracy and reliability of these scores. The results from this comparison show that **VIDEOPHY-2-AUTOEVAL** provides a strong and trustworthy foundation for the initial scores in our pipeline.

---

> ### Author Response · Authors · 2025-11-27
>
> Dear Reviewer JK2k:
>
> We are grateful for your mandatory acknowedgement during this busy period. As the rebuttal period is ending soon, please be free to let us know whether your concerns have been addressed or not, and if there are any further questions.
>
> Thank you for your time and effort in reviewing our paper,
>
> Authors.

---

### Author Response · Authors · 2025-11-23

We would like to sincerely thank all reviewers for taking the time to evaluate our work and for providing many valuable and constructive comments. We deeply appreciate the effort you have invested in helping us improve this paper.

In particular, we thank **Reviewer Jk2k** for several constructive suggestions, such as highlighting the lack of long-video evaluation and encouraging us to incorporate human–AI collaboration into our framework. **Reviewer HYuZ** suggested incorporating factual consistency metrics. **Reviewer oRqx** recognized the practical value of our framework and further recommended analyzing the sensitivity to different backbone LLMs, reporting confidence intervals for our metrics, calibrating scores with human annotations, and adding more case studies. **Reviewer k1Yt** **recognized and appreciated the value of our explainable evaluation framework**, and suggested ranking existing T2V models and providing a more detailed efficiency analysis.

Finally, we are grateful to all reviewers for their comments on the effectiveness and overall clarity of our work. Based on the feedback, we have made the following major additions and revisions:

1. **Cosmos-Eval data construction and training**

   We start from **VideoPhy** and **VideoPhy2**, and build a large-scale *score + rationale* corpus through a multi-stage teacher pipeline. The multi-stage complex reasoning that produces these *score + rationale* pairs is then distilled into a single **7B VLM**, enabling interpretable evaluation of T2V videos within Cosmos-Eval.

2. **Additional datasets and methods**

   Following the reviewers’ suggestions, we have integrated additional benchmarks into our evaluation, including **AIGVE-Bench** and **LG-VQA**, to further validate the effectiveness and generalization ability of our model.

3. **Model bias and human evaluation**

   To better address concerns about model bias and the role of human evaluation, we designed detailed scoring rubrics and prompting templates to support consistent, multi-dimensional evaluation. Using this template, we conducted a **double-blind human study** with **1,500 multiple-choice questions**, constructed from the PC and SA dimensions in Tables 7 and 8, to assess the explanation quality of **Cosmos-Eval**, **GPT-4V**, and **Qwen-VL-Plus**. In addition, we used several strong closed-source models (**GPT-4V**, **Qwen3-VL-Plus**, **Gemini-2.5-Pro**) as judges to evaluate the explanations generated by Cosmos-Eval, GPT-4V, and Qwen-VL-Plus.

4. **Metrics**

   In response to **Reviewer HYuZ**, we have incorporated the **SummaC** factual consistency metric. Following **Reviewer oRqx**’s suggestion, we also report **confidence intervals for the Pearson correlation** between model scores and human scores.

5. **Ranking T2V models**

   Following **Reviewer k1Yt**’s suggestion, we provide a ranking of several existing **T2V models** based on our evaluation framework.

6. **Cost and efficiency analysis**

   In our responses to **Reviewers Jk2k and k1Yt**, we added a detailed analysis of the computational cost and efficiency of each stage of the pipeline, as well as training and inference. We show that **Cosmos-Eval** can serve as a practical **SA/PC evaluator**: the per-clip computational cost is comparable to a single call to a **7B VLM**, while additionally providing **calibrated scores** and **physics-grounded explanations**, rather than just scalar scores.

7. **Theoretical analysis of the framework**

   We now provide a more rigorous formal analysis of our multi-stage framework in **Appendix F**, clarifying its structure and justifying our design choices.

8. **Case studies**

   We have updated the manuscript with **14 additional qualitative case studies** in **Figures 11–24**, which clearly illustrate the evaluation texts produced by our model in concrete T2V scenarios.

Once again, we sincerely thank all reviewers for their thoughtful and constructive feedback. We have carefully considered and addressed each comment to better demonstrate the **effectiveness**, **practical value**, and **generality** of our framework.

---

### Author Response · Authors · 2025-12-01
**A General Response by Authors(part 1/2  )**

Dear Area Chairs and Reviewers,

We sincerely thank you for the time and effort you have devoted to reviewing our paper *Cosmos-Eval* and for the constructive feedback.

We received four reviews with initial ratings of **6, 4, 4, 4** (Reviewers k1Yt, JK2k, HYuZ, oRqx). During the rebuttal phase,we have provided detailed point-by-point responses and added new experiments, **which have addressed all weaknesses and questions.** Due to the OpenReview leak incident and the freeze of the discussion phase, **none of the reviewers could respond to our rebuttal or update their scores**.

Two reviewers explicitly indicated that they were open to **raising their scores** if their concerns were addressed:

- **Reviewer JK2k:**
  _“If the author can effectively solve my doubts, **I will consider improving my score.**”_

- **Reviewer HYuZ:**
  _“If the authors can reasonably and thoroughly address this concern, **I would be open to increasing my score.**”_

The other two reviewers (k1Yt and oRqx) mainly requested additional experiments and clarifications (more baselines, datasets, cost analysis), and both clearly recognized the importance and practicality of our framework. **We have substantially expanded our experiments and fully addressed their concerns** in the revised manuscript.

- **Reviewer oRqx:**
  *“Clear problem framing and practical value.”*

- **Reviewer k1Yt:**
  *“Important and timely problem.”*

Below we summarize (1) how our rebuttal addressed each reviewer’s concerns and (2) the main changes introduced in our response.

---

## 1. How our rebuttal addressed reviewers’ concerns

### **Reviewer JK2k (Rating: 4, Confidence: 4, **explicitly open to raising the score**)**

Reviewer JK2k requested positioning vs. prior work, evidence against LLM self-reinforcement, generalization beyond short/simple videos, clarification of VideoPhy-2-AutoEval, efficiency analysis, and robustness to noisy inputs.
In our rebuttal, we:

- Added cross-benchmark results on **AIGVE-Bench** and **LG-VQA-style**, further validating effectiveness and generalization.

- Added **human and multi-judge evaluations**, including **1,500 human-labeled comparison questions**, showing that Cosmos-Eval’s rationales are preferred, especially for PC, and performed **long-horizon experiments** on **LongCat-Video (~33s)** plus **noise/degradation tests**.

- Provided a **stage-wise cost breakdown** showing that the distilled Cosmos-Eval has practical inference-time cost comparable to a single 7B VLM call for scoring.

---

### **Reviewer HYuZ (Rating: 4, Confidence: 3, **explicitly open to raising the score**)**

Reviewer HYuZ asked about using closed-source VLMs for rationales, direct comparison to these VLMs’ explanations, metrics for explanation quality, and overall novelty/workload.
In our rebuttal, we:

- Conducted **human evaluation** of explanations from Cosmos-Eval vs. GPT-4V/Qwen3-VL-plus, collecting **1,500 human-labeled comparison questions**, and added multi-judge ratings (GPT-4V, Gemini-2.5-Pro), showing that Cosmos-Eval is competitive or superior, especially for PC.

- Augmented BLEU/BERTScore with Summac to better capture factual consistency and reasoning quality, and clarified the novelty and scope: explanation-centric SA/PC formulation, multi-stage pipeline (generation–verification–distillation), theoretical analysis, and ablations plus human/LLM-judge studies.

---

### **Reviewer oRqx (Rating: 4, Confidence: 3)**

Reviewer oRqx raised concerns about teacher–student bias, circularity from using LLM judges, lack of uncertainty estimates, and missing case studies.
In our rebuttal, we:

- Clarified that Stage-0 supervision is anchored on human VideoPhy-2 labels, with VideoPhy-2-AutoEval only used for scale alignment, and that evaluation is always against human labels.

- Performed **cross-judge robustness analyses** using independent LLMs, showing high agreement across judges, and reported Pearson r with 95% confidence intervals and effect sizes for SA/PC metrics.

- Added **qualitative case studies and full CoT traces** comparing Cosmos-Eval, GPT-4V, and Qwen3-VL-plus.

---

### **Reviewer k1Yt (Rating: 6, Confidence: 3)**

Reviewer k1Yt was overall **positive** and requested comparisons with frontier VLMs, direct benchmarking of T2V generators, broader dataset coverage, and clearer cost analysis.

In our rebuttal, we:

- Ran direct comparisons between Cosmos-Eval (7B), **GPT-4V** (GPT-4-based, large-scale closed-source VLM), and **Qwen3-VL-Plus** (large closed-source Qwen3-based VLM) on VideoPhy-2, showing that frontier VLMs excel at SA, while Cosmos-Eval better aligns on PC with more reference-like rationales.

- Used Cosmos-Eval to **benchmark multiple T2V models** on **AIGVE-Bench**, and expanded **cross-benchmark and long-horizon experiments**.

- Provided a **wall-clock and GPU cost analysis** contrasting the offline teacher pipeline with the efficient distilled student.

---

> ### Author Response · Authors · 2025-12-01
> **A General Response by Authors(part 2/2  )**
>
> ## 2\. Reviewers’ recognition of our work, summary of our response, and strengths of our work
>
> ### 2.1 Reviewers’ recognition of our work
>
> Taken together, the reviews highlight that:
>
> - The **gap between score-only evaluation and interpretable diagnostics** for T2V models is important, and an SA/PC evaluator with explicit rationales is valuable for debugging and deployment.
> - The **multi-stage pipeline** (generation–verification–distillation) is well structured and clearly motivated, with each component playing a distinct role.
> - The **empirical evaluation** is strong and carefully designed, with appropriate baselines, ablations, and implementation details that make the work reproducible.
> - The remaining comments mainly requested **additional experiments and clarifications** (e.g., more datasets, more metrics, cost analysis), rather than questioning the core novelty or practical usefulness of Cosmos-Eval.
>
> ### 2.2 Summary of our response
>
> Guided by the reviewers’ comments, we have made several key additions and revisions:
>
> - **Cosmos-Eval data construction, training, and theoretical grounding**
>   We clarified the multi-stage teacher pipeline built on VideoPhy and VideoPhy-2, and how complex reasoning that produces score+rationale pairs is distilled into a single 7B VLM, and added a formal analysis in Appendix F. (W4 for HYuZ)
>
> - **Additional benchmarks, long videos, and robustness**
>   We incorporated **AIGVE-Bench** and **LG-VQA-style**, performed **long-horizon evaluation** on **LongCat-Video (\~33s)**, and conducted robustness tests under synthetic degradations. These experiments show that Cosmos-Eval generalizes beyond VideoPhy-2 and behaves stably on extended and degraded content. (W3–W4/Q2 for JK2k; W2–W3 for k1Yt; W6 for JK2k)
>
> - **Model bias, human evaluation, and judge robustness**
>   We designed SA/PC rubrics and prompting templates, conducted a **double-blind human study** with 1,500 questions comparing explanations from Cosmos-Eval, GPT-4V, and Qwen3-VL-plus, and used strong models (GPT-4V, Qwen3-VL-plus, Gemini-2.5-Pro) as judges, while clarifying consensus extraction in Stage-1/2. (W2/Q1 for JK2k; W1–W3/Q1–Q3 for HYuZ; W1–W2 for oRqx)
>
> - **Improved metrics and statistical reporting**
>   Beyond BLEU and BERTScore, we incorporated the **Summac** factual-consistency metric and reported **confidence intervals** for Pearson correlations between model scores and human scores. (W3/Q3 for HYuZ; W3 for oRqx)
>
> - **Ranking T2V models under a unified protocol**
>   We provided an empirical ranking of several existing T2V models on **AIGVE-Bench** using Cosmos-Eval’s PC scores, showing how the framework can compare and benchmark T2V generators. (W2 for k1Yt)
>
> - **Cost and efficiency analysis**
>   We added a detailed analysis of computational cost and efficiency for each stage, as well as for training and inference, showing that the deployed student offers explanation-centric SA/PC evaluation at **per-clip cost comparable to a single 7B VLM call**. (W5 for JK2k; W4 for k1Yt)
>
> - **Expanded qualitative case studies**
>   We updated the manuscript with **14 additional qualitative case studies**, including direct comparisons between Cosmos-Eval, GPT-4V, and Qwen3-VL-plus and full CoT traces. (W4 for oRqx; Q2 for HYuZ)
>
> Overall, these revisions systematically address the weaknesses and questions raised by all four reviewers and are reflected in the revised manuscript.
>
> ### 2.3 The strengths of our work
>
> We summarize the main strengths of *Cosmos-Eval* as follows:
>
> - **Explanation-centric SA/PC evaluation for T2V**: jointly predicts 5-point SA/PC scores and physics-grounded rationales, turning numeric metrics into an interpretable diagnostic tool.
> - **Structured multi-stage teacher–student pipeline with efficient deployment**: aggregates signals from multiple VLMs and judges into a single student model with practical inference-time cost.
> - **Extensive empirical, human, and cross-benchmark validation with practical impact**: experiments on multiple benchmarks, plus human and multi-judge evaluations, show that Cosmos-Eval can robustly benchmark and analyze modern T2V generators.
>
> We hope this context helps the Area Chairs and reviewers interpret the current reviews in light of the rebuttal, given that the discussion phase was cut short by the OpenReview incident. Thank you again for your time and consideration.
>
> Best regards,
> Authors of submission *\#16433*

---

### Meta-Review · Area_Chair_TXHT · 2026-01-15

**Summary:**

This paper introduces an evaluation framework to jointly assess semantic adherence and physical consistency of generated videos. It received three negative scores with one positive score. I appreciate authors' detailed responses and refine many sections of the paper. But, there are still many questions to be addressed.

**Reviewer Concerns:**

Some concerns are addressed, but there are still many problems, such as (1) the authors introduce human evaluation results to respond to HYuZ and JK2k's main concerns, but it is a bit unreliable without human evaluation details. The author use old closed-source VLMs, GPT-4V (should be a 2024), for comparison, (2) the author evade evaluate state-of-the-art T2V models (e.g., sora2 and veo3.1) and use some old models to prevaricate, (3) the author evade to compare Cosmos-Eval with state-of-the-art VLMs such as GPT-4o, GPT-5. (4) judge models' biases and robustness, (5) dependence on videophy-2-autoeval.

**Reviewer Scores:**

Reviewer oRqx, JK2k, and HYuZ may keep negative scores since there are still many concerns are not well addressed.

---

### Decision · Program_Chairs · 2026-01-26

Reject